# Learning with Restricted Boltzmann Machines: Asymptotics of AMP and GD in High Dimensions

**Yizhou Xu**
Information, Learning & Physics Laboratory
Statistical Physics of Computation Laboratory
EPFL, Switzerland

**Florent Krzakala**
Information, Learning & Physics Laboratory
EPFL, Switzerland

**Lenka Zdeborová**
Statistical Physics of Computation Laboratory
EPFL, Switzerland

## Abstract

The Restricted Boltzmann Machine (RBM) is one of the simplest generative neural networks capable of learning input distributions. Despite its simplicity, the analysis of its performance in learning from the training data is only well understood in cases that essentially reduce to singular value decomposition of the data. Here, we consider the limit of a large dimension of the input space and a constant number of hidden units. In this limit, we simplify the standard RBM training objective into a form that is equivalent to the multi-index model with non-separable regularization. This opens a path to analyze training of the RBM using methods that are established for multi-index models, such as Approximate Message Passing (AMP) and its state evolution, and the analysis of Gradient Descent (GD) via the dynamical mean-field theory. We then give rigorous asymptotics of the training dynamics of RBMs on data generated by the spiked covariance model as a prototype of a structure suitable for unsupervised learning. We show in particular that RBMs reach the optimal computational weak recovery threshold, aligning with the Baik-Ben Arous-Péché (BBP) transition, in the spiked covariance model.

## 1 Introduction

The Restricted Boltzmann Machine (RBM) [Hinton, 2002, Salakhutdinov et al., 2007] is a foundational generative model that effectively learns input distributions in an unsupervised manner and can be extended to multi-layer architectures [Salakhutdinov and Hinton, 2009]. RBMs have significantly influenced machine learning development and, despite not being state-of-the-art for e.g.,image generation, remain of scientific interest and can enhance interpretability [Hu et al., 2018]. They are in particular very popular in modeling the quantum wave function in many-body quantum problems [Carleo and Troyer, 2017, Melko et al., 2019, Krenn et al., 2023, Nys et al., 2024] in physics and in modeling protein folding [Morcos et al., 2011, Muntoni et al., 2021].

In the wake of generative AI's recent successes, achieving a precise mathematical understanding of how RBMs learn from data is a natural first step toward tackling the more intricate dynamics of modern architectures. Yet, the learning behavior of RBMs remains poorly understood, primarily due to the intractability of the likelihood function. In practice, stochastic approximations such as contrastive divergence are widely used. While exactly solvable models—such as Gaussian-Gaussian and Gaussian-Spherical RBMs [Karakida et al., 2016, Decelle and Furtlehner, 2020, Genovese and Tantari, 2020]—are well understood, they essentially reduce to singular value decomposition (SVD) and fail to capture the mechanisms underlying empirical performance. Heuristic analyses using the

replica method also exist in the statistical physics literature [Alemanno et al., 2023, Thériault et al., 2024, Manzan and Tantari, 2024], but in addition to their non-rigorous nature, they often fail to reflect the actual training procedure used in practice—namely, maximum likelihood estimation. Instead, they focus on Bayesian posterior sampling, which yields qualitatively different predictions. The only works studying the full learning on the usual likelihood objective is as far as we know [Bachtis et al., 2024, Harsh et al., 2020], and we will comment on the differences with our work below.

This gap in our theoretical understanding of RBMs stands in stark contrast to recent progress on feed-forward networks, a field that has seen a surge of analytical results, especially in high-dimensional regimes with synthetic data, where rigorous insights are increasingly available, see e.g.,Soltanolkotabi et al. [2018], Mei et al. [2019], Celentano et al. [2020], Gerbelot et al. [2024], Bietti et al. [2023].

Motivated by this progress, we analyze empirical likelihood maximization for RBMs directly, in a setting that mirrors practical training procedures. Specifically, we consider the high-dimensional limit in which both the data dimension and the number of samples tend to infinity, while the number of hidden units remains fixed. This regime is relevant for many real-world applications, where data often lies on low-dimensional manifolds embedded in high-dimensional spaces. In this limit, the likelihood function admits a simplified form that reduces to an unsupervised variant of the multi-index model—a framework that has proven useful for studying neural networks and gradient descent in non-convex, high-dimensional settings [Saad and Solla, 1995, Abbe et al., 2023, Damian et al., 2023, Troiani et al., 2024]. Leveraging this connection, one can characterize the learning dynamics and derive sharp asymptotic results for data generated by the celebrated spiked covariance model [Johnstone, 2001].

**Main results —** Our contributions are the following:

i) We prove that, in high dimensions and with finitely many hidden units, the RBM training objective is asymptotically equivalent to an unsupervised multi-index model with non-separable regularization. This characterization is key to our analysis, as it allows the use of established mathematical techniques for synthetic data. As a model for data, we consider the well-studied spiked covariance model and emphasise its equivalence to a teacher RBM.

ii) Thanks to this mapping, we show that one can rigorously analyze the asymptotics of RBMs trained on the spiked covariance data. We apply a large body of recent results from multi-index and supervised models to RBMs. In particular, we derive sharp asymptotics for the simplified objective and introduce an Approximate Message Passing (AMP) algorithm to train the model. This allows us to characterize the global optimum of the objective for a number of cases. We establish that the weak recovery threshold of RBMs matches the optimal BBP threshold ([Baik et al., 2005], i.e., the minimal signal-to-noise ratio to detect the signal).

iii) We provide rigorous, closed-form equations describing the exact high-dimensional asymptotics of gradient descent dynamics. These results generalize dynamical mean-field theory, which was previously limited to multi-index and tensor learning problems.

Overall, our analysis opens the door to a precise mathematical understanding of unsupervised learning in RBMs, a foundational generative model. By mapping the likelihood landscape of RBMs in the high-dimensional regime to that of an effective model, we are able to import a range of tools and results that were previously restricted to supervised learning in single- and two-layer networks. This connection not only yields sharp predictions for optimization and dynamics in RBMs, but also lays the groundwork for extending these techniques to more complex generative architectures such as diffusion models.

**Related works —** Decelle et al. [2017, 2018] analyze the learning dynamics of RBMs, primarily focusing on the linear regime, which corresponds to Gaussian-Gaussian RBMs. In the non-linear regime, their analysis relies on the assumption that the weights remain sampled from a spiked ensemble throughout training. However, this assumption breaks down in practice, as the training process induces strong correlations among the weights. Certain versions of RBMs have been analyzed within the framework of the Hopfield model [Barra et al., 2012, Mézard, 2017, Barra et al., 2018], but for fixed weights (patterns). A series of works have employed heuristic statistical mechanics techniques—such as the replica method—to study the training of RBMs in teacher-student settings [Huang and Toyoizumi, 2016, Huang, 2017, Hou et al., 2019, Alemanno et al., 2023, Thériault et al., 2024, Manzan and Tantari, 2024]. However, their approach is closer to Bayesian sampling of the posterior over weights and deviates from the standard likelihood maximization typically used in practice (see Appendix A). Additionally, these studies do not frame their results in terms of spiked covariance models, thereby missing the clear connection to the broader literature on learning in

high dimensional statistics. In contrast, we study the actual gradient descent algorithm applied to the maximum likelihood objective, and we rigorously demonstrate that RBMs achieve the optimal weak recovery threshold of the spiked covariance model. Recent works [Bachtis et al., 2024, Harsh et al., 2020] study the early training dynamics of RBMs in a similar setting (large dimension, few hidden units) and offer valuable heuristic insights into GD dynamics over time,. Their analysis is non-rigorous, whereas our results provide exact asymptotics, including the characterization of saddle points; they also do not connect to multi-index models, AMP or the spiked covariance model.

Approximate Message Passing (AMP) has become a tool for the rigorous analysis of high-dimensional statistical inference and machine learning. It provides sharp asymptotics of both Bayesian inference and empirical risk minimization in a wide range of models, including spiked matrices and tensors [Donoho et al., 2009, Javanmard and Montanari, 2013, Montanari and Richard, 2014, Lesieur et al., 2017, Celentano et al., 2021, Alaoui et al., 2023]. In this work, we use AMP as a proof technique to analyze the asymptotics of RBMs, both in the context of empirical risk minimization and in the study of gradient descent dynamics. This leverages on many recent progresses Celentano et al. [2020], Fan [2022], Gerbelot et al. [2022, 2024]. Note that while variants of AMP have been used in the context of RBM training in prior works—e.g., [Gabrié et al., 2015, Tramel et al., 2018]—they were employed for sampling from the posterior distribution over fixed weights, rather than for directly training the weights. This stands in contrast to our approach, where AMP is used to analyze and characterize the actual learning dynamics of RBMs under empirical risk minimization.

Multi-index models have been the subject of extensive recent work; see, for example, [Abbe et al., 2023, Damian et al., 2023, Bietti et al., 2023, Troiani et al., 2024] and references therein. Spiked covariance models have also been studied in detail from multiple perspectives, including information-theoretic limits [Lelarge and Miolane, 2017, Miolane, 2017, El Alaoui et al., 2020], energy landscape analysis [Arous et al., 2019], and spectral methods [Baik et al., 2005].

**Notations —** We use the bold uppercase of a letter to denote a matrix, its bold lowercase to denote the corresponding row vectors, and its lowercase to denote an element. For example, we use $\boldsymbol{X} \in \mathbb{R}^{n \times d}$ to denote the data, $\boldsymbol{x}_\mu \in \mathbb{R}^d$ to denote the $\mu$−th data point, and $x_{\mu i}$ to denote the $i$−th component of the $\mu$−th data point. We use uppercase letters to denote random variables and bold uppercase letters to denote vectors of random variables. We use $||\cdot||$ to denote the $L^2$ norm, $||\cdot||_F$ to denote the Frobenius norm and $||\cdot||_r$ to denote the $L_r$ norm.

## 2 Settings

A Restricted Boltzmann Machine (RBM) learns an underlying distribution from $n$ samples of $d$-dimensional (usually binary) data $\boldsymbol{X} \in \mathbb{R}^{n \times d}$. We will consider an RBM with $d$ visible and $k$ hidden units represented by the following probability distribution

$$\mathrm{d}P(\boldsymbol{v}, \boldsymbol{h}) = \frac{1}{Z(\boldsymbol{W}, \boldsymbol{\theta}, \boldsymbol{b})} e^{\frac{1}{\sqrt{d}} \boldsymbol{v}^\top \boldsymbol{W} \boldsymbol{h} + \frac{1}{\sqrt{d}} \boldsymbol{\theta}^\top \boldsymbol{v} + \boldsymbol{b}^\top \boldsymbol{h}} \mathrm{d}P_v(\boldsymbol{v}) \mathrm{d}P_h(\boldsymbol{h}), \tag{1}$$

where visible and hidden units are $\boldsymbol{v} \in \mathbb{R}^d, \boldsymbol{h} \in \mathbb{R}^k$, learnable parameters are the biases $\boldsymbol{b} \in \mathbb{R}^k$, $\boldsymbol{\theta} \in \mathbb{R}^d$ and the weights $\boldsymbol{W} \in \mathbb{R}^{d \times k}$, and $Z(\boldsymbol{W}, \boldsymbol{\theta}, \boldsymbol{b})$ is the normalization factor (the partition function)

$$Z(\boldsymbol{W}, \boldsymbol{\theta}, \boldsymbol{b}) := \int \mathrm{d}P_v(\boldsymbol{v}) \mathrm{d}P_h(\boldsymbol{h}) \, e^{\frac{1}{\sqrt{d}} \boldsymbol{v}^\top \boldsymbol{W} \boldsymbol{h} + \frac{1}{\sqrt{d}} \boldsymbol{\theta}^\top \boldsymbol{v} + \boldsymbol{b}^\top \boldsymbol{h}}. \tag{2}$$

The scaling $1/\sqrt{d}$ is added to keep the energy per data point of order 1 in the large-dimensional limit. $P_v$ (over $\mathbb{R}^d$) and $P_h$ (over $\mathbb{R}^k$) represent the prior distributions of visible and hidden units, respectively.

RBMs are most commonly trained by maximizing the likelihood (marginalized over the hidden units) of the training set, which means that the weight matrix is selected in a way to maximize

$$\mathcal{L}(\boldsymbol{W}, \boldsymbol{\theta}, \boldsymbol{b}) = \frac{1}{Z(\boldsymbol{W}, \boldsymbol{\theta}, \boldsymbol{b})^n} \prod_{\mu=1}^n \left[ \int \mathrm{d}P_h(\boldsymbol{h}) \, e^{\frac{1}{\sqrt{d}} \boldsymbol{x}_\mu^\top \boldsymbol{W} \boldsymbol{h} + \frac{1}{\sqrt{d}} \boldsymbol{\theta}^\top \boldsymbol{x}_\mu + \boldsymbol{b}^\top \boldsymbol{h}_\mu} \right]. \tag{3}$$

Here $Z(\boldsymbol{W}, \boldsymbol{\theta}, \boldsymbol{b})$ is the normalization (2). Note that the factor $P_v(\boldsymbol{x}_\mu)$ is dropped because it does not depend on the training parameters $\boldsymbol{W}, \boldsymbol{\theta}, \boldsymbol{b}$.

We will consider the limit $d \to \infty$ and $n(d) = \alpha d$, with a fixed sampling rate $\alpha = \Theta(1)$ and a fixed number of units $k = \Theta(1)$.

## 3 Equivalent effective objective in high-dimension

Our first result is that the arguably complicated likelihood of the RBM is asymptotically equivalent, in high-dimension and with any finite number of hidden units, to that of an effective, simpler model. Interestingly, this result holds for any data (not only Gaussian ones).

We assume that the RBM satisfies the following common assumption:

**Assumption 1.** $P_v$ *is a factorizable (i.e.,* $P_v(\boldsymbol{v}) = \Pi_{i=1}^d P_v(v_i)$*) distribution with a bounded support, zero mean and unit variance.* $P_h$ *is a bounded distribution.*

We expect that the bounded assumption and unit variance are purely technical and can be relaxed, e.g., to sub-Gaussian. However, the simplification of the log-likelihood in Theorem 1 requires the prior of the visible units $P_v$ to have zero mean, which is acceptable as we can train the biases.

**Theorem 1.** *Under Assumption 1, we have*

$$\lim_{d\to\infty} \frac{1}{d} |\log \mathcal{L}(\boldsymbol{W}, \boldsymbol{\theta}, \boldsymbol{b}) - \log \tilde{\mathcal{L}}(\boldsymbol{W}, \boldsymbol{\theta}, \boldsymbol{b})| = 0, \tag{4}$$

*where*

$$\log \tilde{\mathcal{L}}(\boldsymbol{W}, \boldsymbol{\theta}, \boldsymbol{b}) := \sum_{\mu=1}^n \eta_1\left(\frac{1}{\sqrt{n}}\boldsymbol{W}^\top \boldsymbol{x}_\mu, \frac{1}{\sqrt{n}}\boldsymbol{\theta}^\top \boldsymbol{x}_\mu, \boldsymbol{b}\right) - n\eta_2\left(\frac{1}{d}\boldsymbol{W}^\top \boldsymbol{W}, \frac{1}{d}\boldsymbol{W}^\top \boldsymbol{\theta}, \frac{1}{d}\boldsymbol{\theta}^\top \boldsymbol{\theta}, \boldsymbol{b}\right) \tag{5}$$

*is the effective log-likelihood function, with $\eta_1 : \mathbb{R}^k \times \mathbb{R} \times \mathbb{R}^k \to \mathbb{R}$ defined as*

$$\eta_1(\boldsymbol{x}_W, x_\theta, \boldsymbol{b}) := \log \int \mathrm{d}P_h(\boldsymbol{h}) e^{\boldsymbol{h}^\top (\sqrt{\alpha}\boldsymbol{x}_W + \boldsymbol{b})} + \sqrt{\alpha} x_\theta \tag{6}$$

*and $\eta_2 : \mathbb{R}^{k\times k} \times \mathbb{R}^k \times \mathbb{R} \times \mathbb{R}^k \to \mathbb{R}$ defined as*

$$\eta_2(\boldsymbol{Q}_W, \boldsymbol{Q}_{W\theta}, Q_\theta, \boldsymbol{b}) := \log \int \mathrm{d}P_h(\boldsymbol{h}) e^{\boldsymbol{h}^\top \boldsymbol{b} + (\boldsymbol{h}^\top \boldsymbol{Q}_W \boldsymbol{h} + 2\boldsymbol{h}^\top \boldsymbol{Q}_{W\theta} + Q_\theta)/2}. \tag{7}$$

*The limit in (4) holds for any sequence $\boldsymbol{X}(d) \in \mathbb{R}^{n\times d}$, and $\boldsymbol{W}(d) \in \mathbb{R}^{d\times k}, \boldsymbol{\theta}(d) \in \mathbb{R}^d, \boldsymbol{b}(d) \in \mathbb{R}^k$ with $\{\frac{1}{d}||\boldsymbol{W}(d)||_F^2, \frac{1}{d}||\boldsymbol{\theta}(d)||^2, ||\boldsymbol{b}(d)||^2\}_{d\geq 1}$ bounded.*

The proof of Theorem 1 is presented in Appendix B.1, where we also discuss visible units with non-zero mean. Theorem 1 maps RBMs to (5), an unsupervised version of the multi-index model for which the likelihood function depends only on $k = \Theta(1)$ projections of the weights on the high-dimensional data, i.e., on $\{\boldsymbol{w}_i^T \boldsymbol{x}\}_{i=1}^k$), see e.g., [Troiani et al., 2024]. The model has a non-separable regularization term $\eta_2$, which only depends on the type of the hidden units. The analysis of the multi-index model with such a penalty $\eta_2$ poses a technical challenge that we resolve in this paper.

**Gradients and expectations** One can further interpret the simplified model through its gradient, which is what matters for training. For simplicity, we consider biasless RBMs here. See Appendix B.3 for more details. The gradient of (5) reads

$$\nabla_{\boldsymbol{W}} \log \tilde{\mathcal{L}}(\boldsymbol{W}, \boldsymbol{\theta}, \boldsymbol{b}) = \sum_{\mu=1}^n \frac{1}{\sqrt{n}} \nabla \eta_1(\boldsymbol{x}_\mu^\top \boldsymbol{W}/\sqrt{n}) \otimes \boldsymbol{x}_\mu - \alpha \nabla \eta_2(\boldsymbol{Q}(\boldsymbol{W}))\boldsymbol{W}^\top, \tag{8}$$

where we denote $\boldsymbol{Q}(\boldsymbol{W}) := \frac{1}{d}\boldsymbol{W}^\top \boldsymbol{W} \in \mathbb{R}^{k\times k}$. This is particularly interesting when contrasted with the original gradient used in standard training techniques. Indeed, the gradient of the original RBM model (3) can be written as the difference between the so-called clamped average $\langle\cdot\rangle_D$ (where the visible units $\mathbf{v}$ are fixed by the dataset and the hidden units are sampled conditionally) and the model average $\langle\cdot\rangle_{\mathrm{model}}$ [Decelle and Furtlehner, 2021]:

$$\nabla_{\boldsymbol{W}} \log \mathcal{L}(\boldsymbol{W}, \boldsymbol{\theta}, \boldsymbol{b}) := \sum_{\mu=1}^n \frac{1}{\sqrt{d}}\langle \boldsymbol{h}\boldsymbol{x}_\mu^\top\rangle_D - \frac{n}{\sqrt{d}}\langle \boldsymbol{h}\boldsymbol{v}^\top\rangle_{\mathrm{model}}. \tag{9}$$

In the effective model, while the first (clamped) term—which is trivial to compute—remains, the second term (which a priori requires a high-dimensional integration in $d$ dimensions) is replaced by

$$\nabla_{\boldsymbol{W}} \log \tilde{\mathcal{L}}(\boldsymbol{W}, \boldsymbol{\theta}, \boldsymbol{b}) := \sum_{\mu=1}^n \frac{1}{\sqrt{d}}\langle \boldsymbol{h}\boldsymbol{x}_\mu^\top\rangle_D - \alpha \langle \boldsymbol{h}\boldsymbol{h}^\top\rangle_H \boldsymbol{W}^\top. \tag{10}$$

This is a drastic simplification that arises in the large-dimensional limit, at least when the number of hidden units remains moderate. In fact, the average $\langle \mathbf{h}\mathbf{h}^\top \rangle_H$ can be computed exactly even with a moderately large number of hidden units (e.g., around 20). The key point is that we have replaced a challenging high-dimensional sampling problem with a much simpler one in lower dimension—namely, in the hidden space.

**Training with the equivalent objective —** Before turning to our theoretical analysis, we note that the aforementioned equivalence can also be used in practice, which is of independent interest. We illustrate this on a number of toy problems in Appendix E, where we show that it is possible to train an RBM using the simplified effective objective and obtain results comparable to those achieved with standard training methods. This highlights not only the practical relevance of our analysis, but also the potential of the effective model to serve as an alternative to traditional training.

## 4 Spiked covariance model

Thanks to the equivalence with the effective model, we are now in a position to present a set of mathematical results that enable us to transfer insights and techniques developed for multi-index models, and shallow supervised architectures to the study of RBMs' learning. For this we must model the input data. The canonical model of high-dimensional data for unsupervised learning is the spiked covariance model Johnstone [2001], where $n$ samples of $d$-dimensional data are generated as follows:

$$\boldsymbol{X} = \frac{1}{\sqrt{d}}\boldsymbol{U}^*\boldsymbol{\Lambda}(\boldsymbol{W}^*)^\top + \boldsymbol{Z} \in \mathbb{R}^{n \times d}, \tag{11}$$

where $\boldsymbol{U}^* \in \mathbb{R}^{n \times r}, \boldsymbol{W}^* \in \mathbb{R}^{d \times r}$ are hidden signals, and $\boldsymbol{\Lambda} \in \mathbb{R}^{r \times r}$ is diagonal, representing the signal-to-noise ratio. $\{Z_{ij}\}_{i,j=1}^{n,d} \overset{iid}{\sim} \mathcal{N}(0,1)$ represent the noise. We further suppose that hidden spikes are sampled from $\{\boldsymbol{u}_i^*\}_{i=1}^n \overset{iid}{\sim} P_u$ and $\{\boldsymbol{w}_i^*\}_{i=1}^d \overset{iid}{\sim} P_w$, where $P_u, P_w$ are some distributions over $\mathbb{R}^r$. We will assume that $r = \Theta(1)$ is fixed when $n, d$ go to infinity. Our results also apply to a slightly more general data model with non-linearity and non-Gaussian noise. See Appendix C.1. As the spiked covariance model is the simplest unsupervised model, any good unsupervised method for the data from the model has to be able to recover in some way the latent structure, i.e., the variables $\boldsymbol{U}^*$ and $\boldsymbol{W}^*$. For a special case we can map the spiked covariance model to a teacher RBM. Consider the spiked covariance model (11) with $\boldsymbol{\Lambda} = \lambda\boldsymbol{I}$. The distribution of $\boldsymbol{X}$ conditioned on $\boldsymbol{W}^*$ reads

$$\mathbb{P}(\boldsymbol{X}|\boldsymbol{W}^*) = \frac{1}{\sqrt{2\pi}}\mathbb{E}_{\boldsymbol{U}^*}\exp\left\{-\frac{1}{2}\left\|\boldsymbol{X} - \frac{1}{\sqrt{d}}\boldsymbol{U}^*\boldsymbol{\Lambda}(\boldsymbol{W}^*)^\top\right\|_F^2\right\}$$

$$= \frac{1}{\sqrt{2\pi}}\mathbb{E}_{\boldsymbol{U}^*}\Pi_{\mu=1}^n\exp\left\{-\frac{1}{2}\|\boldsymbol{x}_\mu\|^2 + \frac{\lambda}{\sqrt{d}}\sum_{a,i=1}^r x_{\mu i}u_{\mu a}^*w_{ai}^* - \frac{\lambda^2}{2d}\sum_{i=1}^n(\sum_{a=1}^r u_{\mu a}^*w_{ai}^*)^2\right\}. \tag{12}$$

Suppose that $\frac{1}{d}\sum_{i=1}^n w_{ai}^*w_{bi}^* \approx 0$, i.e., teacher weights are approximately orthogonal, and that $u_{\mu a}^* \in \{-1, 1\}$. Then we have $\mathbb{P}(\boldsymbol{X}|\boldsymbol{W}^*) \approx \Pi_{\mu=1}^n\mathbb{P}(\boldsymbol{x}_\mu|\boldsymbol{W}^*)$, where

$$\mathbb{P}(\boldsymbol{x}_\mu|\boldsymbol{W}^*) := \frac{1}{\mathcal{Z}(\boldsymbol{W}^*)}\mathbb{E}_{\boldsymbol{U}^*}\exp\left\{-\frac{1}{2}\|\boldsymbol{x}_\mu\|^2 + \frac{\lambda}{\sqrt{d}}\sum_{a,i=1}^r x_{\mu i}u_{\mu a}^*w_{ai}^*\right\}. \tag{13}$$

Meanwhile, for the (teacher) RBM we have $\mathbb{P}(\boldsymbol{v}|\boldsymbol{W}^*) = \frac{P_v(\boldsymbol{v})}{Z(\boldsymbol{W}^*)}\mathbb{E}_{\boldsymbol{h}}e^{\frac{1}{\sqrt{d}}\boldsymbol{v}^\top\boldsymbol{W}^*\boldsymbol{h}}$ by (1), where we remove the biases and use $\boldsymbol{W}^*$ to represent the weights of the (teacher) RBM. This is consistent with the conditional distribution of RBMs analyzed in Huang and Toyoizumi [2016], Huang [2017], Hou et al. [2019], Alemanno et al. [2023], Thériault et al. [2024], Manzan and Tantari [2024]

$$\mathbb{P}(\boldsymbol{v}|\boldsymbol{W}^*) = \frac{P_v(\boldsymbol{v})}{Z_\beta(\boldsymbol{W}^*)}\mathbb{E}_{\boldsymbol{h}}e^{\frac{\beta^*}{\sqrt{d}}\boldsymbol{v}^\top\boldsymbol{W}^*\boldsymbol{h}}, \tag{14}$$

where the additional parameter $\beta^*$ denotes the inverse temperature of the (teacher) RBM.

Therefore, we can map (13) to (14) through the mapping $\boldsymbol{x}_\mu, \boldsymbol{u}_\mu^*, \lambda \to \boldsymbol{v}, \boldsymbol{h}, \beta^*$ and $\mathrm{d}P(\boldsymbol{v}) \propto e^{-\frac{1}{2}\|\boldsymbol{v}\|^2}$. Thus, a spiked covariance model with Bernoulli priors can be regarded as a teacher RBM

---

**Algorithm 1** AMP-RBM

---

**Require:** Initialization $\boldsymbol{U}^0 = 0, \boldsymbol{C}_0 = 0, \hat{\boldsymbol{Q}}^0 = \frac{1}{2}I_k, \boldsymbol{Z}^1 \in \mathbb{R}^{d \times k}$ and damping $\zeta \in [0, 1)$

    **for** $t = 1, 2, \cdots, T$ **do**

        $\boldsymbol{W}^t = f(\boldsymbol{Z}^t, \hat{\boldsymbol{Q}}^t, \boldsymbol{C}_{t-1}) \in \mathbb{R}^{d \times k}, \; \boldsymbol{B}_t = \frac{1}{n} \sum_{i=1}^d \frac{\partial f}{\partial \boldsymbol{z}_i^t}(\boldsymbol{z}_i^t, \hat{\boldsymbol{Q}}^t) \in \mathbb{R}^{k \times k}.$

        $\boldsymbol{Y}^t = \frac{1}{\sqrt{n}} \boldsymbol{X} \boldsymbol{W}^t - \boldsymbol{U}^{t-1} \boldsymbol{B}_t^\top \in \mathbb{R}^{n \times k}.$

        $\boldsymbol{U}^t = g(\boldsymbol{Y}^t, \boldsymbol{B}_t) \in \mathbb{R}^{n \times k}, \; \boldsymbol{C}_t = \frac{1}{n} \sum_{\mu=1}^n \frac{\partial g}{\partial \boldsymbol{y}_\mu^t}(\boldsymbol{y}_\mu^t) \in \mathbb{R}^{k \times k}.$

        $\boldsymbol{Z}^{t+1} = \frac{1}{\sqrt{n}} \boldsymbol{X}^\top \boldsymbol{U}^t - \boldsymbol{W}^t \boldsymbol{C}_t^\top \in \mathbb{R}^{d \times k}.$

        $\hat{\boldsymbol{Q}}^{t+1} = \zeta \hat{\boldsymbol{Q}}^t + (1 - \zeta)\nabla \eta_2(\boldsymbol{Q}(\boldsymbol{W}^t)) \in \mathbb{R}^{k \times k}.$

    **end for**

---

with orthogonal teacher weights, Bernoulli hidden units, and Gaussian visible units, where $\lambda$ can be understood as the inverse temperature, i.e., the strength of the teacher features. However, we emphasize that none of the existing literature concerning the teacher-student RBMs considers the practical training scheme (i.e., maximizing the log-likelihood). See discussions in Appendix A.

The large-dimensional limit of the data model corresponds to an RBM with a large number $d$ of visible units, $k = \Theta(1)$ hidden units learning from a number of samples proportional to the dimension $n = \alpha d$, and the samples have $r = \Theta(1)$ features. Note that we do not require $k = r$ to account for the overparameterized and underparameterized schemes.

## 5  Main theorems and technical results

Now consider synthetic spiked data $\boldsymbol{X}$ as defined in (11) and the optimization of (5):

$$\max_{\boldsymbol{W}} \log \tilde{\mathcal{L}}(\boldsymbol{W}) := \max_{\boldsymbol{W}} \left( \sum_{\mu=1}^n \eta_1(\boldsymbol{x}_\mu^\top \boldsymbol{W}/\sqrt{n}) - n\eta_2(\boldsymbol{Q}(\boldsymbol{W})) \right) \tag{15}$$

for a general choice of $\eta_1$ and $\eta_2$, including the ones defined in (6) and (7) but with all parameters except the first set to zero. We drop the biases $\boldsymbol{\theta}, \boldsymbol{b}$ for simplicity, because it is straightforward to modify our results to include them.

**AMP-RBM —** We first present AMP-RBM as a practical algorithm to optimize (15). While it is common to use AMP for supervised generalized linear regression [Donoho et al., 2009, Javanmard and Montanari, 2013], the novelty is to use a variant of this algorithm to train the unsupervised RBM for spiked data, and its rigorous analysis. Other new aspects are the spiked structure of the input data, and the penalty term $\eta_2$. To deal with $\eta_2$, we introduce the Lagrange multiplier to decouple the joint optimization over $\boldsymbol{Q}$ and $\boldsymbol{W}$, which enables alternating optimization. The new optimization problem

$$\min_{\hat{\boldsymbol{Q}}} \max_{\boldsymbol{W}, \boldsymbol{Q}} \sum_{\mu=1}^n \eta_1(\boldsymbol{x}_\mu^\top \boldsymbol{W}/\sqrt{n}) - n\eta_2(\boldsymbol{Q}) + \hat{\boldsymbol{Q}} \left( \boldsymbol{Q} - \frac{1}{d} \boldsymbol{W}^T \boldsymbol{W} \right)$$

thus becomes separable w.r.t. $\boldsymbol{W}$. We can write an AMP algorithm to optimize it for fixed $\boldsymbol{Q}, \hat{\boldsymbol{Q}}$, and then update $\boldsymbol{Q}, \hat{\boldsymbol{Q}}$ accordingly. This gives AMP-RBM (Algorithm 1). Technically it is a variant of the AMP in Montanari and Venkataramanan [2021] with the Lagrange multiplier $\hat{Q}$ to be the main modification.

In AMP-RBM, we choose the denoisers to be separable and

$$\begin{aligned}
f(\boldsymbol{z}^t, \hat{\boldsymbol{Q}}^t, \boldsymbol{C}_{t-1}) :&= \arg\min_{\boldsymbol{w}} \left\{ \boldsymbol{w}^\top \hat{\boldsymbol{Q}}^t \boldsymbol{w} + \frac{1}{2}(\boldsymbol{C}_{t-1}\boldsymbol{w} + \boldsymbol{z}^t)^\top \boldsymbol{C}_{t-1}^{-1}(\boldsymbol{C}_{t-1}\boldsymbol{w} + \boldsymbol{z}^t) \right\} \\
&= -\left( 2\hat{\boldsymbol{Q}}^t + \boldsymbol{C}_{t-1} \right)^{-1} \boldsymbol{z}^t,
\end{aligned} \tag{16}$$

$$g(\boldsymbol{y}^t, \boldsymbol{B}_t) := \boldsymbol{B}_t^{-1}(\tilde{\boldsymbol{h}}^t - \boldsymbol{y}^t), \; \tilde{\boldsymbol{h}}^t := \arg\min_{\boldsymbol{h}} \left\{ \alpha^{-1} \eta_1(\boldsymbol{h}) + \frac{1}{2}(\boldsymbol{h} - \boldsymbol{y}^t)^\top \boldsymbol{B}_t^{-1}(\boldsymbol{h} - \boldsymbol{y}^t) \right\}. \tag{17}$$

By separability we mean that $[f(\boldsymbol{Z}^t, \hat{\boldsymbol{Q}}^t, \boldsymbol{C}_{t-1})]_i := f(\boldsymbol{z}_i^t, \hat{\boldsymbol{Q}}^t, \boldsymbol{C}_{t-1}) \in \mathbb{R}^k$, and similarly for $g$. Moreover, $\nabla \eta_2 : \mathbb{R}^{k \times k} \to \mathbb{R}^{k \times k}$ is the derivative of $\eta_2 : \mathbb{R}^{k \times k} \to \mathbb{R}$ w.r.t. its argument. The interest of this analysis lies in Theorem 2: the fixed points of AMP-RBM are *exactly* the stationary points of the optimization problem (15), which is proven in Appendix C.3. In Appendix D we show that these match the global optimum for $k = r = 1$ and conjecture that this is true in general.

**Theorem 2.** *Suppose that $\eta_1, \eta_2$ and $g$ are continuously differentiable. Let $\hat{\boldsymbol{W}}$ be the value of $\boldsymbol{W}^t$ at the fixed point of AMP-RBM. Then we have*

$$\nabla_{\hat{\boldsymbol{W}}} \log \tilde{\mathcal{L}}(\hat{\boldsymbol{W}}) = \sum_{\mu=1}^n \frac{1}{\sqrt{n}} \nabla \eta_1(\boldsymbol{x}_\mu^\top \hat{\boldsymbol{W}}/\sqrt{n}) \otimes \boldsymbol{x}_\mu - \alpha \hat{\boldsymbol{W}} \nabla \eta_2(\boldsymbol{Q}(\hat{\boldsymbol{W}})) = 0, \qquad (18)$$

*where $\nabla \eta_2 : \mathbb{R}^{k \times k} \to \mathbb{R}^{k \times k}$ and $\nabla \eta_1 : \mathbb{R}^k \to \mathbb{R}^k$ denote the derivatives of $\eta_1$ and $\eta_2$.*

AMP-RBM follows a family of general AMP iterations described in Appendix C.2. The most attractive property of the AMP family is that there is a low-dimensional, deterministic recursion called the state evolution (SE) that can track its performance:

$$
\begin{aligned}
\bar{\boldsymbol{M}}_t &= \alpha^{-1} \mathbb{E}[f(\boldsymbol{M}_t \boldsymbol{W} + \boldsymbol{\Sigma}_t^{1/2} \boldsymbol{G}, \bar{\hat{\boldsymbol{Q}}}_t, \bar{\boldsymbol{C}}_{t-1}) \boldsymbol{W}^\top] \boldsymbol{\Gamma}, \\
\bar{\boldsymbol{\Sigma}}_t &= \alpha^{-1} \mathbb{E}[f(\boldsymbol{M}_t \boldsymbol{W} + \boldsymbol{\Sigma}_t^{1/2} \boldsymbol{G}, \bar{\hat{\boldsymbol{Q}}}_t, \bar{\boldsymbol{C}}_{t-1}) f(\boldsymbol{M}_t \boldsymbol{W} + \boldsymbol{\Sigma}_t^{1/2} \boldsymbol{G}, \bar{\hat{\boldsymbol{Q}}}_t, \bar{\boldsymbol{C}}_{t-1})^\top], \\
\bar{\boldsymbol{Q}}_t &= \alpha \bar{\boldsymbol{\Sigma}}_t, \quad \bar{\boldsymbol{B}}_t := \alpha^{-1} \mathbb{E}[\nabla f(\boldsymbol{M}_t \boldsymbol{W} + \boldsymbol{\Sigma}_t^{1/2} \boldsymbol{G}, \bar{\hat{\boldsymbol{Q}}}_t, \bar{\boldsymbol{C}}_{t-1}))] \\
\boldsymbol{M}_{t+1} &= \mathbb{E}[g(\bar{\boldsymbol{M}}_t \boldsymbol{U} + \bar{\boldsymbol{\Sigma}}_t \boldsymbol{G}, \bar{\boldsymbol{B}}_t)(\boldsymbol{U})^\top] \boldsymbol{\Gamma}, \\
\boldsymbol{\Sigma}_{t+1} &= \mathbb{E}[g(\bar{\boldsymbol{M}}_t \boldsymbol{U} + \bar{\boldsymbol{\Sigma}}_t \boldsymbol{G}, \bar{\boldsymbol{B}}_t) g(\bar{\boldsymbol{M}}_t \boldsymbol{U} + \bar{\boldsymbol{\Sigma}}_t \boldsymbol{G}, \bar{\boldsymbol{B}}_t)^\top], \\
\bar{\hat{\boldsymbol{Q}}}^{t+1} &= \zeta \bar{\hat{\boldsymbol{Q}}}^t + (1 - \zeta) \nabla \eta_2(\bar{\boldsymbol{Q}}^t), \quad \bar{\boldsymbol{C}}_t := \mathbb{E}[\nabla g(\bar{\boldsymbol{M}}_t \boldsymbol{U} + \bar{\boldsymbol{\Sigma}}_t \boldsymbol{G}, \bar{\boldsymbol{B}}_t)]
\end{aligned}
\qquad (19)
$$

where $\bar{\boldsymbol{M}}_t, \boldsymbol{M}_t, \bar{\boldsymbol{\Sigma}}_t, \boldsymbol{\Sigma}_t, \bar{\boldsymbol{B}}_t, \bar{\boldsymbol{C}}_t, \bar{\boldsymbol{Q}}_t, \bar{\hat{\boldsymbol{Q}}}_t \in \mathbb{R}^{k \times k}$, and $\boldsymbol{\Gamma} := \sqrt{\alpha} \boldsymbol{\Lambda} \in \mathbb{R}^{k \times k}$ is the effective signal-noise ratio. $\bar{\boldsymbol{M}}_1$ is determined by the initialization. The expectation is w.r.t. independent random variables $\boldsymbol{U}, \boldsymbol{W}, \boldsymbol{G} \in \mathbb{R}^k$ sampled from $\boldsymbol{U}_{[1:r]} \sim P_u$, $\boldsymbol{W}_{[1:r]} \sim P_w$, $[G_i]_{i=1}^k \overset{iid}{\sim} \mathcal{N}(0, 1)$, and $U_a = W_a = 0$ for $a > r$, where $\boldsymbol{U}_{[1:r]}, \boldsymbol{W}_{[1:r]}$ refer to the first $r$ elements of $\boldsymbol{U}, \boldsymbol{W}$. $\nabla f : \mathbb{R}^k \times \mathbb{R}^{k \times k} \times \mathbb{R}^{k \times k} \to \mathbb{R}^{k \times k}$ refers to the derivatives of $f$ w.r.t its first argument, and similarly for $g$.

We have the following theorem formally describing how AMP-RBM is tracked by its SE. It is a special case of the SE of a general AMP iteration proven in Appendix C.2.

**Theorem 3.** *Suppose that $f$, $g$ and $\nabla \eta_2$ are Lipschitz continuous, uniformly at $\{\bar{\boldsymbol{B}}_t, \bar{\boldsymbol{C}}_t, \bar{\hat{\boldsymbol{Q}}}_t\}$[1], and further assume that the initialization satisfies $\{\boldsymbol{z}_1^1, \cdots, \boldsymbol{z}_d^1\} \overset{iid}{\sim} \boldsymbol{M}_1 \boldsymbol{W} + \boldsymbol{\Sigma}_0^{1/2} \boldsymbol{G}$. Then for any $PL(2)$ function[2] $\psi$, the following[3] holds almost surely for $t \geq 1$:*

$$\lim_{d \to \infty} \frac{1}{d} \sum_{i=1}^d \psi(\boldsymbol{w}_i^*, \boldsymbol{w}_i^t) = \mathbb{E}[\psi(\boldsymbol{W}, f(\boldsymbol{M}_t \boldsymbol{W} + \boldsymbol{\Sigma}_t^{1/2} \boldsymbol{G}, \bar{\hat{\boldsymbol{Q}}}_t, \bar{\boldsymbol{C}}_{t-1}))]. \qquad (20)$$

It is quite straightforward to extend Theorem 3 to the spectral initialization as in [Montanari and Venkataramanan, 2021, Theorem 5]. We emphasize that AMP-RBM is indeed a practical algorithm and it empirically works well for both the spectral initialization and the random initialization. For spectral initialization, the SE (Theorem 3) still holds. However, for random initialization, the correlation between the signal and the estimation might keep zero until $\Theta(\log d)$ iterations [Li et al., 2023], while the SE only holds for $\Theta(1)$ iterations.

An important performance criterion concerning the spiked covariance model is the SNR at which the algorithm begins to obtain a non-zero correlation with the signal, which is referred to as the weak recovery threshold [Troiani et al., 2024]. For the weak recovery threshold, we can consider

---

[1] $f(x, c)$ is uniformly Lipschitz at $c_0$ if there is an open neighborhood $U$ of $c_0$ and a constant $L$ such that $f(\cdot, c)$ is $L-$Lipschitz for any $c \in U$ and $|f(x, c_1) - f(x, c_2)| \leq L(1 + ||x||)|c_1 - c_2|$ for any $c_1, c_2 \in U$.

[2] i.e.,pseudo-Lipschitz function of order 2, satisfying $|\psi(\boldsymbol{x}_1) - \psi(\boldsymbol{x}_2)| \leq C||\boldsymbol{x}_1 - \boldsymbol{x}_2||(1 + ||\boldsymbol{x}_1|| + ||\boldsymbol{x}_2||)$ for some constant $C \geq 0$.

[3] A similar result also holds for $\boldsymbol{U}$. See Theorem 6.

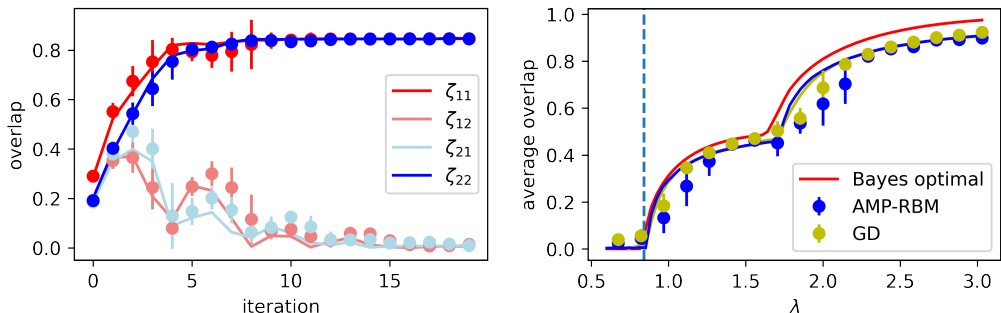

**Figure 1: Left**: Iteration curves of AMP-RBM, $r = k = 2, \Lambda = 1.4I_2$, so the overlap is a $2 \times 2$ matrix containing $\zeta_{11}, \zeta_{12}, \zeta_{21}, \zeta_{22}$. Lines denote the state evolution. **Right**: The performance of AMP-RBM, GD (over (15)) and the Bayes optimality Lesieur et al. [2017] for $r = k = 2$ and $\Lambda = \text{diag}(\lambda, 0.5\lambda)$, where AMP-RBM and GD use random initialization. The dashed blue line represents the BBP transition. The purple and yellow lines represent the SE of AMP-RBM and GD, which almost overlap. We use the Rademacher prior and $n = 8000, d = 4000$.

the linearization of AMP-RBM around 0. For $\boldsymbol{x}$ close to 0, we have $\alpha^{-1}\eta_1(\boldsymbol{x}) \approx \frac{1}{2}\boldsymbol{x}^\top\boldsymbol{x}$, and thus $g(\boldsymbol{y}^t, \boldsymbol{B}_t) \approx -(\boldsymbol{B}_t + \boldsymbol{I}_k)^{-1}\boldsymbol{y}^t$.

Moreover, around $\boldsymbol{W} = 0$ we have $\hat{\boldsymbol{Q}} \approx \frac{1}{2}\boldsymbol{I}_k$, and thus $f(\boldsymbol{z}^t, \hat{\boldsymbol{Q}}, \boldsymbol{C}_{t-1}) \approx -(\boldsymbol{I}_k + \boldsymbol{C}_{t-1})^{-1}\boldsymbol{z}^t$. In this case, AMP-RBM becomes equivalent to the power method, so its weak recovery threshold is exactly the celebrated BBP threshold [Baik et al., 2005], below which it is information theoretically impossible to distinguish the signals from the noise [Deshpande and Montanari, 2014]. Thus AMP-RBM as well as RBMs (because Theorem 2 suggests that their fixed points match) are in this sense optimal in detecting the signals. To see this, we can assume $\boldsymbol{M}_t \ll \boldsymbol{\Sigma}_t \ll 1$, and expand the state evolution as

$$\bar{\boldsymbol{M}}_t = -\alpha^{-1}(\bar{\boldsymbol{C}}_{t-1} + \boldsymbol{I}_k)^{-1}\boldsymbol{M}_{t-1}\boldsymbol{\Gamma}, \ \boldsymbol{M}_t = -(\bar{\boldsymbol{B}}_t + \boldsymbol{I}_k)^{-1}\bar{\boldsymbol{M}}_t\boldsymbol{\Gamma},$$
$$\bar{\boldsymbol{\Sigma}}_t = \alpha^{-1}(\bar{\boldsymbol{C}}_{t-1} + \boldsymbol{I}_k)^{-1}\boldsymbol{\Sigma}_t(\bar{\boldsymbol{C}}_{t-1} + \boldsymbol{I}_k)^{-1}, \ \boldsymbol{\Sigma}_{t+1} = (\bar{\boldsymbol{B}}_t + \boldsymbol{I}_k)^{-1}\bar{\boldsymbol{\Sigma}}_t(\bar{\boldsymbol{B}}_t + \boldsymbol{I}_k)^{-1}, \quad (21)$$

which gives $\boldsymbol{M}_{t+1}^\top\boldsymbol{\Sigma}_{t+1}^{-1}\boldsymbol{M}_{t+1} = \alpha^{-1}\boldsymbol{\Gamma}^2(\boldsymbol{M}_t^\top\boldsymbol{\Sigma}_t^{-1}\boldsymbol{M}_t)\boldsymbol{\Gamma}^2$. Therefore, 0 is a stable fixed point of AMP-RBM if and only of

$$\alpha\lambda_{\max}^4 \le 1. \quad (22)$$

The transition point is just standard BBP transition $\lambda_{\max} = \alpha^{-1/4}$. Note that all the above analysis does not assume $k = r$, so the weak recovery threshold remains the same when the number of hidden units does not match the number of signals, which is common in practice. This is in sharp contrast with the results in Thériault et al. [2024], Manzan and Tantari [2024], which essentially use a different loss function (in the zero temperature limit) and suggest the system is in the spin glass phase.

Our numerics are presented in Figure 1, where we measure the overlap between $\boldsymbol{W}^*$ and the AMP-RBM algorithm output. The left side of Figure 1 verifies Theorem 3 and the right side of Figure 1 verifies that the weak recovery threshold of AMP-RBM matches the BBP transition. See Appendix F.1 for more experiments, including degenerate eigenvalues and more comparisons with other algorithms.

**Asymptotics of the likelihood extremizers —** AMP-RBM can also be utilized as a proof technique to obtain an asymptotic description of the stationary points of (15) under a further assumption.

**Assumption 2.** *AMP-RBM converges to its fixed point (i.e., the stationary point of* (15)*) uniformly, that is, there exists* $\hat{\boldsymbol{W}}(d) = \lim_{t\to\infty} \boldsymbol{W}^t(d)$ *such that*

$$\lim_{t\to\infty}\lim_{d\to\infty}\frac{1}{d}||\boldsymbol{W}^t(d) - \hat{\boldsymbol{W}}(d)||_F^2 = 0, \quad (23)$$

*and this holds for any output of Algorithm 1.*

While Assumption 2 might seem artificial, in Appendix D we will show that actually it comes from a more essential condition (Assumption 6) called the replicon stability condition or de Almeida-Thouless condition [de Almeida and Thouless, 1978], which characterizes the complexity of landscapes, i.e.,the landscape is well behaved if the replicon stability condition holds. Assumption 2 (the convergence of AMP-RBM) is also verified by the numerics in Figure 1.

Now in Theorem 4, we characterize the asymptotic property of the stationary point $\hat{\boldsymbol{W}}$.

**Theorem 4.** *Under Assumptions 1-2 and assuming that $g$ is uniformly Lipschitz at $\bar{\boldsymbol{B}}_t$, we have* $\lim_{d\to\infty} ||\sqrt{d}\nabla \log \mathcal{L}(\hat{\boldsymbol{W}}(d))|| = 0$, *where the asymptotic stationary point $\hat{\boldsymbol{W}}(d)$ satisfies*

$$\lim_{d\to\infty} \frac{1}{d} \sum_{i=1}^{d} \psi(\boldsymbol{w}_i^*, \hat{\boldsymbol{w}}_i) = \mathbb{E}[\psi(\boldsymbol{W}, f(\boldsymbol{M}_\infty \boldsymbol{W} + \Sigma_\infty^{1/2}\boldsymbol{G}, \bar{\hat{\boldsymbol{Q}}}_\infty, \bar{\boldsymbol{C}}_\infty)]. \tag{24}$$

*Here $\bar{\boldsymbol{C}}_\infty = \lim_{t\to\infty} \bar{\boldsymbol{C}}_t$, $\boldsymbol{M}_\infty := \lim_{t\to\infty} \boldsymbol{M}_t$, $\Sigma_\infty := \lim_{t\to\infty} \Sigma_t$ and $\bar{\hat{\boldsymbol{Q}}}_\infty := \lim_{t\to\infty} \bar{\hat{\boldsymbol{Q}}}_t$ are well-defined matrices.*

Specifically, by choosing $\psi(\boldsymbol{w}_i^*, \hat{\boldsymbol{w}}_i) := (\boldsymbol{w}_i^*)^\top \hat{\boldsymbol{w}}_i$, we can measure how much the stationary points $\hat{\boldsymbol{W}}$ overlap with the features $\boldsymbol{W}^*$. Theorem 4 is proved in Appendix C.4 by combining Theorem 2 and Theorem 3. In Appendix D, we will prove that for the special case $r = k = 1$, Theorem 4 also describes the global optimum. We conjecture that this is true in general, but our proof technique, that relies on the Gordon mini-max theorem [Gordon, 1988, Thrampoulidis et al., 2015, Vilucchio et al., 2025], is notably difficult to adapt to larger values of $k$. An interesting corollary is that there exists a stationary point where different units align with different signals, even when the number of hidden units is different from the number of signals.

**Corollary 1.** *Under the conditions in Theorem 4, and further assuming that $P_u, P_w, P_h$ are factorizable and symmetric, there exists an asymptotic stationary point with all matrices in Theorem 4 diagonal.*

Corollary 1 is proved in Appendix C.5. It suggests that at the stationary points it specified, the weights of different hidden units are orthogonal. For the overparameterized scheme, $k$ units of the RBM are aligned with $k$ independent signals, and the other $r - k$ weight vectors are pure noise. For the underparameterized scheme, $k$ units of the RBM are still aligned with $k$ independent signals, randomly chosen out of $r$ independent signals. This outperforms SVD, which cannot distinguish different signals. See Appendix F.2 for the training dynamics of RBMs concerning this point.

However, algorithms might not converge to the stationary points specified by Corollary 1. It is the interaction between the signal prior and $\eta_1$ that determines how much the hidden units align with the signals. Inspired by the global optimality of rank-1 case proved in Appendix D, we conjecture that the stationary points specified by Corollary 1 are actually global minima, which is reachable by informed initialization of AMP-RBM, but uninformed algorithms might not necessarily find it. Numerical evidence can be found in Appendix F.1.

**Asymptotic analysis of the gradient descent dynamics —** Finally, we provide the asymptotics of GD for (15), sometimes referred to as DMFT (dynamic mean-field theory) [Maimbourg et al., 2016, Roy et al., 2019, Mignacco et al., 2020, Mignacco and Urbani, 2022, Gerbelot et al., 2024]. While contrastive divergence is preferred to GD in practice, we believe that they have qualitatively similar performances —contrastive divergence is essentially an approximation of GD — which is also validated in Appendix E on real datasets. GD over (5) reads

$$\tilde{\boldsymbol{W}}^{t+1} = \tilde{\boldsymbol{W}}^t + \kappa \left( \sum_{\mu=1}^{n} \frac{1}{\sqrt{n}} \nabla \eta_1(\boldsymbol{x}_\mu^\top \tilde{\boldsymbol{W}}^t/\sqrt{n}) \otimes \boldsymbol{x}_\mu - \alpha \tilde{\boldsymbol{W}} \nabla \eta_2(\boldsymbol{Q}(\tilde{\boldsymbol{W}}^t)) \right), \tag{25}$$

with $\kappa$ the learning rate. Its asymptotics can be described by the following low dimensional iteration.

A sequence of $k$-dimensional random variables $\{\mathring{\boldsymbol{U}}_t, \mathring{\boldsymbol{Y}}_t, \mathring{\boldsymbol{W}}_t, \mathring{\boldsymbol{Z}}_t\}_{t\geq 1}$ are defined as

$$\begin{aligned} \mathring{\boldsymbol{U}}_t &= \mathring{f}_t(\mathring{\boldsymbol{Y}}_t, \mathring{\boldsymbol{Y}}_{t-1}, \cdots, \mathring{\boldsymbol{Y}}_1), \\ \mathring{\boldsymbol{W}}_{t+1} &= \mathring{g}_{t+1}(\mathring{\boldsymbol{Z}}_t, \mathring{\boldsymbol{Z}}_{t-1}, \cdots, \mathring{\boldsymbol{Z}}_1), \end{aligned} \tag{26}$$

where

$$\begin{aligned} (\mathring{\boldsymbol{Y}}_1, \cdots, \mathring{\boldsymbol{Y}}_t) &= (\mathring{\boldsymbol{M}}_1, \cdots, \mathring{\boldsymbol{M}}_t)\boldsymbol{U} + \mathcal{N}(0, \mathring{\Sigma}_t), \\ (\mathring{\boldsymbol{Z}}_1, \cdots, \mathring{\boldsymbol{Z}}_t) &= (\mathring{\boldsymbol{N}}_1, \cdots, \mathring{\boldsymbol{N}}_t)\boldsymbol{W} + \mathcal{N}(0, \mathring{\Omega}_t), \end{aligned} \tag{27}$$

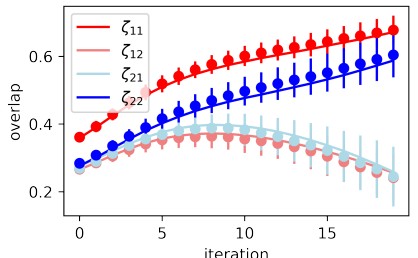

**Figure 2:** Iteration curves of GD, where the lines denote its asymptotics. The same setting as Figure 1.

and

$$\mathring{\boldsymbol{M}}_t = \alpha^{-1} \mathbb{E}[\mathring{\mathring{\boldsymbol{W}}}_t^\top \boldsymbol{W}] \boldsymbol{\Gamma}, \; [\mathring{\boldsymbol{\Sigma}}_t]_{ij} = \alpha^{-1} \mathbb{E}[\mathring{\mathring{\boldsymbol{W}}}_i^\top \boldsymbol{W}_j],$$
$$\mathring{\boldsymbol{N}}_{t+1} = \mathbb{E}[\mathring{\boldsymbol{U}}_{t+1}^\top \boldsymbol{U}] \boldsymbol{\Gamma}, \; [\mathring{\boldsymbol{\Omega}}_{t+1}]_{ij} = \mathbb{E}[\mathring{\boldsymbol{U}}_i^\top \mathring{\boldsymbol{U}}_j]. \tag{28}$$

$\mathring{\boldsymbol{M}}_1$ and $[\mathring{\boldsymbol{\Sigma}}_t]_{11}$ are given by initialization $\tilde{\boldsymbol{W}}^0 \sim \mathring{\boldsymbol{W}}_0$. We note that $\mathring{\boldsymbol{M}}_t, \mathring{\boldsymbol{N}}_t \in \mathbb{R}^{k \times k}$ and $\mathring{\boldsymbol{\Sigma}}_t, \mathring{\boldsymbol{\Omega}}_t \in \mathbb{R}^{k \times k \times t \times t}$. The functions $\mathring{f}_t, \mathring{g}_{t+1} : \mathbb{R}^{k \times t} \to \mathbb{R}^k$ are rather involved and are defined in an iterative way, so we put their definitions in Appendix C.6.

The following theorem gives the asymptotics of GD, which is proven in Appendix C.6.

**Theorem 5.** *Assume that $\nabla \eta_1, \nabla \eta_2$ are Lipschitz continuous. Then, for any $PL(2)$ function $\psi$, the following holds almost surely for $t \geq 0$:*

$$\lim_{d \to \infty} \frac{1}{d} \sum_{i=1}^d \psi(\boldsymbol{w}_i^*, \tilde{\boldsymbol{w}}_i^t) = \mathbb{E}[\psi(\boldsymbol{W}, \mathring{\boldsymbol{W}}_t)]. \tag{29}$$

*Moreover, if we further assume that GD converges to $\hat{\boldsymbol{W}}$ uniformly as in Assumption 2, we have*

$$\lim_{d \to \infty} \frac{1}{d} \sum_{i=1}^d \psi(\boldsymbol{w}_i^*, \hat{\boldsymbol{w}}_i) = \lim_{t \to \infty} \mathbb{E}[\psi(\boldsymbol{W}, \mathring{\boldsymbol{W}}_t)] \tag{30}$$

*almost surely, where the limits on both sides are well defined.*

Theorem 5 is illustrated in Figure 2. Similar to the linearization of AMP-RBM, linearization of GD yields the same weak recovery threshold because both of them reduces to the power methods. This is demonstrated empirically in Figure 1. Figure 1 also suggests that GD converges to a similar point as AMP-RBM. Some preliminary analysis of GD dynamics is presents in Appendix F.2.

## Acknowledgement

We would like to thank CECAM at EPFL for organising a workshop where Gianluca Manzan presented his work Manzan and Tantari [2024] that inspired our work. We would also like to thank Remi Monasson for insightful discussions. We acknowledge funding from the Swiss National Science Foundation grants SNSF SMArtNet (grant number 212049), OperaGOST (grant number 200021 200390) and DSGIANGO (grant number 225837).

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

## A Additional literature review on the teacher-student setting

In many physics papers [Huang and Toyoizumi, 2016, Huang, 2017, Thériault et al., 2024, Manzan and Tantari, 2024], the weights $\boldsymbol{W}$ are sampled from the posterior of the student RBM

$$\mathbb{P}(\boldsymbol{W}|\boldsymbol{X}) = \frac{1}{\mathcal{Z}(\boldsymbol{X})} \Pi_{\mu=1}^{n} P_v(\boldsymbol{x}_\mu) \mathbb{E}_{\boldsymbol{h}} e^{\frac{\beta}{\sqrt{d}} \boldsymbol{x}_\mu^\top \boldsymbol{W} \boldsymbol{h}}, \tag{31}$$

where $\beta$ represents the inverse temperature of the student RBM and $\mathcal{Z}(\boldsymbol{X})$ is the normalization factor. Their settings differ from the maximal likelihood training scheme used in practice. When $\beta = \beta^*$ (see (14)), it is the Bayesian estimation, equivalent to the Bayesian optimal performance presented in Section 5. For example, Thériault et al. [2024] points out that Bayes optimal students can learn teachers in a one-to-one pattern for high SNR ($\beta = \beta^*$ large) but not for small SNR, which they call the permutation symmetry breaking phenomenon. This is consistent with our numerics in Appendix F.1. The paramagnetic-to-ferromagnetic transition in Manzan and Tantari [2024] also reproduces the BBP transition.

While they also analyze the teacher-student RBM model with $\beta \neq \beta^*$, we need to emphasize $\beta \to \infty$ is not equivalent to maximizing the empirical likelihood $p(\boldsymbol{x}_\mu|\boldsymbol{W}^*)$ in this paper and in practice. Actually, by (31) $\beta \to \infty$ maximizes the term in the exponent $\sum_{\mu=1}^{n} \boldsymbol{x}_\mu^\top \boldsymbol{W} \boldsymbol{h}$ instead. As a consequence, Thériault et al. [2024], Manzan and Tantari [2024] predict that at zero temperature ($\beta \to \infty$), the system is in the spin glass phase and it is statistically intractable to infer the planted signal ($\boldsymbol{W}^*$), while we show that the weak recovery threshold of the student RBM trained with likelihood maximization always matches the BBP transition (22).

# B Simplification of RBMs' log-likelihood

## B.1 Proof of Theorem 1

*Proof.* For the following, we will prove a related result

$$\lim_{d\to\infty} \sqrt{d}||\nabla_{\boldsymbol{W},\boldsymbol{\theta},\sqrt{d}\boldsymbol{b}} \log \mathcal{L}(\boldsymbol{W},\boldsymbol{\theta},\boldsymbol{b}) - \nabla_{\boldsymbol{W},\boldsymbol{\theta},\sqrt{d}\boldsymbol{b}} \log \tilde{\mathcal{L}}(\boldsymbol{W},\boldsymbol{\theta},\boldsymbol{b})||_F = 0, \tag{32}$$

which is useful to find the stationary points of $\log \mathcal{L}(\boldsymbol{W},\boldsymbol{\theta},\boldsymbol{b})$ (e.g.,for Theorem 4). The $\frac{1}{\sqrt{d}}$ scaling is appropriate because $||\nabla_{\boldsymbol{W},\boldsymbol{\theta},\sqrt{d}\boldsymbol{b}} \log \tilde{\mathcal{L}}(\boldsymbol{W},\boldsymbol{\theta},\boldsymbol{b})||_F = \Theta(\sqrt{d})$.

According to (3), we have

$$\log \mathcal{L}(\boldsymbol{W},\boldsymbol{\theta},\boldsymbol{b}) = \log \Pi_{\mu=1}^n \int \mathrm{d}P_h(\boldsymbol{h}_\mu) e^{\frac{1}{\sqrt{d}}\boldsymbol{x}_\mu^\top \boldsymbol{W}\boldsymbol{h}_\mu + \frac{1}{\sqrt{d}}\boldsymbol{x}_\mu^\top \boldsymbol{\theta} + \boldsymbol{b}^\top \boldsymbol{h}_\mu}$$

$$- n\log \int \Pi_{i=1}^d \mathrm{d}P_v(v_i)\mathrm{d}P_h(\boldsymbol{h}) e^{\frac{1}{\sqrt{d}}\boldsymbol{v}^\top \boldsymbol{W}\boldsymbol{h} + \frac{1}{\sqrt{d}}\boldsymbol{v}^\top \boldsymbol{\theta} + \boldsymbol{b}^\top \boldsymbol{h}}. \tag{33}$$

The first term is exactly $\eta_1$. We then take derivative of the log-likelihood. We can expand the derivative of the second term of (33) using

$$\frac{\partial}{\partial w_{ai}} \int \mathrm{d}P_v(v_i) e^{\frac{1}{\sqrt{d}}v_i\boldsymbol{h}^\top \boldsymbol{w}_i + \frac{1}{\sqrt{d}}v_i\theta_i}$$

$$= \int \mathrm{d}P_v(v_i) \frac{1}{\sqrt{d}} v_i h_a e^{\frac{1}{\sqrt{d}}v_i\boldsymbol{h}^\top \boldsymbol{w}_i + \frac{1}{\sqrt{d}}v_i\theta_i}$$

$$= \int \mathrm{d}P_v(v_i) \left( \frac{1}{\sqrt{d}} v_i h_a + \frac{1}{d} v_i^2 h_a (\boldsymbol{h}^\top \boldsymbol{w}_i + \theta_i) \right) + O\left( \frac{1}{d^{3/2}} (\boldsymbol{h}^\top \boldsymbol{w}_i + \theta_i)^2 \right) \tag{34}$$

$$= \frac{1}{d} h_a (\boldsymbol{h}^\top \boldsymbol{w}_i + \theta_i) + O\left( \frac{1}{d^{3/2}} (\boldsymbol{h}^\top \boldsymbol{w}_i + \theta_i)^2 \right)$$

$$= \frac{\partial}{\partial w_{ai}} e^{\frac{1}{2d}(\boldsymbol{h}^\top \boldsymbol{w}_i + \theta_i)^2} + O\left( \frac{1}{d^{3/2}} (\theta_i^2 + ||\boldsymbol{w}_i||^2) \right).$$

For the second line we exchange the derivative and the integral because the integrand is over a bounded domain (Assumption 1) and is continuously differentiable. For the third line we use the Taylor's expansion

$$e^{(v_i\boldsymbol{h}^T\boldsymbol{w}_i + v_i\theta_i)/\sqrt{d}} = 1 + (v_i\boldsymbol{h}^T\boldsymbol{w}_i + v_i\theta_i)/\sqrt{d} + O((v_i\boldsymbol{h}^T\boldsymbol{w}_i + v_i\theta_i)^2/d)$$

and that $v_i$ is bounded according to Assumption 1. For the fourth line we use $\int \mathrm{d}P_v(v_i)v_i = 0$ and $\int \mathrm{d}P_v(v_i)v_i^2 = 1$. For the last line we use the Taylor's expansion

$$e^{(\boldsymbol{h}^T w_i + \theta_i)^2/2d} = 1 + (\boldsymbol{h}^T w_i + \theta_i)^2/2d + O((\boldsymbol{h}^T w_i + \theta_i)^4/d^2)$$

and that $\boldsymbol{h}$ is bounded according to Assumption 1. Thus we have

$$\sum_{a,i=1}^{k,d} \left( \frac{\partial}{\partial w_{ai}} \int \mathrm{d}P_h(\boldsymbol{h})\mathrm{d}P_v(\boldsymbol{v}) e^{\frac{1}{\sqrt{d}}\boldsymbol{v}^\top \boldsymbol{W}\boldsymbol{h} + \frac{1}{\sqrt{d}}\boldsymbol{v}^\top \boldsymbol{\theta} + \boldsymbol{h}^\top \boldsymbol{b}} \right.$$

$$\left. - \frac{\partial}{\partial w_{ai}} \int \mathrm{d}P_h(\boldsymbol{h}) e^{\frac{1}{2d}\sum_{i=1}^d (\boldsymbol{h}^\top \boldsymbol{w}_i + \theta_i)^2 + \boldsymbol{h}^\top \boldsymbol{b}} \right)^2 = o(1), \tag{35}$$

where we use the fact that $P_v$ is factorizable (Assumption 1) and the boundedness assumption in Theorem 1. Moreover, we have

$$
\begin{aligned}
Z(\boldsymbol{W}, \boldsymbol{\theta}, \boldsymbol{b}) &= \int \mathrm{d}P_h(\boldsymbol{h})\mathrm{d}P_v(\boldsymbol{v})e^{\frac{1}{\sqrt{d}}\boldsymbol{v}^\top \boldsymbol{W}\boldsymbol{h} + \frac{1}{\sqrt{d}}\boldsymbol{v}^\top \boldsymbol{\theta} + \boldsymbol{h}^\top \boldsymbol{b}} \\
&= \int \mathrm{d}P_h(\boldsymbol{h})e^{\boldsymbol{h}^\top \boldsymbol{b}} \int \Pi_{i=1}^d \mathrm{d}P_v(v_i)e^{\frac{1}{\sqrt{d}}v_i(\boldsymbol{h}^\top \boldsymbol{w}_i + \theta_i)} \\
&= \int \mathrm{d}P_h(\boldsymbol{h})e^{\boldsymbol{h}^\top \boldsymbol{b}} \int \Pi_{i=1}^d \mathrm{d}P_v(v_i)\left(1 + \frac{1}{\sqrt{d}}v_i(\boldsymbol{h}^\top \boldsymbol{w}_i + \theta_i) + \frac{1}{2d}v_i^2(\boldsymbol{h}^\top \boldsymbol{w}_i + \theta_i)^2\right) \\
&\quad + O\left(\frac{1}{d^{3/2}}(||\boldsymbol{\theta}||^2 + ||\boldsymbol{W}||^2)\right) \\
&= \int \mathrm{d}P_h(\boldsymbol{h})e^{\frac{1}{2d}\sum_{i=1}^d (\boldsymbol{h}^\top \boldsymbol{w}_i + \theta_i)^2 + \boldsymbol{h}^\top \boldsymbol{b}} + o(1),
\end{aligned}
$$
(36)

where for the last line, we integrate over $v_i$, and use $1 + (h^T w_i + \theta_i)^2/2d \approx e^{(h^T w_i + \theta_i)^2/2d}$. The higher order terms can be bounded under the bounded assumptions (Assumption 1).

(36) directly proves (4), because we can notice that the first term of (33) remains the same, and the second term is $n \log Z(\boldsymbol{W}, \boldsymbol{\theta}, \boldsymbol{b})$, which is equal to $n(\eta_2 + o(1))$ by (36).

(36) also gives

$$
\begin{aligned}
&d\left\|\nabla_{\boldsymbol{W}} \log \int \mathrm{d}P_h(\boldsymbol{h})\mathrm{d}P_v(\boldsymbol{v})e^{\frac{1}{\sqrt{d}}\boldsymbol{v}^\top \boldsymbol{W}\boldsymbol{h} + \frac{1}{\sqrt{d}}\boldsymbol{v}^\top \boldsymbol{\theta} + \boldsymbol{h}^\top \boldsymbol{b}} - \nabla_{\boldsymbol{W}}\eta_2\left(\frac{1}{d}\boldsymbol{W}^\top \boldsymbol{W}, \frac{1}{d}\boldsymbol{W}^\top \boldsymbol{\theta}, \frac{1}{d}\boldsymbol{\theta}^\top \boldsymbol{\theta}\right)\right\|^2 \\
&= d\sum_{a,i=1}^{k,d}\left(\frac{\frac{\partial}{\partial w_{ai}}\int \mathrm{d}P_v(v_i)e^{\frac{1}{\sqrt{d}}v_i\boldsymbol{h}^\top \boldsymbol{w}_i + \frac{1}{\sqrt{d}}v_i\theta_i}}{Z(\boldsymbol{W}, \boldsymbol{\theta}, \boldsymbol{b})} - \frac{\partial}{\partial w_{ai}}\eta_2\left(\frac{1}{d}\boldsymbol{W}^\top \boldsymbol{W}, \frac{1}{d}\boldsymbol{W}^\top \boldsymbol{\theta}, \frac{1}{d}\boldsymbol{\theta}^\top \boldsymbol{\theta}\right)\right)^2 \\
&= d\sum_{a,i=1}^{k,d}\left(\frac{\frac{\partial}{\partial w_{ai}}\int \mathrm{d}P_v(v_i)e^{\frac{1}{\sqrt{d}}v_i\boldsymbol{h}^\top \boldsymbol{w}_i + \frac{1}{\sqrt{d}}v_i\theta_i}}{Z(\boldsymbol{W}, \boldsymbol{\theta}, \boldsymbol{b})} - \frac{\frac{\partial}{\partial w_{ai}}\int \mathrm{d}P_h(\boldsymbol{h})e^{\frac{1}{2d}(\boldsymbol{h}^\top \boldsymbol{w}_i + \theta_i)^2}}{\int \mathrm{d}P_h(\boldsymbol{h})e^{\frac{1}{2d}\sum_{i=1}^d (\boldsymbol{h}^\top \boldsymbol{w}_i + \theta_i)^2 + \boldsymbol{h}^\top \boldsymbol{b}}}\right)^2 \\
&= o(1),
\end{aligned}
$$
(37)

where we use (35) and (36). Similarly, we have

$$
\begin{aligned}
&\frac{\partial}{\partial \theta_i}\int \mathrm{d}P_v(v_i)e^{\frac{1}{\sqrt{d}}v_i\boldsymbol{h}^\top \boldsymbol{w}_i + \frac{1}{\sqrt{d}}v_i\theta_i} \\
&= \int \mathrm{d}P_v(v_i)\left(\frac{1}{\sqrt{d}}v_i + \frac{1}{d}v_i^2(\boldsymbol{h}^\top \boldsymbol{w}_i + \theta_i)\right) + O\left(\frac{1}{d^{3/2}}(\theta_i^2 + ||\boldsymbol{w}_i||^2)\right) \\
&= \frac{\partial}{\partial \theta_i}e^{\frac{1}{2d}(\boldsymbol{h}^\top \boldsymbol{w}_i + \theta_i)^2} + O\left(\frac{1}{d^{3/2}}(\theta_i^2 + ||\boldsymbol{w}_i||^2)\right),
\end{aligned}
$$
(38)

where analogous to (34) we use Taylor's expansions and integrate over $v_i$. Analogously to (37) we obtain

$$
\lim_{d\to\infty}\left\|\nabla_{\boldsymbol{\theta}} \log \int \mathrm{d}P_h(\boldsymbol{h})\mathrm{d}P_v(\boldsymbol{v})e^{\frac{1}{\sqrt{d}}\boldsymbol{v}^\top \boldsymbol{W}\boldsymbol{h} + \frac{1}{\sqrt{d}}\boldsymbol{v}^\top \boldsymbol{\theta} + \boldsymbol{h}^\top \boldsymbol{b}} \right. \\
\left. - \nabla_{\boldsymbol{\theta}}\eta_2\left(\frac{1}{d}\boldsymbol{W}^\top \boldsymbol{W}, \frac{1}{d}\boldsymbol{W}^\top \boldsymbol{\theta}, \frac{1}{d}\boldsymbol{\theta}^\top \boldsymbol{\theta}\right)\right\| = 0.
$$
(39)

We also have

$$
\begin{aligned}
&\frac{\partial}{\partial b_a}\int \mathrm{d}P_h(\boldsymbol{h})\mathrm{d}P_v(\boldsymbol{v})e^{\frac{1}{\sqrt{d}}\boldsymbol{v}^\top \boldsymbol{W}\boldsymbol{h} + \frac{1}{\sqrt{d}}\boldsymbol{v}^\top \boldsymbol{\theta} + \boldsymbol{h}^\top \boldsymbol{b}} \\
&= \frac{\partial}{\partial b_a}\int \mathrm{d}P_h(\boldsymbol{h})\mathrm{d}P_v(\boldsymbol{v})e^{\frac{1}{2d}\sum_{i=1}^d (\boldsymbol{h}^\top \boldsymbol{w}_i + \theta_i)^2 + \boldsymbol{h}^\top \boldsymbol{b}} + o(1),
\end{aligned}
$$
(40)

and thus
$$\lim_{d\to\infty}\left\|\nabla_{\boldsymbol{b}}\log\int\mathrm{d}P_h(\boldsymbol{h})\mathrm{d}P_v(\boldsymbol{v})e^{\frac{1}{\sqrt{d}}\boldsymbol{v}^\top\boldsymbol{W}\boldsymbol{h}+\frac{1}{\sqrt{d}}\boldsymbol{v}^\top\boldsymbol{\theta}+\boldsymbol{h}^\top\boldsymbol{b}}\right.$$
$$\left.-\nabla_{\boldsymbol{b}}\eta_2\left(\frac{1}{d}\boldsymbol{W}^\top\boldsymbol{W},\frac{1}{d}\boldsymbol{W}^\top\boldsymbol{\theta},\frac{1}{d}\boldsymbol{\theta}^\top\boldsymbol{\theta}\right)\right\|=0. \tag{41}$$

We prove (32) by combining (37), (39) and (41). $\qquad\square$

## B.2 Visible units with nonzero mean

When $P_v$ has a nonzero mean $\bar{v}$, (36) gives

$$\mathcal{Z}(\boldsymbol{W})=\int\mathrm{d}P_h(\boldsymbol{h})e^{\frac{1}{\sqrt{d}}\sum_{i=1}^d\bar{v}(\boldsymbol{h}^\top\boldsymbol{w}_i+\theta_i)+\frac{1}{2d}\sum_{i=1}^d(\boldsymbol{h}^\top\boldsymbol{w}_i+\theta_i)^2+\boldsymbol{h}^\top\boldsymbol{b}}+o(1). \tag{42}$$

The gradients can be dealt with similarly to (34) and (35), which leads to the following corollary.

**Corollary 2.** *Under the conditions in Theorem 1, but when $P_v$ has mean $\bar{v}$, we have*

$$\lim_{d\to\infty}\sqrt{d}\|\nabla_{\boldsymbol{W},\boldsymbol{\theta},\sqrt{d}\boldsymbol{b}}\log\mathcal{L}(\boldsymbol{W},\boldsymbol{\theta},\boldsymbol{b})-\nabla_{\boldsymbol{W},\boldsymbol{\theta},\sqrt{d}\boldsymbol{b}}\log\tilde{\mathcal{L}}(\boldsymbol{W},\boldsymbol{\theta},\boldsymbol{b})\|_F=0, \tag{43}$$

*where*

$$\log\tilde{\mathcal{L}}(\boldsymbol{W},\boldsymbol{\theta},\boldsymbol{b}):=\sum_{\mu=1}^n\eta_1\left(\frac{1}{\sqrt{n}}\boldsymbol{x}_\mu^\top\boldsymbol{W},\frac{1}{\sqrt{n}}\boldsymbol{x}_\mu^\top\boldsymbol{\theta},\boldsymbol{b}\right)$$
$$-n\log\int\mathrm{d}P_h(\boldsymbol{h})\exp\left\{\boldsymbol{h}^\top\boldsymbol{b}+\frac{1}{\sqrt{d}}\bar{v}\sum_{i=1}^d(\boldsymbol{h}^\top\boldsymbol{w}_i+\theta_i)+\frac{1}{2d}(\boldsymbol{h}^\top\boldsymbol{W}^\top\boldsymbol{W}\boldsymbol{h}+2\boldsymbol{h}^\top\boldsymbol{W}^\top\boldsymbol{\theta}+\boldsymbol{\theta}^\top\boldsymbol{\theta})\right\}. \tag{44}$$

In this case, we must have $\sum_{i=1}^d\boldsymbol{w}_i,\sum_{i=1}^d\theta_i=O\left(\sqrt{d}\right)$, because otherwise the second term in (44) is much larger than the first term. We cannot directly apply Theorem 4, because under Theorem 4 we have $\sum_{i=1}^d\boldsymbol{w}_i,\sum_{i=1}^d\theta_i=\Theta\left(\sqrt{d}\right)$, and thus the $\frac{1}{\sqrt{d}}\bar{v}\sum_{i=1}^d(\boldsymbol{h}^\top\boldsymbol{w}_i+\theta_i)$ term in (44) cannot be overlooked. One option is to consider the constrained optimization problem

$$\max_{\sum_{i=1}^d\boldsymbol{w}_i=\boldsymbol{0},\sum_{i=1}^d\theta_i=0,\boldsymbol{b}}\left(\sum_{\mu=1}^n\eta_1\left(\frac{1}{\sqrt{n}}\boldsymbol{x}_\mu^\top\boldsymbol{W},\frac{1}{\sqrt{n}}\boldsymbol{x}_\mu^\top\boldsymbol{\theta},\boldsymbol{b}\right)-n\eta_2\left(\frac{1}{d}\boldsymbol{W}^\top\boldsymbol{W},\frac{1}{d}\boldsymbol{W}^\top\boldsymbol{\theta},\frac{1}{d}\boldsymbol{\theta}^\top\boldsymbol{\theta},\boldsymbol{b}\right)\right), \tag{45}$$

which we leave as the future work.

## B.3 Gradients of the simplified log-likelihood

The gradient of the log-likelihood (5) w.r.t. the weights read

$$\nabla_{\boldsymbol{W}}\log\tilde{\mathcal{L}}(\boldsymbol{W},\boldsymbol{\theta},\boldsymbol{b})=\sum_{\mu=1}^n\frac{1}{\sqrt{d}}\frac{\int\mathrm{d}P_h(\boldsymbol{h})\boldsymbol{h}\boldsymbol{x}_\mu^\top e^{\frac{1}{\sqrt{d}}\boldsymbol{x}_\mu^\top\boldsymbol{W}\boldsymbol{h}+\boldsymbol{b}^T\boldsymbol{h}}}{\int\mathrm{d}P_h(\boldsymbol{h})e^{\frac{1}{\sqrt{d}}\boldsymbol{x}_\mu^\top\boldsymbol{W}\boldsymbol{h}+\boldsymbol{b}^T\boldsymbol{h}}}$$
$$-\alpha\frac{\int\mathrm{d}P_h(\boldsymbol{h})(\boldsymbol{h}\boldsymbol{h}^\top\boldsymbol{W}^\top+\boldsymbol{h}\boldsymbol{\theta}^\top)e^{\boldsymbol{h}^\top\boldsymbol{b}+(\boldsymbol{h}^\top\boldsymbol{Q}_W\boldsymbol{h}+2\boldsymbol{h}^\top\boldsymbol{Q}_{W\theta}+Q_\theta)/2}}{\int\mathrm{d}P_h(\boldsymbol{h})e^{\boldsymbol{h}^\top\boldsymbol{b}+(\boldsymbol{h}^\top\boldsymbol{Q}_W\boldsymbol{h}+2\boldsymbol{h}^\top\boldsymbol{Q}_{W\theta}+Q_\theta)/2}}, \tag{46}$$

where we denote $\boldsymbol{Q}_W:=\frac{1}{d}\boldsymbol{W}^\top\boldsymbol{W},\boldsymbol{Q}_{W\theta}:=\frac{1}{d}\boldsymbol{W}^\top\boldsymbol{\theta},Q_\theta:=\frac{1}{d}\boldsymbol{\theta}^\top\boldsymbol{\theta}$. This can be rewritten as

$$\nabla_{\boldsymbol{W}}\log\tilde{\mathcal{L}}(\boldsymbol{W},\boldsymbol{\theta},\boldsymbol{b}):=\sum_{\mu=1}^n\frac{1}{\sqrt{d}}\langle\boldsymbol{h}\boldsymbol{x}_\mu^\top\rangle_D-\alpha\langle\boldsymbol{h}\boldsymbol{h}^\top\rangle_H\boldsymbol{W}^\top-\alpha\langle\boldsymbol{h}\rangle_H\boldsymbol{\theta}^\top, \tag{47}$$

where we denote

$$\langle f(\boldsymbol{h})\rangle_D:=\frac{\int\mathrm{d}P_h(\boldsymbol{h})f(\boldsymbol{h})e^{\frac{1}{\sqrt{d}}\boldsymbol{x}_\mu^\top\boldsymbol{W}\boldsymbol{h}}}{\int\mathrm{d}P_h(\boldsymbol{h})e^{\frac{1}{\sqrt{d}}\boldsymbol{x}_\mu^\top\boldsymbol{W}\boldsymbol{h}}} \tag{48}$$

and

$$\langle f(\boldsymbol{h}) \rangle_H := \frac{\int \mathrm{d}P_h(\boldsymbol{h}) f(\boldsymbol{h}) e^{\boldsymbol{h}^\top \boldsymbol{Q} \boldsymbol{h}/2}}{\int \mathrm{d}P_h(\boldsymbol{h}) e^{\boldsymbol{h}^\top \boldsymbol{Q} \boldsymbol{h}/2}} \tag{49}$$

Note that the latter comes from a simple quadratic model with the interaction matrix $\boldsymbol{Q}(\boldsymbol{W}) = \frac{1}{d}\boldsymbol{W}^\top \boldsymbol{W}$ instead of the original model

$$\langle f(\boldsymbol{v}, \boldsymbol{h}) \rangle_{model} := \frac{\int \mathrm{d}P_v(\boldsymbol{v}) \mathrm{d}P_h(\boldsymbol{h}) f(\boldsymbol{v}, \boldsymbol{h}) e^{\frac{1}{\sqrt{d}}\boldsymbol{v}^\top \boldsymbol{W} \boldsymbol{h} + \frac{1}{\sqrt{d}}\boldsymbol{\theta}^\top \boldsymbol{v} + \boldsymbol{b}^\top \boldsymbol{h}}}{\int \mathrm{d}P_v(\boldsymbol{v}) \mathrm{d}P_h(\boldsymbol{h}) e^{\frac{1}{\sqrt{d}}\boldsymbol{v}^\top \boldsymbol{W} \boldsymbol{h} + \frac{1}{\sqrt{d}}\boldsymbol{\theta}^\top \boldsymbol{v} + \boldsymbol{b}^\top \boldsymbol{h}}}, \tag{50}$$

which suggests that in high dimensions, only the pairwise interaction of hidden units matters.

Similarly, we also have

$$\nabla_{\boldsymbol{\theta}} \log \tilde{\mathcal{L}}(\boldsymbol{W}, \boldsymbol{\theta}, \boldsymbol{b}) = \sum_{\mu=1}^{n} \frac{1}{\sqrt{d}} \langle \boldsymbol{x}_\mu \rangle_D - \alpha \boldsymbol{W} \langle \boldsymbol{h} \rangle_H - \alpha \boldsymbol{\theta},$$

$$\nabla_{\boldsymbol{b}} \log \tilde{\mathcal{L}}(\boldsymbol{W}, \boldsymbol{\theta}, \boldsymbol{b}) = \sum_{\mu=1}^{n} \frac{1}{\sqrt{d}} \langle \boldsymbol{h} \rangle_D - \alpha \langle \boldsymbol{h} \rangle_H. \tag{51}$$

All the gradients above take the form of the contrastive divergence.

## C  Proofs in Section 5

### C.1  Generalization of the data model

All results in Section 5 apply to the following data model

$$\boldsymbol{X} = \mathcal{F}\left(\frac{1}{\sqrt{d}}\boldsymbol{U}^* \boldsymbol{\Lambda} (\boldsymbol{W}^*)^\top + \boldsymbol{Z}\right) - \mathbb{E}\mathcal{F}(\boldsymbol{Z}) \in \mathbb{R}^{n \times d}, \tag{52}$$

under Assumption 3, where $\mathcal{F}$ is a general element-wise non-liearity and $\{Z_{ij}\}_{i,j=1}^{n,d} \overset{iid}{\sim} P_z$ is the noise.

**Assumption 3.**  *(i)  $P_z$ has a distribution $\mu_Z$ with stretched exponential tails[4].*

*(ii)  $\mathcal{F} \in C^2$ such that $\mathcal{F}' \in L^4(\mu_Z)$ and $\sup |\mathcal{F}''| < \infty$. Moreover, $\vartheta_1(\mathcal{F}) \neq 0$, where $\vartheta_1(\mathcal{F}) = \mathbb{E}_Z \mathcal{F}'(Z)$ for $Z \sim P_z$ is the information coefficient. We also assume that $\vartheta_0(\mathcal{F}^2) - \vartheta_0(\mathcal{F})^2 = 1$ i.e., the data are well normalized, where $\vartheta_0(\mathcal{F}) = \mathbb{E}_Z \mathcal{F}(Z)$.*

*(iii)  $P_u, P_w$ are bounded.*

Assumption 3 ensures that the data are generated from an effective linear spiked covariance model. It is mainly technical and might be relaxed. See e.g.,Assumptions $(H2)$, $(\tilde{H}3)$ in Guionnet et al. [2023] and Hypothesis 2.1 in Mergny et al. [2024].

More specifically, (52) is asymptotically equivalent to (11) because of the following lemma.

**Lemma 1.**  *Under Assumption 3, we have*

$$\frac{1}{\sqrt{n}}\boldsymbol{X} = \tilde{\boldsymbol{Z}} + \frac{\sqrt{\alpha}\vartheta_1(\mathcal{F})}{n}\boldsymbol{U}^* \boldsymbol{\Lambda} (\boldsymbol{W}^*)^\top + \boldsymbol{E}, \tag{53}$$

*where $\tilde{\boldsymbol{Z}}$ is a matrix with iid, centered elements with unit variance and $\boldsymbol{E}$ is the error matrix which vanishes $||\boldsymbol{E}|| = O(n^{-1/2})$ almost surely.*

Its proof follows from [Guionnet et al., 2023, Lemma 5.1], which we present in the following.

*Proof.*  We Taylor expand $\boldsymbol{X}$ around $\boldsymbol{Z}$, which gives

$$\frac{1}{\sqrt{n}}X_{ij} = \frac{1}{\sqrt{n}}(\mathcal{F}(Z_{ij}) - \mathbb{E}\mathcal{F}(Z_{ij})) + \frac{\sqrt{\alpha}}{n}\mathcal{F}'(Z_{ij})(\boldsymbol{u}_i^*)^\top \boldsymbol{\Lambda} \boldsymbol{w}_j^* + \frac{\alpha}{n^{3/2}}\mathcal{F}''(\zeta_{ij})((\boldsymbol{u}_i^*)^\top \boldsymbol{\Lambda} \boldsymbol{w}_j^*)^2, \tag{54}$$

---

[4]i.e., there exists $\alpha, c, C > 0$ such that for every $M > 0$, $\mathbb{P}(|Z| > M) \leq Ce^{-cM^\alpha}$

where $\zeta_{ij} \in [Z_{ij} - \frac{1}{\sqrt{d}}|(\boldsymbol{u}_i^*)^\top \boldsymbol{\Lambda} \boldsymbol{w}_j^*|, Z_{ij} + \frac{1}{\sqrt{d}}|(\boldsymbol{u}_i^*)^\top \boldsymbol{\Lambda} \boldsymbol{w}_j^*|]$. Therefore, we can define

$$\tilde{\boldsymbol{Z}} := \frac{1}{\sqrt{n}}(\mathcal{F}(\boldsymbol{Z}) - \mathbb{E}\mathcal{F}(\boldsymbol{Z})) \tag{55}$$

to be the effective noise and

$$E_{ij} := \frac{\sqrt{\alpha}}{n}(\mathcal{F}'(Z_{ij}) - \mathbb{E}\mathcal{F}'(Z_{ij}))(\boldsymbol{u}_i^*)^\top \boldsymbol{\Lambda} \boldsymbol{w}_j^* + \frac{\alpha}{n^{3/2}}\mathcal{F}''(\zeta_{ij})((\boldsymbol{u}_i^*)^\top \boldsymbol{\Lambda} \boldsymbol{w}_j^*)^2. \tag{56}$$

to be the error matrix. Note that we have $\mathbb{E}[\tilde{Z}_{ij}^2] = \frac{1}{n}$ by Assumption 3(ii). We can bound $||\boldsymbol{E}||$ by

$$
\begin{aligned}
||\boldsymbol{E}|| \leq &\frac{\sqrt{\alpha}}{\sqrt{n}} \sum_{k=1}^{r} \lambda_k \left\| \frac{1}{\sqrt{n}}(\mathcal{F}'(\boldsymbol{Z}) - \mathbb{E}\mathcal{F}'(\boldsymbol{Z})) \right\|_F ||\mathrm{Diag}(\boldsymbol{u}^{*(k)})||_F ||\mathrm{Diag}(\boldsymbol{w}^{*(k)})||_F \\
&+ \frac{\alpha}{n^{3/2}} \sum_{i,j=1}^{n,d} \mathcal{F}''(\zeta_{ij})((\boldsymbol{u}_i^*)^\top \boldsymbol{\Lambda} \boldsymbol{w}_j^*)^2,
\end{aligned}
\tag{57}
$$

where $\mathrm{Diag}(\boldsymbol{u}^{*(k)}) \in \mathbb{R}^{n \times n}, \mathrm{Diag}(\boldsymbol{w}^{*(k)}) \in \mathbb{R}^{d \times d}$ denote the diagonal matrices containing the $k-$th column of $\boldsymbol{U}^*$ and $\boldsymbol{W}^*$. We use the fact that the $L^2$ norm is always smaller than the Frobenius norm. The first term is $O(n^{-1/2})$ because $||\mathrm{Diag}(\boldsymbol{u}^{*(k)})||_F, ||\mathrm{Diag}(\boldsymbol{w}^{*(k)})||_F$ are bounded by Assumption 3 and that $\frac{1}{\sqrt{n}}(\mathcal{F}'(\boldsymbol{Z}) - \mathbb{E}\mathcal{F}'(\boldsymbol{Z}))$ is a matrix having iid, centered elements with finite forth moments by Assumption 3. It is well known that the largest singular value of such matrices almost surely converges (see e.g.,Jiang [2004]). The second term is $O(n^{-1/2})$ because each term in the sum is bounded by Assumption 3. This finishes the proof. $\square$

## C.2 General AMP Iterations

In this section we present a general version of the Approximate Message Passing (AMP) algorithm [Montanari and Venkataramanan, 2021] (and a variant of the AMP in Zhong et al. [2024]) for spiked covariance models, defined as the iterations

$$
\begin{aligned}
\boldsymbol{Y}^t &= \frac{1}{\sqrt{n}}\boldsymbol{X}\boldsymbol{W}^t - \sum_{i=1}^{t-1} \boldsymbol{B}_{ti}\boldsymbol{U}^i \in \mathbb{R}^{n \times k}, \ \boldsymbol{U}^t = f_t(\boldsymbol{Y}^t, \boldsymbol{Y}^{t-1}, \cdots, \boldsymbol{Y}^1, \boldsymbol{E}^t) \in \mathbb{R}^{n \times k}, \\
\boldsymbol{Z}^t &= \frac{1}{\sqrt{n}}\boldsymbol{X}^\top \boldsymbol{U}^t - \sum_{i=1}^{t-1} \boldsymbol{C}_{ti}\boldsymbol{W}^i \in \mathbb{R}^{d \times k}, \ \boldsymbol{W}^{t+1} = g_{t+1}(\boldsymbol{Z}^t, \boldsymbol{Z}^{t-1}, \cdots, \boldsymbol{Z}_1, \boldsymbol{F}^t) \in \mathbb{R}^{d \times k},
\end{aligned}
\tag{58}
$$

where Onsager coefficients $\boldsymbol{B}_{ti}, \boldsymbol{C}_{ti} \in \mathbb{R}^{k \times k}$ are given by

$$\boldsymbol{B}_{t+1,i} = \frac{1}{n}\sum_{j=1}^{d} \frac{\partial f_t}{\partial \boldsymbol{z}_j^i}(\boldsymbol{z}_j^t, \boldsymbol{z}_j^{t-1}, \cdots, \boldsymbol{z}_j^1, \boldsymbol{E}^t), \ \boldsymbol{C}_{t+1,i} = \frac{1}{n}\sum_{\mu=1}^{n} \frac{\partial g_{t+1}}{\partial \boldsymbol{y}_\mu^i}(\boldsymbol{y}_\mu^t, \boldsymbol{y}_\mu^{t-1}, \cdots, \boldsymbol{y}_\mu^1, \boldsymbol{F}^t) \tag{59}$$

for $i = 1, \cdots, t$. $\boldsymbol{E}^t, \boldsymbol{F}^t$ are generic side information of finite dimensions. Its SE is given by a sequence of $k$-dimensional random variables $\{\mathring{\boldsymbol{U}}_t, \mathring{\boldsymbol{Y}}_t, \mathring{\boldsymbol{W}}_t, \mathring{\boldsymbol{Z}}_t\}_{t \geq 1}$ defined as follows

$$
\begin{aligned}
\mathring{\boldsymbol{U}}_t &= f_t(\mathring{\boldsymbol{Y}}_t, \mathring{\boldsymbol{Y}}_{t-1}, \cdots, \mathring{\boldsymbol{Y}}_1, \mathring{\boldsymbol{E}}), \\
\mathring{\boldsymbol{W}}_{t+1} &= g_{t+1}(\mathring{\boldsymbol{Z}}_t, \mathring{\boldsymbol{Z}}_{t-1}, \cdots, \mathring{\boldsymbol{Z}}_1, \mathring{\boldsymbol{F}}),
\end{aligned}
\tag{60}
$$

where

$$
\begin{aligned}
(\mathring{\boldsymbol{Y}}_1, \cdots, \mathring{\boldsymbol{Y}}_t) &= (\mathring{\boldsymbol{M}}_1, \cdots, \mathring{\boldsymbol{M}}_t)\boldsymbol{U} + \mathcal{N}(0, \mathring{\boldsymbol{\Sigma}}_t), \\
(\mathring{\boldsymbol{Z}}_1, \cdots, \mathring{\boldsymbol{Z}}_t) &= (\mathring{\boldsymbol{N}}_1, \cdots, \mathring{\boldsymbol{N}}_t)\boldsymbol{W} + \mathcal{N}(0, \mathring{\boldsymbol{\Omega}}_t),
\end{aligned}
\tag{61}
$$

and

$$
\begin{aligned}
\mathring{\boldsymbol{M}}_t &= \alpha^{-1}\mathbb{E}[\mathring{\boldsymbol{W}}_t^\top \boldsymbol{W}]\boldsymbol{\Gamma}, \ [\mathring{\boldsymbol{\Sigma}}_t]_{ij} = \alpha^{-1}\mathbb{E}[\mathring{\boldsymbol{W}}_i^\top \boldsymbol{W}_j], \\
\mathring{\boldsymbol{N}}_{t+1} &= \mathbb{E}[\mathring{\boldsymbol{U}}_{t+1}^\top \boldsymbol{U}]\boldsymbol{\Gamma}, \ [\mathring{\boldsymbol{\Omega}}_{t+1}]_{ij} = \mathbb{E}[\mathring{\boldsymbol{U}}_i^\top \mathring{\boldsymbol{U}}_j],
\end{aligned}
\tag{62}
$$

where $\boldsymbol{\Gamma} := \sqrt{\alpha}\boldsymbol{\Lambda}\vartheta_1(\mathcal{F}) \in \mathbb{R}^{k \times k}$. $\mathring{\boldsymbol{M}}_1$ and $[\mathring{\boldsymbol{\Sigma}}_t]_{11}$ are given by initialization. We note that $\mathring{\boldsymbol{M}}_t, \mathring{\boldsymbol{N}}_t \in \mathbb{R}^{k \times k}$ and $\mathring{\boldsymbol{\Sigma}}_t, \mathring{\boldsymbol{\Omega}}_t \in \mathbb{R}^{k \times k \times t \times t}$. The following theorem suggests that the iterations converge to the SE.

**Theorem 6.** *Under Assumption 3 or for $\mathcal{F}(x) = x$, and further assuming that*

- *(i) the iterations are initialized with $\boldsymbol{W}^0 \sim \mathring{\boldsymbol{W}}_0$ independent of $\boldsymbol{X}$, having finite moments of all orders,*

- *(ii) $\boldsymbol{E}^t, \boldsymbol{F}^t$ are of finite dimensions and almost surely converge to $\mathring{\boldsymbol{E}}^t, \mathring{\boldsymbol{F}}^t$,*

- *(iii) $f_t, g_t$ are separable, continuously differentiable and uniformly Lipschitz at $\mathring{\boldsymbol{E}}^t, \mathring{\boldsymbol{F}}^t$ w.r.t. $(\boldsymbol{Y}^t, \boldsymbol{Y}^{t-1}, \cdots, \boldsymbol{Y}^1), (\boldsymbol{Z}^t, \boldsymbol{Z}^{t-1}, \cdots, \boldsymbol{Z}^1)$,*

*then for any $PL(2)$ function $\psi$, the following holds almost surely for $t \geq 0$:*

$$
\begin{aligned}
\lim_{d \to \infty} \frac{1}{d} \sum_{i=1}^{d} \psi(\boldsymbol{w}_i^*, \boldsymbol{w}_i^1, \cdots, \boldsymbol{w}_i^t, \boldsymbol{u}_i^*, \boldsymbol{u}_i^1, \cdots, \boldsymbol{u}_i^t, \boldsymbol{y}_i^1, \cdots, \boldsymbol{y}_i^t, \boldsymbol{z}_i^1, \cdots, \boldsymbol{z}_i^t) \\
= \mathbb{E}[\psi(\boldsymbol{W}, \mathring{\boldsymbol{W}}_1, \cdots, \mathring{\boldsymbol{W}}_t, \mathring{\boldsymbol{U}}_1, \cdots, \mathring{\boldsymbol{U}}_t, \mathring{\boldsymbol{Y}}_1, \cdots, \mathring{\boldsymbol{Y}}_t, \mathring{\boldsymbol{Z}}_1, \cdots, \mathring{\boldsymbol{Z}}_t)],
\end{aligned}
\tag{63}
$$

Theorem 3 in the main text is a special case of Theorem 6.

Theorem 6 is very similar to [Fan, 2022, Theorem 3.4] and [Zhong et al., 2024, Theorem 2.3], and the only difference is to deal with the non-linearity and the side information, so we only provide its sketch.

By Lemma 1, we only need to show that (58) is aymptotically equivalent to

$$
\begin{aligned}
\boldsymbol{Y}^t = \tilde{\boldsymbol{X}}\boldsymbol{W}^t - \sum_{i=1}^{t-1} \boldsymbol{B}_{ti}\boldsymbol{U}^i \in \mathbb{R}^{n \times k}, \ \boldsymbol{U}^t = f_t(\boldsymbol{Y}^t, \boldsymbol{Y}^{t-1}, \cdots, \boldsymbol{Y}^1, \mathring{\boldsymbol{E}}) \in \mathbb{R}^{n \times k}, \\
\boldsymbol{Z}^t = \tilde{\boldsymbol{X}}^\top \boldsymbol{U}^t - \sum_{i=1}^{t-1} \boldsymbol{C}_{ti}\boldsymbol{W}^i \in \mathbb{R}^{d \times k}, \ \boldsymbol{W}^{t+1} = g_{t+1}(\boldsymbol{Z}^t, \boldsymbol{Z}^{t-1}, \cdots, \boldsymbol{Z}_1, \mathring{\boldsymbol{F}}) \in \mathbb{R}^{d \times k},
\end{aligned}
\tag{64}
$$

where $\tilde{\boldsymbol{X}} := \tilde{\boldsymbol{Z}} + \frac{\sqrt{\alpha}\vartheta_1(\mathcal{F})}{n}\boldsymbol{U}^*\boldsymbol{\Lambda}(\boldsymbol{W}^*)^\top$, because standard AMP results [Zhong et al., 2024] together with the universality results [Chen and Lam, 2021] give its SE in Theorem 6.

In fact we can replace $\boldsymbol{E}^t, \boldsymbol{F}^t$ with $\mathring{\boldsymbol{E}}, \mathring{\boldsymbol{F}}$ because of the almost sure convergence and the uniform Lipschitz property. See Pandit et al. [2020] for a similar argument. Moreover, as the error $\boldsymbol{E}$ has vanishing norm, we can inductively show that $\frac{1}{\sqrt{n}}\boldsymbol{X}$ in the iteration (58) can be replaced by $\tilde{\boldsymbol{X}}$ with $O(n^{-1/2})$ error. See Mergny et al. [2024] for a similar argument. This finishes the proof of Theorem 6.

### C.3 Proof of Theorem 2

*Proof.* We first prove that $\boldsymbol{B}_t$ and $\boldsymbol{C}_t$ are always symmetric by induction. Suppose that $\boldsymbol{C}_{t-1}$ is symmetric (which holds at $t = 1$). We define the following Moreau envelop

$$
F(\boldsymbol{z}^t, \hat{\boldsymbol{Q}}^t, \boldsymbol{C}_{t-1}) := \min_{\boldsymbol{w}} \left\{ \boldsymbol{w}^\top \hat{\boldsymbol{Q}}^t \boldsymbol{w} + \frac{1}{2}(\boldsymbol{z}^t - \boldsymbol{w})^\top \boldsymbol{C}_{t-1}^{-1}(\boldsymbol{z}^t - \boldsymbol{w}) \right\}.
\tag{65}
$$

By the property of the Moreau envelop, we have

$$
\nabla_z F(\boldsymbol{z}^t, \hat{\boldsymbol{Q}}^t, \boldsymbol{C}_{t-1}) = \boldsymbol{C}_{t-1}^{-1}(\boldsymbol{z}^t + \boldsymbol{C}_{t-1}f(\boldsymbol{z}^t, \hat{\boldsymbol{Q}}^t, \boldsymbol{C}_{t-1})).
\tag{66}
$$

Then

$$
\nabla_z f(\boldsymbol{z}^t, \hat{\boldsymbol{Q}}^t, \boldsymbol{C}_{t-1})) = \nabla_z^2 F(\boldsymbol{z}^t, \hat{\boldsymbol{Q}}^t, \boldsymbol{C}_{t-1}) - \boldsymbol{C}_{t-1}^{-1}
\tag{67}
$$

is symmetric, and thus $\boldsymbol{B}_t$ is symmetric. Similarly we can show that $\boldsymbol{C}_t$ is symmetric.

Now we denote the fixed point values of $\boldsymbol{U}^t, \boldsymbol{W}^t, \boldsymbol{Y}^t, \boldsymbol{Z}^t, \boldsymbol{B}^t, \boldsymbol{C}^t, \hat{\boldsymbol{Q}}^t$ to be $\hat{\boldsymbol{U}}, \hat{\boldsymbol{W}}, \boldsymbol{Y}, \boldsymbol{Z}, \boldsymbol{B}, \boldsymbol{C}, \hat{\boldsymbol{Q}}$, respectively. At the fixed point of AMP-RBM we have

$$
\begin{aligned}
\boldsymbol{Y} &= \frac{1}{\sqrt{n}}\boldsymbol{X}\hat{\boldsymbol{W}} - \hat{\boldsymbol{U}}\boldsymbol{B}^\top, \\
\boldsymbol{Z} &= \frac{1}{\sqrt{n}}\boldsymbol{X}^\top \hat{\boldsymbol{U}} - \hat{\boldsymbol{W}}\boldsymbol{C}^\top.
\end{aligned}
\tag{68}
$$

According to the definition of $f$ and $g$, we have

$$
\begin{aligned}
(\boldsymbol{C} + \hat{\boldsymbol{Q}})\hat{\boldsymbol{W}}^\top + \boldsymbol{Z}^\top &= 0, \\
\alpha^{-1}\nabla\eta_1(\hat{\boldsymbol{u}}_\mu \boldsymbol{B}^\top + \boldsymbol{y}_\mu) + \hat{\boldsymbol{u}}_\mu &= 0, \ \mu = 1, 2, \cdots, n,
\end{aligned}
\tag{69}
$$

where we use the first-order conditions of (16), (17) and fact that $\boldsymbol{B}$ and $\boldsymbol{C}$ are symmetric. Taking the first equality of (68) into the second equality of (69), we have

$$
\alpha^{-1}\nabla\eta_1(\boldsymbol{x}_\mu^\top \hat{\boldsymbol{W}}/\sqrt{n}) + \hat{\boldsymbol{u}}_\mu = 0.
\tag{70}
$$

Taking it into the second equality of (68), and using the first equality of (69), we have

$$
\sum_{\mu=1}^n \frac{1}{\sqrt{n}}\alpha^{-1}\nabla\eta_1(\boldsymbol{x}_\mu^\top \hat{\boldsymbol{W}}/\sqrt{n}) \otimes \boldsymbol{x}_\mu = -\boldsymbol{Z} - \hat{\boldsymbol{W}}\boldsymbol{C}^\top = \hat{\boldsymbol{W}}\hat{\boldsymbol{Q}}^\top.
\tag{71}
$$

Moreover, at the fixed point we have $\hat{\boldsymbol{Q}} = \nabla\eta_2(\boldsymbol{Q}(\hat{\boldsymbol{W}}))$ by the last line of Algorithm 1, which thus gives (18). $\qquad\square$

### C.4 Proof of Theorem 4

*Proof.* By the triangle inequality, we have

$$
\begin{aligned}
&\left| \frac{1}{d}\sum_{i=1}^d \psi(\boldsymbol{w}_i^*, \hat{\boldsymbol{w}}_i) - \mathbb{E}[\psi(\boldsymbol{W}, f(\boldsymbol{M}_t\boldsymbol{W} + \Sigma_t^{1/2}\boldsymbol{G}, \bar{\hat{\boldsymbol{Q}}}_t, \bar{\boldsymbol{C}}_{t-1}))] \right| \\
&\leq \left| \frac{1}{d}\sum_{i=1}^d \psi(\boldsymbol{w}_i^*, \hat{\boldsymbol{w}}_i) - \frac{1}{d}\sum_{i=1}^d \psi(\boldsymbol{w}_i^*, \hat{\boldsymbol{w}}_i^t) \right| \\
&\quad + \left| \frac{1}{d}\sum_{i=1}^d \psi(\boldsymbol{w}_i^*, \hat{\boldsymbol{w}}_i^t) - \mathbb{E}[\psi(\boldsymbol{W}, f(\boldsymbol{M}_t\boldsymbol{W} + \Sigma_t^{1/2}\boldsymbol{G}, \bar{\hat{\boldsymbol{Q}}}_t, \bar{\boldsymbol{C}}_{t-1}))] \right|.
\end{aligned}
\tag{72}
$$

For the second term, Theorem 3 indicates that

$$
\lim_{d\to\infty} \left| \frac{1}{d}\sum_{i=1}^d \psi(\boldsymbol{w}_i^*, \hat{\boldsymbol{w}}_i^t) - \mathbb{E}[\psi(\boldsymbol{W}, f(\boldsymbol{M}_t\boldsymbol{W} + \Sigma_t^{1/2}\boldsymbol{G}, \bar{\hat{\boldsymbol{Q}}}_t, \bar{\boldsymbol{C}}_{t-1}))] \right| = 0
\tag{73}
$$

almost surely. For the first term, by the pseudo-Lipschitz property of $\psi$, we have

$$
\begin{aligned}
&\left| \frac{1}{d}\sum_{i=1}^d \psi(\boldsymbol{w}_i^*, \hat{\boldsymbol{w}}_i) - \frac{1}{d}\sum_{i=1}^d \psi(\boldsymbol{w}_i^*, \hat{\boldsymbol{w}}_i^t) \right| \\
&\leq \frac{C}{d}\sum_{i=1}^d (1 + 2\|\boldsymbol{w}_i^*\| + \|\hat{\boldsymbol{w}}_i^t\| + \|\hat{\boldsymbol{w}}_i\|)\|\hat{\boldsymbol{w}}_i^t - \hat{\boldsymbol{w}}_i\| \\
&\leq \frac{C}{d}\sum_{i=1}^d (1 + 2\|\boldsymbol{w}_i^*\| + 2\|\hat{\boldsymbol{w}}_i\|)\|\hat{\boldsymbol{w}}_i^t - \hat{\boldsymbol{w}}_i\| + \frac{C}{d}\sum_{i=1}^d \|\hat{\boldsymbol{w}}_i^t - \hat{\boldsymbol{w}}_i\|^2 \\
&\leq C\sqrt{\frac{1}{d}\sum_{i=1}^d (1 + 2\|\boldsymbol{w}_i^*\| + 2\|\hat{\boldsymbol{w}}_i\|)^2}\sqrt{\frac{1}{d}\sum_{i=1}^d \|\hat{\boldsymbol{w}}_i^t - \hat{\boldsymbol{w}}_i\|^2} + \frac{C}{d}\sum_{i=1}^d \|\hat{\boldsymbol{w}}_i^t - \hat{\boldsymbol{w}}_i\|^2.
\end{aligned}
\tag{74}
$$

Taking $d \to \infty$ and then $t \to \infty$, by Assumption 2 we have

$$
\lim_{t\to\infty}\lim_{d\to\infty} \left| \frac{1}{d}\sum_{i=1}^d \psi(\boldsymbol{w}_i^*, \hat{\boldsymbol{w}}_i) - \frac{1}{d}\sum_{i=1}^d \psi(\boldsymbol{w}_i^*, \hat{\boldsymbol{w}}_i^t) \right| = 0.
\tag{75}
$$

Therefore we have almost surely

$$
\lim_{t\to\infty}\lim_{d\to\infty} \left| \frac{1}{d}\sum_{i=1}^d \psi(\boldsymbol{w}_i^*, \hat{\boldsymbol{w}}_i) - \mathbb{E}[\psi(\boldsymbol{W}, f(\boldsymbol{M}_t\boldsymbol{W} + \Sigma_t^{1/2}\boldsymbol{G}, \bar{\hat{\boldsymbol{Q}}}_t, \bar{\boldsymbol{C}}_{t-1}))] \right| = 0.
\tag{76}
$$

By [Emami et al., 2020, Lemma 1] we have

$$\lim_{d \to \infty} \frac{1}{d} \sum_{i=1}^{d} \psi(\boldsymbol{w}_i^*, \hat{\boldsymbol{w}}_i) = \lim_{t \to \infty} \mathbb{E}[\psi(\boldsymbol{W}, f(\boldsymbol{M}_t \boldsymbol{W} + \boldsymbol{\Sigma}_t^{1/2} \boldsymbol{G}, \bar{\hat{\boldsymbol{Q}}}_t, \bar{\boldsymbol{C}}_{t-1}))] \tag{77}$$

almost surely and the limits on both sides exist.

Finally let us verify that the limit of $\boldsymbol{M}_t$ exists. By Theorem 3 we have

$$\lim_{t \to \infty} \boldsymbol{M}_t = \lim_{t \to \infty} \lim_{d \to \infty} \boldsymbol{z}_i^t (\boldsymbol{w}_i^*)^\top (\boldsymbol{w}_i^* (\boldsymbol{w}_i^*)^\top)^{-1} = \lim_{d \to \infty} \hat{\boldsymbol{z}}_i (\boldsymbol{w}_i^*)^\top (\boldsymbol{w}_i^* (\boldsymbol{w}_i^*)^\top)^{-1} \tag{78}$$

almost surely, where for the first equality we use Theorem 6 and for the second equality we use Assumption 2 to replace $\boldsymbol{z}_i^t$ with $\hat{\boldsymbol{z}}_i$. By similar arguments as above we have

$$\lim_{d \to \infty} \frac{1}{d} \sum_{i=1}^{d} \psi(\boldsymbol{w}_i^*, \hat{\boldsymbol{z}}_i) = \lim_{t \to \infty} \mathbb{E}[\psi(\boldsymbol{W}, \boldsymbol{M}_t \boldsymbol{W} + \boldsymbol{\Sigma}_t^{1/2} \boldsymbol{G})], \tag{79}$$

almost surely, which suggests that the limit on the right side of (78) exists, and thus $\lim_{t \to \infty} \boldsymbol{M}_t$ exists. In a similar way we can show that the limits of $\bar{\boldsymbol{C}}_t, \boldsymbol{\Sigma}_t, \bar{\hat{\boldsymbol{Q}}}_t$ exist.

A direct combination of (32) and Theorem 2 suggests that $\lim_{d \to \infty} ||\sqrt{d} \nabla \log \mathcal{L}(\hat{\boldsymbol{W}}(d))|| = 0$ at the stationary point. Then by (79) we finish the proof. $\qquad \square$

## C.5  Proof of Corollary 1

*Proof.* We only need to iteratively prove that all matrices in (19) are diagonal. Assuming that $\boldsymbol{M}_t, \boldsymbol{\Sigma}_t, \bar{\hat{\boldsymbol{Q}}}_t, \bar{\boldsymbol{C}}_{t-1}$ are diagonal, which holds at initialization, we first have

$$\mathbb{E}[f(M_{t,ii} W_i + \sqrt{\Sigma_{t,ii}} G, \bar{\hat{\boldsymbol{Q}}}_t, \bar{\boldsymbol{C}}_{t-1}) W_j] = \mathbb{E}[f(M_{t,ii} W_i + \sqrt{\Sigma_{t,ii}} G, \bar{\hat{\boldsymbol{Q}}}_t, \bar{\boldsymbol{C}}_{t-1})] \mathbb{E}[W_j] = 0, \tag{80}$$

for $1 \le i \ne j \le k$ and thus $\bar{\boldsymbol{M}}_t$ is a diagonal matrix, where $\mathbb{E}[W_j] = 0$ because $P_w$ is symmetric. Moreover, we have

$$\bar{\Sigma}_{t,ij} = \alpha^{-1} \mathbb{E}[f(M_{t,ii} W_i + \sqrt{\Sigma_{t,ii}} G, \bar{\hat{\boldsymbol{Q}}}_t, \bar{\boldsymbol{C}}_{t-1})] \mathbb{E}[f(M_{t,jj} W_j + \boldsymbol{\Sigma}_t^{1/2} \boldsymbol{G}, \bar{\hat{\boldsymbol{Q}}}_t, \bar{\boldsymbol{C}}_{t-1})^\top] = 0, \tag{81}$$

$1 \le i \ne j \le k$, so $\bar{\boldsymbol{\Sigma}}$ and $\bar{\boldsymbol{Q}}_t$ are diagonal, where $\mathbb{E}[f(M_{t,ii} W_i + \sqrt{\Sigma_{t,ii}} G, \bar{\hat{\boldsymbol{Q}}}_t, \bar{\boldsymbol{C}}_{t-1})] = 0$ because $P_w$ is symmetric and $f(\cdot, \bar{\hat{\boldsymbol{Q}}}_t, \bar{\boldsymbol{C}}_{t-1})$ is an odd function. As $f(\cdot, \bar{\hat{\boldsymbol{Q}}}_t, \bar{\boldsymbol{C}}_{t-1})$ is separable, $\bar{\boldsymbol{B}}_t$ is diagonal.

When $P_h$ is separable and symmetric, $\eta_1$ is an odd function, and thus $g(\cdot, \bar{\boldsymbol{B}}_t)$ is an odd function. Similarly, $\boldsymbol{M}_{t+1}, \boldsymbol{\Sigma}_{t+1}, \bar{\boldsymbol{C}}_t$ are diagonal. Finally, as $\bar{\boldsymbol{Q}}_t$ is diagonal and $P_h$ is separable and symmetric, $\nabla \eta_2(\bar{\boldsymbol{Q}}_t)$ is diagonal, so $\bar{\hat{\boldsymbol{Q}}}_t$ is diagonal, which finishes the proof. $\qquad \square$

## C.6  Proof of Theorem 5

We now provide a detailed definition of the functions $\mathring{f}_t, \mathring{g}_{t+1} : \mathbb{R}^{k \times t} \to \mathbb{R}^k$ as follows. We define

$$\mathring{f}_t(\mathring{\boldsymbol{Y}}_t, \mathring{\boldsymbol{Y}}_{t-1}, \cdots, \mathring{\boldsymbol{Y}}_1) = \tilde{f}(\mathring{\boldsymbol{Y}}_t + \sum_{i=1}^{t-1} \mathring{\boldsymbol{B}}_{ti} \mathring{\boldsymbol{U}}_i),$$

$$\mathring{g}_{t+1}(\mathring{\boldsymbol{Z}}_t, \mathring{\boldsymbol{Z}}_{t-1}, \cdots, \mathring{\boldsymbol{Z}}_1) = \tilde{g}(\mathring{\boldsymbol{Z}}_t + \sum_{i=1}^{t} \mathring{\boldsymbol{C}}_{ti} \mathring{\boldsymbol{W}}_i, \mathring{\boldsymbol{W}}_t, \mathring{\boldsymbol{Q}}_t), \tag{82}$$

where $\mathring{\boldsymbol{U}}_i$ should be regarded as a function of $\mathring{\boldsymbol{Y}}_i, \mathring{\boldsymbol{Y}}_{i-1}, \cdots, \mathring{\boldsymbol{Y}}_1$ as defined in (26), and similarly for $\mathring{\boldsymbol{W}}_i$. $\tilde{f}, \tilde{g}$ are separable functions defined by $\tilde{f}(\boldsymbol{y}) := \nabla \eta_1(\boldsymbol{y}) \in \mathbb{R}^k$ and $\tilde{g}(\boldsymbol{z}, \hat{\boldsymbol{w}}, \hat{\boldsymbol{Q}}) := \hat{\boldsymbol{w}} + \kappa(\boldsymbol{z} - \alpha \hat{\boldsymbol{Q}} \hat{\boldsymbol{w}}) \in \mathbb{R}^k$ for $\boldsymbol{y}, \boldsymbol{z}, \hat{\boldsymbol{w}} \in \mathbb{R}^k$. Here $\mathring{\boldsymbol{Q}}_t = \mathbb{E}[\nabla \eta_2(\boldsymbol{Q}(\mathring{\boldsymbol{W}}_t))] \in \mathbb{R}^{k \times k}$ and Onsager coefficients $\mathring{\boldsymbol{B}}_{ti}, \mathring{\boldsymbol{C}}_{ti} \in \mathbb{R}^{k \times k}$ are given by

$$\mathring{\boldsymbol{B}}_{t+1,i} = \alpha^{-1} \mathbb{E}[\partial_i \mathring{g}_{t+1}(\mathring{\boldsymbol{Z}}_t, \mathring{\boldsymbol{Z}}_{t-1}, \cdots, \mathring{\boldsymbol{Z}}_1)], \ \mathring{\boldsymbol{C}}_{ti} = \mathbb{E}[\partial_i \mathring{f}_t(\mathring{\boldsymbol{Y}}_t, \mathring{\boldsymbol{Y}}_{t-1}, \cdots, \mathring{\boldsymbol{Y}}_1)], \tag{83}$$

for $i = 1, \cdots, t$, where $\partial_i \mathring{f}_t : \mathbb{R}^{k \times t} \to \mathbb{R}^{k \times k}$ denotes the partial derivative of $\mathring{f}_t$ w.r.t. $\mathring{\boldsymbol{Y}}_i$, and $\partial_i \mathring{g}_{t+1} : \mathbb{R}^{k \times t} \to \mathbb{R}^{k \times k}$ denotes the partial derivative of $\mathring{g}_{t+1}$ w.r.t. $\mathring{\boldsymbol{Z}}_i$. The derivatives can be calculated recursively

$$\partial_i \mathring{f}_t(\mathring{\boldsymbol{Y}}_t, \cdots, \mathring{\boldsymbol{Y}}_1) = \nabla \tilde{f}(\mathring{\boldsymbol{Y}}^t + \sum_{a=1}^{t-1} \mathring{\boldsymbol{B}}_{ta} \mathring{\boldsymbol{U}}_a)(\delta_{ti} \boldsymbol{I}_{k \times k} + \sum_{j=i}^{t-1} \mathring{\boldsymbol{B}}_{tj} \partial_i \mathring{f}_j(\mathring{\boldsymbol{Y}}_j, \cdots, \mathring{\boldsymbol{Y}}_1)), \quad (84)$$

$$\begin{aligned}
\partial_i \mathring{g}^{t+1}(\mathring{\boldsymbol{Z}}_t, \cdots, \mathring{\boldsymbol{Z}}_1) = {}& \delta_{ti} \partial_1 g(\mathring{\boldsymbol{Z}}_t + \sum_{a=1}^{t-1} \mathring{\boldsymbol{C}}_{ta} \mathring{\boldsymbol{W}}_a, \mathring{\boldsymbol{W}}_t) \\
& + \sum_{j=i+1}^{t-1} \mathring{\boldsymbol{C}}_{tj} \partial_1 \tilde{g}(\mathring{\boldsymbol{Z}}_t + \sum_{a=1}^{t-1} \mathring{\boldsymbol{C}}_{ta} \mathring{\boldsymbol{W}}_a, \mathring{\boldsymbol{W}}_t) \partial_i \mathring{g}^j(\mathring{\boldsymbol{Z}}_j, \cdots, \mathring{\boldsymbol{Z}}_1) \quad (85) \\
& + \partial_2 \tilde{g}(Z^t + \sum_{a=1}^{t-1} \mathring{\boldsymbol{C}}_{ta} \mathring{\boldsymbol{W}}_a, \mathring{\boldsymbol{W}}_t) \partial_i \mathring{g}^t(\mathring{\boldsymbol{Z}}_{t-1}, \cdots, \mathring{\boldsymbol{Z}}_1),
\end{aligned}$$

where $\nabla \tilde{f} : \mathbb{R}^k \to \mathbb{R}^{k \times k}$ denotes the derivative of $\tilde{f}$, and $\partial_1 \tilde{g}, \partial_2 \tilde{g} : \mathbb{R}^{k \times 2} \to \mathbb{R}^{k \times k}$ denotes the derivatives of $\tilde{g}$ w.r.t. its first and second variable.

We can notice that (25) can be rewritten as

$$\begin{aligned}
\boldsymbol{Y}^t &= \frac{1}{\sqrt{n}} \boldsymbol{X} \tilde{\boldsymbol{W}}^t \in \mathbb{R}^{n \times k}, \ \tilde{\boldsymbol{U}}^t = \tilde{f}(\boldsymbol{Y}^t) \in \mathbb{R}^{n \times k}, \ \hat{\boldsymbol{Q}}^t = \nabla \eta_2(\boldsymbol{Q}(\tilde{\boldsymbol{W}}^t)) \in \mathbb{R}^{k \times k} \\
\boldsymbol{Z}^t &= \frac{1}{\sqrt{n}} \boldsymbol{X}^\top \tilde{\boldsymbol{U}}^t \in \mathbb{R}^{d \times k}, \ \tilde{\boldsymbol{W}}^{t+1} = \tilde{g}(\boldsymbol{Z}^t, \tilde{\boldsymbol{W}}^t, \hat{\boldsymbol{Q}}^t) \in \mathbb{R}^{d \times k}.
\end{aligned} \quad (86)$$

Then it suffices to map (86) to a general AMP iteration as follows

$$\begin{aligned}
\tilde{\boldsymbol{Y}}^t &= \frac{1}{\sqrt{n}} \boldsymbol{X} \tilde{\boldsymbol{W}}^t - \sum_{i=1}^{t-1} \mathring{\boldsymbol{B}}_{ti} \tilde{\boldsymbol{U}}^i, \ \tilde{\boldsymbol{U}}^t = \mathring{f}_t(\tilde{\boldsymbol{Y}}^t, \tilde{\boldsymbol{Y}}^{t-1}, \cdots, \tilde{\boldsymbol{Y}}^1), \ \hat{\boldsymbol{Q}}^t = \nabla \eta_2(\boldsymbol{Q}(\tilde{\boldsymbol{W}}^t)) \\
\tilde{\boldsymbol{Z}}^t &= \frac{1}{\sqrt{n}} \boldsymbol{X}^\top \tilde{\boldsymbol{U}}^t - \sum_{i=1}^{t-1} \mathring{\boldsymbol{C}}_{ti} \tilde{\boldsymbol{W}}^i, \ \tilde{\boldsymbol{W}}^{t+1} = \mathring{g}_{t+1}(\tilde{\boldsymbol{Z}}^t, \tilde{\boldsymbol{Z}}^{t-1}, \cdots, \tilde{\boldsymbol{Z}}_1),
\end{aligned} \quad (87)$$

which is a special case of (58). Therefore, the first part of Theorem 5 simply follows from Theorem 6. The second part of Theorem 5 is the same as Theorem 4, so we omit its proof.

## D From stationary points to global optimum

In this section, we will generalize the results of Vilucchio et al. [2025] to the spiked covariance model with non-separable constraints. We will show that for a special case, the global optimum is among the stationary points we describe in Theorem 4. The idea is that Theorem 4 provides an upper bound of the minimum, and we can find a matching lower bound by Gordon's Gaussian comparison inequality [Gordon, 1988]. Specifically, we will consider the data model with one spike and Gaussian noise, i.e.,

$$\boldsymbol{X} = \frac{\lambda}{\sqrt{d}} \boldsymbol{u}^*(\boldsymbol{w}^*)^\top + \boldsymbol{Z} \in \mathbb{R}^{n \times d}, \quad (88)$$

where $\{u_i^*\}_{i=1}^n \overset{iid}{\sim} P_u$, $\{w_i^*\}_{i=1}^d \overset{iid}{\sim} P_w$ and $\{Z_{ij}\}_{i,j=1}^{n,d} \overset{iid}{\sim} \mathcal{N}(0, 1)$. We will consider the following optimization problem

$$\begin{aligned}
\mathcal{A}_d &:= \inf_{\boldsymbol{w} \in \mathbb{R}^d} \mathcal{L}(\boldsymbol{w}) \\
&:= \inf_{\boldsymbol{w} \in \mathbb{R}^d} \frac{1}{d} \sum_{\mu=1}^n \eta_1\left(\frac{1}{\sqrt{d}} \boldsymbol{x}_\mu^\top \boldsymbol{w}\right) + \frac{1}{d} \sum_{i=1}^d \eta_2\left(w_i, \frac{||\boldsymbol{w}||_r^r}{d}\right)
\end{aligned} \quad (89)$$

as a special case of (15) with $k = 1$. We change some scalings for notational convenience. Although (15) only requires $r = 2$, we find that it is more convenient to state the general result for any $r > 0$.

**Remark 1.** *It is easy to modify our results to include the biases of RBMs. However, the restrictions $k = 1$ and Gaussian $\mathbf{Z}$ cannot be easily generalized as they are required by the Gordon's Gaussian comparison inequality (Lemma 4). We expect that this is a purely technical problem and can be resolved in the future.*

Our first assumption concerning (89) is the following.

**Assumption 4.** *The global minimum $\mathbf{w}$ of (89) satisfies $\frac{1}{d}||\mathbf{w}||^2, \frac{1}{d}||\mathbf{w}||_r^r < b$ almost surely for some $b > 0$.*

Assumption 4 actually requires that $\eta_1$ grows much slower than $\eta_2$. We can justify it by the following example.

**Example.** *We can consider an RBM with Bernoulli hidden units. In this case, we have $\eta_1(x) = -\frac{1}{\alpha} \log \cosh(\sqrt{\alpha}x)$ and $\eta_2(w) = w^2$. As $\eta_1(x)$ grows almost linearly for large $x$, for any $\varepsilon > 0$, we have*

$$\frac{1}{d} \sum_{\mu=1}^n \eta_1 \left( \frac{1}{\sqrt{n}} \mathbf{x}_\mu^\top \mathbf{w} \right) \geq -\frac{1}{d} \varepsilon \left\| \frac{1}{\sqrt{n}} \mathbf{X} \mathbf{w} \right\|^2 \geq -\frac{1}{d} \epsilon (\lambda_{max} + \lambda) ||\mathbf{w}||^2 \tag{90}$$

*for $\frac{1}{d}||\mathbf{w}||^2 > M$ and some $M > 0$, where $\lambda_{max}$ represents the largest eigenvalue of $\frac{1}{n} \mathbf{Z}^\top \mathbf{Z}$. By the standard concentration results for Wishart matrices (see e.g.,[Xu et al., 2025, Proposition A.1]), we have $\lambda_{max} < M'$ almost surely for some $M' > 0$ and all $d$. Therefore, we almost surely have*

$$\mathcal{L}(\mathbf{w}) \geq \frac{1}{d}(1 - \epsilon M')||\mathbf{w}||^2 > 0 \tag{91}$$

*for $\frac{1}{d}||\mathbf{w}||^2 > M$ by choosing $\epsilon < \frac{1}{M'}$. As $\mathcal{L}(0) = 0$, Assumption 4 holds for $b := M$.*

Our main result, Theorem 7, provides an exact description of the global minimum of (89). Before doing so, we need to introduce some notation. We first define the potential function

$$\mathcal{E}(m, q, p, \tau, \kappa, \nu, \chi, \phi) := \frac{\kappa\tau}{2} + \alpha \mathbb{E} \mathcal{M}_{\frac{\tau}{\kappa} \eta_1} (\lambda m U^* + qG) + \mathbb{E} \mathcal{M}_{\frac{1}{\chi}(\eta_2(\cdot, p) + \phi(\cdot)^r)} \left( \frac{1}{\chi}(\nu W^* + \kappa H) \right)$$
$$- \frac{1}{2\chi}(\nu^2 \rho + \kappa^2) + \nu m - \frac{1}{2}\chi q^2 - \phi p, \tag{92}$$

where the expectation is w.r.t. independent random variables $G, H \sim \mathcal{N}(0,1), U^* \sim P_u, W^* \sim P_w$ and we define $\rho := \mathbb{E}[(W^*)^2]$. We also use the following notation to represent the Moreau envelope

$$\mathcal{M}_{\tau f}(x) = \inf_{y \in \mathbb{R}} \left[ f(y) + \frac{1}{2\tau}(y - x)^2 \right] \tag{93}$$

for some function $f$ and scalar $\tau$. We define the corresponding minimizer as

$$\mathcal{P}_{\tau f}(x) \in \arg \inf_{y \in \mathbb{R}} \left[ f(y) + \frac{1}{2\tau}(y - x)^2 \right]. \tag{94}$$

Finally, we define the set

$$S := \{\hat{m}, \hat{q}, \hat{p}, \hat{\tau}, \hat{\kappa}, \hat{\nu}, \hat{\chi}, \hat{\phi} : \mathcal{E}(\hat{m}, \hat{q}, \hat{p}, \hat{\tau}, \hat{\kappa}, \hat{\nu}, \hat{\chi}, \hat{\phi}) = \sup_{\kappa, \nu, \chi, \phi} \inf_{p, q \leq b, m, \tau} \mathcal{E}(m, q, p, \tau, \kappa, \nu, \chi, \phi)\}. \tag{95}$$

There are two additional assumptions.

**Assumption 5.** *$P_u, P_w$ are sub-Gaussian. $\eta_1$ is $PL(2)$. $\eta_2$ is $PL(2)$ w.r.t. its first variable and continuous w.r.t. its second variable. Moreover, there exists $(\hat{m}, \hat{q}, \hat{p}, \hat{\tau}, \hat{\kappa}, \hat{\nu}, \hat{\chi}, \hat{\phi}) \in S$ such that we can choose (as the minimizer might not be unique) $\mathcal{P}_{\frac{\hat{\tau}}{\hat{\kappa}} \eta_1}$ and $\mathcal{P}_{\frac{1}{\hat{\chi}}(\eta_2(\cdot, \hat{p}) + \hat{\phi}(\cdot)^r)}$ to be Lipschitz continuous, uniformly at $(\hat{p}, \hat{\tau}, \hat{\kappa}, \hat{\phi})$.*

**Assumption 6** (Replicon Stability). *Under Assumption 5, $(\hat{m}, \hat{q}, \hat{p}, \hat{\tau}, \hat{\kappa}, \hat{\nu}, \hat{\chi}, \hat{\phi})$ also satisfies*

$$\alpha \mathbb{E} \left[ \mathcal{P}'_{\frac{\hat{\tau}}{\hat{\kappa}} \eta_1} \left( \frac{\hat{\nu}}{\hat{\chi}} W^* + \frac{\hat{\kappa}}{\hat{\chi}} H \right)^2 \right] \mathbb{E} \left[ \left( \mathcal{P}'_{\frac{1}{\hat{\chi}}(\eta_2(\cdot, \hat{p}) + \hat{\phi}(\cdot)^r)} (\lambda \hat{m} U^* + \hat{q}G) - 1 \right)^2 \right] < 1, \tag{96}$$

*where the expectation is w.r.t. $W^* \sim P_w, U^* \sim P_u$ and $G, H \sim \mathcal{N}(0,1)$.*

Note that under Assumption 5, the derivatives of the proximal operators are well-defined almost everywhere. While Assumption 5 is mainly technical, Assumption 6 is essential and related to the inherent difficulty of the optimization problem, which needs to be verified numerically. See discussions in Vilucchio et al. [2025]. Now we present our main results in this section, which give sharp asymptotics for the global minimum of (89).

**Theorem 7.** *Under Assumptions 4 to 6,*

*(i) We almost surely have*

$$\lim_{d\to\infty} \mathcal{A}_d = \mathcal{A} := \sup_{\kappa,\nu,\chi,\phi} \inf_{p,q\le b,m,\tau} \mathcal{E}(m,q,p,\tau,\kappa,\nu,\chi,\phi) \tag{97}$$

*(ii) For any $PL(2)$ function $\psi$, if*

$$\Psi(s) := \sup_{\kappa,\nu,\chi,\phi} \inf_{p,q\le b,m,\tau} \left[ \mathcal{E}(m,q,p,\tau,\kappa,\nu,\chi,\phi) \right.$$
$$\left. +s\mathbb{E}\psi\left(W^*,\mathcal{P}_{\frac{1}{\chi}(\eta_2(\cdot,p)+\phi(\cdot)^r)}\left(\frac{1}{\chi}(\nu W^*+\kappa H)\right)\right)\right] \tag{98}$$

*is differentiable at $s = 0$, then we almost surely have*

$$\lim_{d\to\infty} \frac{1}{d}\sum_{i=1}^d \psi\left(w_i^*,w_i\right) = \frac{\mathrm{d}}{\mathrm{d}s}\Psi(s)\Big|_{s=0}, \tag{99}$$

*where $\boldsymbol{w}$ is the global minimum of (89) and the expectation is w.r.t. $W^* \sim P_w$ and $H \sim \mathcal{N}(0,1)$.*

Note that the second part implies that if $\mathcal{E}$ is regular enough such that we can interchange the derivative and the $\sup\inf$, we should have

$$\lim_{d\to\infty} \frac{1}{d}\sum_{i=1}^d \psi\left(w_i^*,w_i\right) = \mathbb{E}\psi\left(W^*,\mathcal{P}_{\frac{1}{\hat\chi}\eta_2}\left(\frac{1}{\hat\chi}(\hat v W^*+\hat\kappa H)\right)\right), \tag{100}$$

which is reminiscent of Theorem 4.

As in [Vilucchio et al., 2025, Section D.1], Theorem 7 results from the combination of the following two lemmas for lower and upper bounds. The lower bound is obtained via Gordon's Gaussian comparison inequality, and the upper bound is obtained via a variant of Algorithm 1.

**Lemma 2.** *Under Assumptions 4 and 5, for any $\varepsilon > 0$,*

$$\mathbb{P}[\liminf_{d\to\infty} \mathcal{A}_d \ge \mathcal{A} - \varepsilon] = 1. \tag{101}$$

**Lemma 3.** *Under Assumptions 4 to 6, there exists an algorithm that outputs $\boldsymbol{w}^t$, which converges uniformly (as in Assumption 2), satisfying*

$$\lim_{t\to\infty}\lim_{d\to\infty} \frac{1}{d}\sum_{i=1}^d \psi\left(w_i^*,w_i^t\right) = \mathbb{E}\psi\left(W^*,\mathcal{P}_{\frac{1}{\hat\chi}\eta_2(\cdot,q)}\left(\frac{1}{\hat\chi}(\hat v W^*+\hat\kappa H)\right)\right) \tag{102}$$

*for any $PL(2)$ function $\psi$ and*

$$\lim_{t\to\infty}\lim_{d\to\infty} \mathcal{L}(\boldsymbol{w}^t) = \mathcal{A}. \tag{103}$$

### D.1 Lower bound: proof of Lemma 2

We first lower bound the objective via Gordon's Gaussian comparison inequality.

**Lemma 4** (Gordon's Gaussian comparison inequality [Gordon, 1988]). *Let $\boldsymbol{Z} \in \mathbb{R}^{n\times d}$ have iid standard Gaussian elements, $\boldsymbol{g} \in \mathbb{R}^n, \boldsymbol{h} \in \mathbb{R}^d$ also have iid standard Gaussian elements. For compact sets $S_u \subset \mathbb{R}^n, S_w \subset \mathbb{R}^d$ and any continuous function $\psi$ on $S_u \times S_w$, define*

$$C(\boldsymbol{Z}) := \min_{\boldsymbol{w}\in S_w} \max_{\boldsymbol{u}\in S_u} \boldsymbol{u}^\top Z\boldsymbol{w} + \psi(\boldsymbol{w},\boldsymbol{u}) \tag{104}$$

*and*

$$C(\boldsymbol{g}, \boldsymbol{h}) := \min_{\boldsymbol{w} \in S_w} \max_{\boldsymbol{u} \in S_u} ||\boldsymbol{w}||_2 \boldsymbol{g}^\top \boldsymbol{u} - ||\boldsymbol{u}||_2 \boldsymbol{h}^\top \boldsymbol{w} + \psi(\boldsymbol{w}, \boldsymbol{u}). \tag{105}$$

*Then for any $c \in \mathbb{R}$, we have*

$$\mathbb{P}[C(\boldsymbol{Z}) \leq c] \leq 2\mathbb{P}[C(\boldsymbol{g}, \boldsymbol{h}) \leq c]. \tag{106}$$

Now we can reformulate the optimization problem towards (104) by introducing Lagrange multipliers w.r.t. $\boldsymbol{y} := \frac{1}{\sqrt{d}} \boldsymbol{X} \boldsymbol{w}$ and $p := \frac{||\boldsymbol{w}||_r^r}{d}$. We have

$$\begin{aligned}
\mathcal{A}_d &:= \inf_{\boldsymbol{w}} \frac{1}{d} \sum_{\mu=1}^{n} \eta_1 \left( \frac{1}{\sqrt{d}} \boldsymbol{x}_\mu^\top \boldsymbol{w} \right) + \frac{1}{d} \sum_{i=1}^{d} \eta_2 \left( w_i, \frac{||\boldsymbol{w}||_r^r}{d} \right) \\
&= \inf_{\boldsymbol{w}, \boldsymbol{y}, p} \sup_{\boldsymbol{f}, \phi} \frac{1}{d} \left[ \boldsymbol{f}^\top \left( \frac{1}{\sqrt{d}} \boldsymbol{Z} \boldsymbol{w} + \frac{\lambda}{d} \boldsymbol{u}^*(\boldsymbol{w}^*)^\top \boldsymbol{w} - \boldsymbol{y} \right) \right] \\
&\quad + \frac{1}{d} \sum_{\mu=1}^{n} \eta_1(y_\mu) + \frac{1}{d} \sum_{i=1}^{d} \eta_2(w_i, p) - \phi \left( p - \frac{||\boldsymbol{w}||_r^r}{d} \right).
\end{aligned} \tag{107}$$

Now we can apply Lemma 4 to obtain $\mathbb{P}[\mathcal{A}_d \leq c] \leq 2\mathbb{P}[\tilde{\mathcal{A}}_d \leq c]$ (we will argue below that under Assumption 4 everything is almost surely bounded, so we can consider optimization over compact sets), where

$$\begin{aligned}
\tilde{\mathcal{A}}_d &:= \inf_{p, \boldsymbol{w}, \boldsymbol{y}} \sup_{\phi, \boldsymbol{f}} \frac{1}{\sqrt{d^3}} (||\boldsymbol{w}|| \boldsymbol{g}^\top \boldsymbol{f} - ||\boldsymbol{f}|| \boldsymbol{h}^\top \boldsymbol{w}) - \frac{1}{d} \boldsymbol{f}^\top \boldsymbol{y} + \frac{\lambda}{d^2} \boldsymbol{f}^\top \boldsymbol{u}^*(\boldsymbol{w}^*)^\top \boldsymbol{w} \\
&\quad + \frac{1}{d} \sum_{\mu=1}^{n} \eta_1(y_\mu) + \frac{1}{d} \sum_{i=1}^{d} \eta_2(w_i, p) - \phi \left( p - \frac{||\boldsymbol{w}||_r^r}{d} \right).
\end{aligned} \tag{108}$$

As the objective is linear in $\boldsymbol{f}$, we can keep $\kappa := ||\boldsymbol{f}||/\sqrt{d}$ fixed and optimize over the direction of $\boldsymbol{f}$, which gives

$$\begin{aligned}
\tilde{\mathcal{A}}_d &:= \inf_{p, \boldsymbol{y}, \boldsymbol{w}} \sup_{\phi, \kappa} -\frac{1}{d} \kappa \boldsymbol{h}^\top \boldsymbol{w} + \frac{1}{\sqrt{d}} \kappa \left\| \frac{\lambda}{d} \boldsymbol{u}^*(\boldsymbol{w}^*)^\top \boldsymbol{w} + \frac{1}{\sqrt{d}} ||\boldsymbol{w}|| \boldsymbol{g} - \boldsymbol{y} \right\| \\
&\quad + \frac{1}{d} \sum_{\mu=1}^{n} \eta_1(y_\mu) + \frac{1}{d} \sum_{i=1}^{d} \eta_2(w_i, p) - \phi \left( p - \frac{||\boldsymbol{w}||_r^r}{d} \right).
\end{aligned} \tag{109}$$

We further introduce Lagrange multipliers w.r.t. $m := \frac{1}{d}(\boldsymbol{w}^*)^\top \boldsymbol{w}$ and $q := \frac{1}{\sqrt{d}} ||\boldsymbol{w}||$ to simplify the objective, which gives

$$\begin{aligned}
\tilde{\mathcal{A}}_d &:= \inf_{m, q, \boldsymbol{y}, \boldsymbol{w}} \sup_{\nu, \eta, \kappa, \phi} -\frac{1}{d} \kappa \boldsymbol{h}^\top \boldsymbol{w} + \frac{1}{\sqrt{d}} \kappa \left\| \lambda m \boldsymbol{u}^* + q \boldsymbol{g} - \boldsymbol{y} \right\| + \frac{1}{d} \sum_{\mu=1}^{n} \eta_1(y_\mu) + \frac{1}{d} \sum_{i=1}^{d} (\eta_2(w_i, q) \\
&\quad + \phi w_i^r) - \phi p + \nu \left( m - \frac{1}{d}(\boldsymbol{w}^*)^\top \boldsymbol{w} \right) - \frac{\chi}{2} \left( q^2 - \frac{1}{d} \boldsymbol{w}^\top \boldsymbol{w} \right).
\end{aligned} \tag{110}$$

Then we interchange $\sup$ and $\inf$ to obtain a lower bound and use the inequality $||\boldsymbol{v}|| \geq \min_\tau \frac{\tau}{2} + \frac{||\boldsymbol{v}||^2}{2\tau}$, which gives

$$
\tilde{\mathcal{A}}_d \geq \sup_{\nu,\chi,\kappa,\phi} \inf_{\boldsymbol{y},\boldsymbol{w},\tau,m,q,p} \frac{\kappa\tau}{2} + \frac{\kappa}{2\tau d}||\lambda m \boldsymbol{u}^* + q\boldsymbol{g} - \boldsymbol{y}||^2 + \frac{1}{d}\sum_{\mu=1}^n \eta_1(y_\mu) + \nu m - \frac{1}{2}\chi q^2
$$

$$
+ \frac{1}{d}\left(\frac{\chi}{2}\boldsymbol{w}^\top \boldsymbol{w} - \nu(\boldsymbol{w}^*)^\top \boldsymbol{w} - \kappa \boldsymbol{h}^\top\right) + \frac{1}{d}\sum_{i=1}^d (\eta_2(w_i,p) + \phi w_i^r) - \phi p
$$

$$
= \sup_{\nu,\chi,\kappa,\phi} \inf_{\tau,m,q,p} \frac{\kappa\tau}{2} + \frac{1}{d}\sum_{\mu=1}^n \mathcal{M}_{\frac{\tau}{\kappa}\eta_1}(\lambda m u_\mu^* + q g_\mu) + \frac{1}{d}\sum_{i=1}^d \mathcal{M}_{\frac{1}{\chi}(\eta_2(\cdot,p)+\phi(\cdot)^r)}\left(\frac{1}{\chi}(\nu w_i^* + \kappa h_i)\right)
$$

$$
- \frac{1}{2d\chi}||\nu \boldsymbol{w}^* + \kappa \boldsymbol{h}||^2 + \nu m - \frac{1}{2}\chi q^2 - \phi p,
$$

(111)

where we use the definition of the Moreau envelope. We denote the right side as $\sup_{\nu,\chi,\kappa,\phi} \inf_{\tau,m,q,p} \mathcal{E}_d(m,q,p,\tau,\kappa,\nu,\chi,\phi)$, and thus $\mathcal{E}_d(m,q,p,\tau,\kappa,\nu,\chi,\phi) \to \mathcal{E}(m,q,p,\tau,\kappa,\nu,\chi,\phi)$ almost surely, which follows from the law of large numbers because the Moreau envelopes are pseudo-Lipschitz of the same order as $\eta_1$ and $\eta_2$ [Vilucchio et al., 2025, Lemma 5].

Now we are going to show that the sequence of minimizers of $\{\mathcal{E}_d\}_{d=1}^\infty$ is almost surely bounded in order to interchange $\sup\inf$ and the limit. By Assumption 4, we have that $p, q < b$ is almost surely bounded. By the Cauchy-Schwarz inequality, $m \leq \frac{1}{2}(q + \rho)$ is almost surely bounded. By the arguments after Assumption 4, we have that $\frac{1}{d}||\boldsymbol{y}||^2$ is almost surely bounded. By (107), we have that $\boldsymbol{f} = \sum_{\mu=1}^n \eta_1'(y_\mu)$, and thus $\frac{1}{d}||\boldsymbol{f}||^2 \leq \frac{1}{d}\sum_{\mu=1}^n L_{\eta_1}(1 + |y_\mu|)$ is almost surely bounded, where $L_{\eta_1}$ is the pseudo-Lipschitz constant of $\eta_1$. Thus $\kappa$ is almost surely bounded.

Next, because

$$
\frac{1}{d}\sum_{i=1}^d \mathcal{M}_{\frac{1}{\chi}\eta_2(\cdot,q)}\left(\frac{1}{\chi}(\nu w_i^* + \kappa h_i)\right) \leq \eta_2(0,q) + \frac{1}{2d\chi}||\nu \boldsymbol{w}^* + \kappa \boldsymbol{h}||^2,
$$

(112)

$\mathcal{E}_d$ goes to $-\infty$ for $m = \tau = 0, \chi, \nu, \phi \to \infty$, uniformly in $d$, so $\chi$, $\nu$ and $\phi$ are almost surely bounded. Finally, as

$$
\lim_{\tau \to \infty} \frac{1}{d}\sum_{\mu=1}^n \mathcal{M}_{\frac{\tau}{\kappa}\eta_1}(\lambda m u_\mu^* + q g_\mu) = \inf_{\boldsymbol{y}} \frac{1}{d}\sum_{\mu=1}^n \eta_1(y_\mu) \geq \inf_{\boldsymbol{y}} \frac{1}{d}\sum_{\mu=1}^n L_{\eta_1}(1 + |y_\mu|)|y_\mu| + \eta_1(0) \quad (113)
$$

is almost surely bounded (as $\frac{1}{d}||\boldsymbol{y}||^2$ is almost surely bounded), we have that $\tau$ is almost surely bounded. Thus we can assume that we optimize over $(m,q,p,\tau,\kappa,\nu,\chi,\phi) \in \hat{S}$, where $\hat{S}$ is a compact set.

Moreover, by the exponential concentration of sub-Gaussian variables $\boldsymbol{u}^*, \boldsymbol{w}^*$ and Gaussian variables $\boldsymbol{g}, \boldsymbol{h}$, we have that $\{\frac{1}{d}||\boldsymbol{u}^*||^2, \frac{1}{d}||\boldsymbol{w}^*||^2, \frac{1}{d}||\boldsymbol{g}||^2, \frac{1}{d}||\boldsymbol{h}||^2\}_{d=1}^\infty$ is almost surely bounded by the Borel-Cantelli Lemma. Therefore, by the pseudo-Lipschitz property of the Moreau envolopes, we have that $\mathcal{E}_d(m,q,p,\tau,\kappa,\nu,\chi,\phi)$ is almost surely equicontinuous in $\hat{S}$ (according to the Arzelà-Ascoli theorem), and thus it converges to $\mathcal{E}(m,q,p,\tau,\kappa,\nu,\chi,\phi)$ almost surely uniformly in $\hat{S}$, which gives

$$
\lim_{d \to \infty} \sup_{\nu,\chi,\kappa,\phi} \inf_{\tau,m,q,p} \mathcal{E}_d(m,q,p,\tau,\kappa,\nu,\chi,\phi) = \sup_{\nu,\chi,\kappa,\phi} \inf_{\tau,m,q,p} \mathcal{E}(m,q,p,\tau,\kappa,\nu,\chi,\phi) \quad (114)
$$

almost surely. We finish the proof of Lemma 2 by using $\tilde{\mathcal{A}}_d \geq \sup_{\nu,\chi,\kappa,\phi} \inf_{\tau,m,q,p} \mathcal{E}_d(m,q,p,\tau,\kappa,\nu,\chi,\phi)$ and Lemma 4.

### D.2 Upper bound: proof of Lemma 3

We will show that the following variant of Algorithm 1 can give the upper bound.

$$
\boldsymbol{y}^t = \frac{1}{\sqrt{n}}\boldsymbol{X}\boldsymbol{w}^t - b_t \boldsymbol{u}^{t-1} \in \mathbb{R}^n, \ \boldsymbol{u}^t = f^*(\boldsymbol{y}^t) \in \mathbb{R}^n,
$$

$$
\boldsymbol{z}^t = \frac{1}{\sqrt{n}}\boldsymbol{X}^\top \boldsymbol{u}^t - c_t \boldsymbol{w}^{t-1} \in \mathbb{R}^d, \ \boldsymbol{w}^{t+1} = g^*(\boldsymbol{z}^t) \in \mathbb{R}^d,
$$

(115)

with Onsager coefficients given by

$$b_{t+1} = \frac{1}{n}\sum_{i=1}^{d}(f^*)'(y_i^t), \ c_{t+1} = \frac{1}{n}\sum_{\mu=1}^{n}(g^*)'(z_\mu^t) \tag{116}$$

and denoisers given by

$$f^*(x) := \frac{\sqrt{\alpha}\hat{\kappa}}{\hat{\tau}\hat{\chi}}(\mathcal{P}_{\frac{\hat{\tau}}{\hat{\kappa}}\eta_1}(x) - x), \ g^*(x) = \sqrt{\alpha}\mathcal{P}_{\frac{1}{\hat{\chi}}(\eta_2(\cdot,\hat{p})+\hat{\phi}(\cdot)^r)}(x). \tag{117}$$

It is essentially Algorithm 1 at the fixed point. Lemma 3 thus follows from the following lemma.

**Lemma 5.** *If we run* (115) *with initialization* $\boldsymbol{w}^1 = g^*(\frac{\hat{\nu}}{\hat{\chi}}\boldsymbol{w}^* + \frac{\hat{\kappa}}{\hat{\chi}}\boldsymbol{h})$ *and* $b_1 = 0$, *where* $\boldsymbol{h}$ *is an iid standard Gaussian vector, then* $\boldsymbol{w}^t/\sqrt{\alpha}$ *satisfies Lemma 3.*

*Proof.* By Theorem 6, we can write the SE of (115) as

$$\begin{aligned}
(Y_1, \cdots, Y_t) &= (M_1, \cdots, M_t)U^* + \mathcal{N}(0, \boldsymbol{\Sigma}_t),\\
(Z_1, \cdots, Z_t) &= (N_1, \cdots, N_t)W^* + \mathcal{N}(0, \mathring{\boldsymbol{\Omega}}_t),\\
U_t &= f^*(Y_t), \ W_t = g^*(Z_{t-1}),
\end{aligned} \tag{118}$$

where

$$\begin{aligned}
M_t &= \sqrt{\alpha^{-1}}\lambda\mathbb{E}[W_tW^*], \ [\boldsymbol{\Sigma}_t]_{ij} = \alpha^{-1}\mathbb{E}[W_iW_j],\\
N_t &= \sqrt{\alpha}\lambda\mathbb{E}[U_tU^*], \ [\boldsymbol{\Omega}_t]_{ij} = \mathbb{E}[U_iU_j].
\end{aligned} \tag{119}$$

Equivalently we write it as

$$\begin{aligned}
M_t &= \frac{1}{\sqrt{\alpha}}\lambda\mathbb{E}[g^*(N_{t-1}W^* + \sqrt{[\boldsymbol{\Omega}_{t-1}]_{t-1,t-1}}H)W^*],\\
[\boldsymbol{\Sigma}_t]_{tt} &= \alpha^{-1}\mathbb{E}[(g^*(N_{t-1}W^* + \sqrt{[\boldsymbol{\Omega}_{t-1}]_{t-1,t-1}}H))^2],\\
N_t &= \sqrt{\alpha}\lambda\mathbb{E}[f^*(M_tU^* + \sqrt{[\boldsymbol{\Sigma}_t]_{tt}}U^*)], \ [\boldsymbol{\Omega}_t]_{tt} = \mathbb{E}[(f^*(M_tU^* + \sqrt{[\boldsymbol{\Sigma}_t]_{tt}}))^2].
\end{aligned} \tag{120}$$

By comparing it with the stationary points of (92)

$$\begin{aligned}
\partial_\nu : \ \hat{m} &= \mathbb{E}\left[W^*\mathcal{P}_{\frac{1}{\hat{\chi}}(\eta_2(\cdot,\hat{p})+\hat{\phi}(\cdot)^r)}\left(\frac{1}{\hat{\chi}}(\hat{\nu}W^* + \hat{\kappa}H)\right)\right],\\
\partial_\chi : \ \hat{q}^2 &= \mathbb{E}\left[\mathcal{P}_{\frac{1}{\hat{\chi}}(\eta_2(\cdot,\hat{p})+\hat{\phi}(\cdot)^r)}\left(\frac{1}{\hat{\chi}}(\hat{\nu}W^* + \hat{\kappa}H)\right)^2\right]\\
\partial_m : \ \hat{\nu} &= \lambda\alpha\frac{\hat{\kappa}}{\hat{\tau}}\mathbb{E}\left[U^*\left(\mathcal{P}_{\frac{\hat{\tau}}{\hat{\kappa}}\eta_1}(\lambda\hat{m}U^* + \hat{q}G) - \lambda\hat{m}U^* - \hat{q}G\right)\right],\\
\partial_\tau : \ \hat{\kappa}^2 &= \alpha\frac{\hat{\kappa}^2}{\hat{\tau}^2}\mathbb{E}\left[\left(\mathcal{P}_{\frac{\hat{\tau}}{\hat{\kappa}}\eta_1}(\lambda\hat{m}U^* + \hat{q}G) - \lambda\hat{m}U^* - \hat{q}G\right)^2\right]
\end{aligned} \tag{121}$$

and the definition of $f^*, g^*$, we have

$$M_t = \lambda\hat{m}, [\boldsymbol{\Sigma}_t]_{tt} = \hat{q}^2, N_t = \frac{\hat{\nu}}{\hat{\chi}}, [\boldsymbol{\Omega}_t]_{tt} = \frac{\hat{\kappa}^2}{\hat{\chi}^2} \tag{122}$$

for any $t \geq 1$. Other three stationary point equations of (92) will be useful later

$$\begin{aligned}
\partial_\kappa : \ \hat{\tau} &= \mathbb{E}\left[H\mathcal{P}_{\frac{1}{\hat{\chi}}(\eta_2(\cdot,\hat{p})+\phi(\cdot)^r)}\left(\frac{1}{\hat{\chi}}(\hat{\nu}W^* + \hat{\kappa}H)\right)\right],\\
\partial_q : \ \hat{\chi}\hat{q} &= \alpha\frac{\hat{\kappa}}{\hat{\tau}}\mathbb{E}\left[G\left(\mathcal{P}_{\frac{\hat{\tau}}{\hat{\kappa}}\eta_1}(\lambda\hat{m}U^* + \hat{q}G) - \lambda\hat{m}U^* - \hat{q}G\right)\right],\\
\partial_\phi : \ \hat{p} &= \mathbb{E}\left[\left(\mathcal{P}_{\frac{1}{\hat{\chi}}(\eta_2(\cdot,\hat{p})+\phi(\cdot)^r)}\left(\frac{1}{\hat{\chi}}(\hat{\nu}W^* + \hat{\kappa}H)\right)\right)^r\right].
\end{aligned} \tag{123}$$

To obtain these stationary point equations, we use the identities

$$\frac{\partial \mathcal{M}_{\tau f}(z)}{\partial z} = \frac{z - \mathcal{P}_{\tau f}(z)}{\tau}, \ \frac{\partial \mathcal{M}_{\tau f}(z)}{\partial \tau} = -\frac{(z - \mathcal{P}_{\tau f}(z))^2}{2\tau^2}, \ \frac{\partial \mathcal{M}_{\tau(f+\phi g)}(z)}{\partial \phi} = g(\mathcal{P}_{\tau(f+\phi g)}(z))$$

$$(124)$$

and interchange the derivative and expectation under Assumption 5. With a slight abuse of notation, we denote $M := M_t = \lambda \hat{m}$, $N := N_t = \frac{\hat{\nu}}{\hat{\chi}}$, $\Sigma := [\boldsymbol{\Sigma}_t]_{tt} = \hat{q}^2$ and $\Omega := [\boldsymbol{\Omega}_t]_{tt} = \frac{\hat{\kappa}^2}{\hat{\chi}^2}$. We further denote

$$\lim_{d \to \infty} \frac{1}{n}||\boldsymbol{y}^t - \boldsymbol{y}^{t-1}||^2 := 2(\Sigma - C_t), \ \lim_{d \to \infty} \frac{1}{d}||\boldsymbol{z}^t - \boldsymbol{z}^{t-1}||^2 := 2(\Sigma - \hat{C}_t). \tag{125}$$

Then by (118), we have

$$C_t = [\boldsymbol{\Sigma}_t]_{t,t-1}$$

$$= \alpha^{-1}\mathbb{E}[g^*(NW^* + \sqrt{\hat{C}_{t-1}}Z_1 + \sqrt{\Sigma - \hat{C}_{t-1}}Z_1')g^*(NW^* + \sqrt{\hat{C}_{t-1}}Z_1 + \sqrt{\Sigma - \hat{C}_{t-1}}Z_1'')],$$

$$\hat{C}_t = [\boldsymbol{\Omega}_t]_{t,t-1} = \mathbb{E}[f^*(MU^* + \sqrt{C_t}Z_2 + \sqrt{\Omega - C_t}Z_2')f^*(MU^* + \sqrt{C_t}Z_2 + \sqrt{\Omega - C_t}Z_2'')],$$

$$(126)$$

where $Z_1, Z_1', Z_1'', Z_2, Z_2', Z_2''$ are iid standard Gaussian variable. By the same argument as [Vilucchio et al., 2025, Lemma 3], we have

$$\partial_{\hat{C}_{t-1}} C_t \leq \alpha^{-1}\mathbb{E}[(g^*)'(NW^* + \sqrt{\Sigma}Z_1)^2], \ \partial_{C_t} \hat{C}_t \leq \mathbb{E}[(f^*)'(MU^* + \sqrt{\Omega}Z_2)^2]. \tag{127}$$

Therefore, under Assumption 6, the iteration (126) globally converges to its fixed point, i.e., $\lim_{t\to\infty} \hat{C}_t = \Sigma$, $\lim_{t\to\infty} C_t = \Omega$, which gives

$$\lim_{t\to\infty}\lim_{d\to\infty} \frac{1}{n}||\boldsymbol{y}^t - \boldsymbol{y}^{t-1}||^2 = 0, \ \lim_{t\to\infty}\lim_{d\to\infty} \frac{1}{d}||\boldsymbol{z}^t - \boldsymbol{z}^{t-1}||^2 = 0. \tag{128}$$

Now we are going to prove that $\lim_{t\to\infty}\lim_{d\to\infty} \mathcal{L}(\boldsymbol{w}^t) = \mathcal{A}$ almost surely. The objective reads

$$\mathcal{L}\left(\frac{\boldsymbol{w}^t}{\sqrt{\alpha}}\right) = \frac{1}{d}\sum_{\mu=1}^n \eta_1\left(y_\mu^t + b_t u_\mu^{t-1}\right) + \frac{1}{d}\sum_{i=1}^d \eta_2\left(\frac{w_i^t}{\sqrt{\alpha}}, \frac{||\boldsymbol{w}^t||_r^r}{\alpha^{r/2}d}\right). \tag{129}$$

By the law of large numbers and the pseudo-Lipschitz property of $\eta_1, \eta_2$, we have

$$\lim_{d\to\infty} \frac{1}{d}\sum_{i=1}^d \eta_2\left(\frac{w_i^t}{\sqrt{\alpha}}, \frac{||\boldsymbol{w}^t||_r^r}{\alpha^{r/2}d}\right) = \mathbb{E}\eta_2\left(\frac{1}{\sqrt{\alpha}}g^*\left(\frac{\hat{\nu}}{\hat{\chi}}W^* + \frac{\hat{\kappa}}{\hat{\chi}}H\right), \left(\frac{1}{\sqrt{\alpha}}g^*\left(\frac{\hat{\nu}}{\hat{\chi}}W^* + \frac{\hat{\kappa}}{\hat{\chi}}H\right)\right)^r\right)$$

$$= \mathbb{E}\eta_2\left(\mathcal{P}_{\frac{1}{\hat{\chi}}(\cdot,\hat{q})}\left(\frac{\hat{\nu}}{\hat{\chi}}W^* + \frac{\hat{\kappa}}{\hat{\chi}}H\right), \hat{p}\right),$$

$$(130)$$

and

$$\lim_{d\to\infty} b_t = \frac{1}{\alpha}\mathbb{E}[(f^*)'(\lambda\hat{m}U^* + \hat{q}G)] = \frac{\hat{\tau}\hat{\chi}}{\alpha\hat{\kappa}\hat{q}}\mathbb{E}[Gf^*(\lambda\hat{m}U^* + \hat{q}G)] = \frac{1}{\sqrt{\alpha}}\frac{\hat{\tau}\hat{\chi}}{\hat{\kappa}} \tag{131}$$

almost surely, where we use the Stein's lemma and (123). Thus

$$\lim_{d\to\infty} \frac{1}{d}\sum_{\mu=1}^n \eta_1\left(y_\mu^t + b_t u_\mu^t\right) = \alpha\mathbb{E}[\eta_1(\lambda\hat{m}U^* + \hat{q}G + \frac{1}{\sqrt{\alpha}}\frac{\tau\eta}{\kappa}f^*(\lambda\hat{m}U^* + \hat{q}G))]$$

$$= \alpha\mathbb{E}[\eta_1(\mathcal{P}_{\frac{\hat{\tau}}{\hat{\kappa}}\eta_1}(\lambda\hat{m}U^* + \hat{q}G))].$$

$$(132)$$

By the pseudo-Lipschitz property of $\eta_1$, we have

$$\frac{1}{d}\sum_{\mu=1}^n |\eta_1\left(y_\mu^t + b_t u_\mu^{t-1}\right) - \eta_1\left(y_\mu^t + b_t u_\mu^t\right)|$$

$$\leq \frac{1}{d}\sum_{\mu=1}^n L_{\eta_1}(1 + 2|y_\mu^t| + |b_t u_\mu|^{t-1} + |b_t u_\mu^t|)(|b_t||u_\mu^t - u_\mu^{tt-1}|) \to 0 \tag{133}$$

for $d \to \infty$ and then $t \to \infty$ by (128). Therefore, we have

$$\lim_{t\to\infty}\lim_{d\to\infty}\mathcal{L}\left(\frac{\boldsymbol{w}^t}{\sqrt{\alpha}}\right) = \alpha\mathbb{E}\eta_1(\mathcal{P}_{\frac{\hat{\tau}}{\hat{\kappa}}\eta_1}(\hat{m}U^* + \hat{q}G)) + \mathbb{E}\eta_2\left(\mathcal{P}_{\frac{1}{\hat{\chi}}(\eta_2(\cdot,\hat{p})+\hat{\phi}(\cdot)^r)}\left(\frac{1}{\hat{\chi}}(\hat{\nu}W^* + \hat{\kappa}H)\right),\hat{p}\right). \tag{134}$$

The optimum of (92) reads

$$\mathcal{A} = \frac{\hat{\kappa}\hat{\tau}}{2} + \alpha\mathbb{E}\eta_1(\mathcal{P}_{\frac{\hat{\tau}}{\hat{\kappa}}\eta_1}(\hat{m}U^* + \hat{q}G)) + \alpha\frac{\hat{\kappa}}{2\hat{\tau}}(\mathcal{P}_{\frac{\hat{\tau}}{\hat{\kappa}}\eta_1}(\hat{m}U^* + \hat{q}G) - (\hat{m}U^* + \hat{q}G))^2$$

$$+ \mathbb{E}\eta_2\left(\mathcal{P}_{\frac{1}{\hat{\chi}}(\eta_2(\cdot,\hat{p})+\hat{\phi}(\cdot)^r)}\left(\frac{1}{\hat{\chi}}(\hat{\nu}W^* + \hat{\kappa}H)\right),\hat{q}\right)$$

$$+ \frac{\hat{\chi}}{2}\left(\mathcal{P}_{\frac{1}{\hat{\chi}}(\eta_2(\cdot,\hat{p})+\hat{\phi}(\cdot)^r)}\left(\frac{1}{\hat{\chi}}(\hat{\nu}W^* + \hat{\kappa}H)\right) - \frac{1}{\hat{\chi}}(\hat{\nu}W^* + \hat{\kappa}H)\right)^2 - \frac{1}{2\hat{\chi}}(\hat{\nu}^2\rho + \hat{\kappa}^2) + \hat{\nu}\hat{m} - \frac{1}{2}\hat{\chi}\hat{q}, \tag{135}$$

where we use the identity

$$\mathcal{M}_{\tau f(\cdot)}(x) = f(\mathcal{P}_{\tau f(\cdot)}(x)) + \frac{1}{2\tau}(\mathcal{P}_{\tau f(\cdot)}(x) - x)^2. \tag{136}$$

By using (121) and (123), we can obtain

$$\mathcal{A} = \alpha\mathbb{E}\eta_1(\mathcal{P}_{\frac{\hat{\tau}}{\hat{\kappa}}\eta_1}(\hat{m}U^* + \hat{q}G)) + \mathbb{E}\eta_2\left(\mathcal{P}_{\frac{1}{\hat{\chi}}(\eta_2(\cdot,\hat{p})+\hat{\phi}(\cdot)^r)}\left(\frac{1}{\hat{\chi}}(\hat{\nu}W^* + \hat{\kappa}H)\right),\hat{p}\right). \tag{137}$$

We finish the proof by combining it with (134). $\qquad\square$

# E   Experiments on Fashion MNIST

We verify Theorem 1 on Fashion MNIST in Figure 3. We choose Bernoulli hidden units and Rademacher visible units with trainable biases to satisfy Assumption 1. We choose to use 10 hidden units to exactly compute (7). For GD-equiv (GD over the simplified loss (5)), we obtain the weights of RBMs by using the Adam optimizer to maximize (5). To train RBMs with contrastive divergence (CD), we follow the standard implementation [Hinton, 2012]. Specifically, we update the weights through

$$w_{ai}^{t+1} = w_{ai}^t + \frac{1}{\sqrt{d}}\kappa(\langle x_i h_a\rangle_D - \langle x_i h_a\rangle_H), \tag{138}$$

where $\kappa$ is the learning rate,

$$\langle x_i h_a\rangle_D := \frac{1}{n}\sum_{\mu}x_{\mu i}h_{\mu a} \tag{139}$$

and $\langle x_i h_a\rangle_H$ is obtained through 1 cycle MCMC starting from the data. For both GD-equiv and CD, we choose batchsize 50 and run for 20 epochs on the training set. The learning rate is chosen as $\kappa = 0.1\sqrt{d}$. The initial weights are Gaussian. We present the reconstruction performance of 10 images randomly chosen from the test set, where we occlude half of the images or add random salt-and-pepper noise.

We compare how RBMs with weights obtained by GD-equiv and CD reconstruct half-occluded images and noisy images in Figure 3. Specifically, the RBM first samples the hidden units based on the occluded or noisy image and then samples the visible units to generate the reconstructed image. To better reconstruct occluded images, RBM takes the reconstructed image to replace the occluded part and iterates several times. Figure 3 suggests that RBMs trained with GD on the equivalent objective (5) can reconstruct the original images as well as standard RBMs. The small difference in the quality of the reconstruction should be attributed to the approximation error.

# F   Experiments on synthetic data

## F.1   Comparison between different algorithms

For the experiments on the spiked covariance model in Section 5, we set $d = 4000$, $n = 8000$ and report the results and standard errors averaged over 10 runs. We choose $P_u$ and $P_w$ to be $P^{\otimes r}$, where

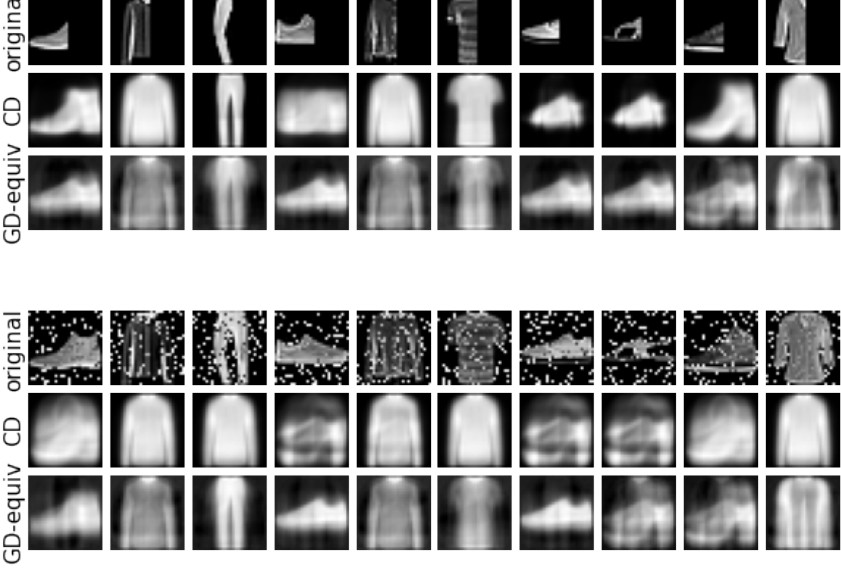

**Figure 3:** Reconstruction of occluded (**Upper**) and noisy (**Lower**) Fashion MNIST figures by RBMs trained with CD and GD. The RBMs have 10 hidden units.

$P(x) := \frac{1}{2}(\delta(x+1) + \delta(x-1))$ is the Rademacher distribution. In this case, we have

$$\eta_1(x) = \log \cosh(\sqrt{\alpha}x), \tag{140}$$

and $g(\boldsymbol{y}^t, \boldsymbol{B}_t)$ is given by the solution of

$$\frac{1}{\sqrt{\alpha}} \tanh(\sqrt{\alpha}(\boldsymbol{B}_t\boldsymbol{g} + \boldsymbol{y}^t)) + \boldsymbol{g} = 0. \tag{141}$$

We measure the overlap $\zeta_{ij} := \frac{|\boldsymbol{w}_i^\top(\boldsymbol{w}_j^*)|}{||\boldsymbol{w}_i||||\boldsymbol{w}_j^*||} \in [0, 1]$ between the $j$-th signal $\boldsymbol{w}_j^* \in \mathbb{R}^d$ and the weight vector $\boldsymbol{w}_i \in \mathbb{R}^d$ of each unit for standard RBMs. We also measure the Bayes optimal overlap [Lesieur et al., 2017], and the overlap obtained by AMP-RBM (Algorithm 1), GD (25). For both AMP-RBM and GD, (7) is calculated exactly by averaging over all $2^{10}$ possibilities. The overlap can be predicted by the state evolution via

$$\hat{\zeta}_{ij} = \frac{\mathbb{E}[[f(\boldsymbol{M}_\infty\boldsymbol{W} + \boldsymbol{\Sigma}_\infty^{1/2}\boldsymbol{G}, \bar{\hat{\boldsymbol{Q}}}_\infty, \bar{\boldsymbol{C}}_\infty)]_i W_j]}{\sqrt{\mathbb{E}[[f(\boldsymbol{M}_\infty\boldsymbol{W} + \boldsymbol{\Sigma}_\infty^{1/2}\boldsymbol{G}, \bar{\hat{\boldsymbol{Q}}}_\infty, \bar{\boldsymbol{C}}_\infty)]_i^2]\mathbb{E}[W_j^2]}}. \tag{142}$$

In this section, we also compare our algorithms with RBMs trained by CD, where both the hidden and visible units are chosen to be Rademacher units without biases. The iteration is provided in Figure 4, which is qualitatively similar to that of GD in Figure 1, but CD converges much slower. In this section we consider the degenerate case $\Lambda = \lambda I$ to better compare different algorithms, because it is generally harder and SVD cannot distinguish signals in the degenerate subspace.

For the iteration curves in the left side of Figure 1 and Figure 4, we use the informed initialization to better demonstrate the state evolution. For other figures, we only consider the random initialization (i.e., $\boldsymbol{M}_0 = 0$) for more realistic comparison, and we will use the Adam optimizer to optimize (5) to obtain better convergence.

The comparison between standard RBMs (trained with CD) and other methods for $k > 1$ is given in Figure 5. The left panel of Figure 5 considers the $r = k = 2$ case, where the y axis corresponds to the average overlap $\zeta := \frac{1}{2}(\max(\zeta_{11}, \zeta_{21}) + \max(\zeta_{12}, \zeta_{22}))$. In this case, the SVD can only

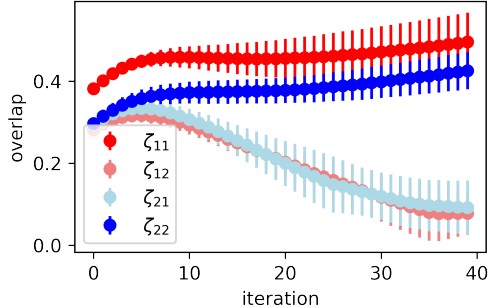

**Figure 4:** Iteration of CD. The setting is the same as Figure 1.

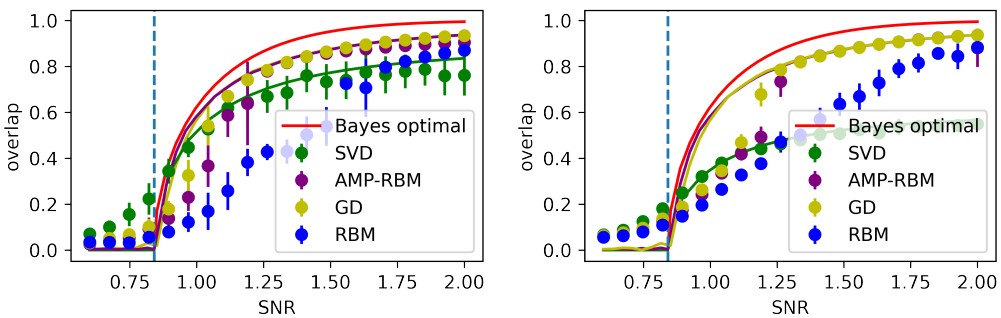

**Figure 5:** Comparision between standard RBMs (trained with CD) and other methods for $r = k = 2$ (**Left**), $r = k = 10$ (**Right**), Rademacher prior. The red line represents the overlap of the Bayes optimal estimation. The purple lines represent the asymptotics of stationary points (Theorem 4). The yellow lines represent the asymptotics of the fixed points of DMFT (Theorem 5), which overlap with the purple lines. Note that the red, yellow and purple lines are all for rank-1. The green lines represent the asymptotics of SVD. Dashed blue lines represent the BBP threshold.

find the two-dimensional subspace but not the signal. The results of SVD converge to a uniform distribution over the subspace [Montanari and Venkataramanan, 2021], which gives the green curve $\zeta = \max(\frac{2\sqrt{2}}{\pi} \frac{1-\alpha/\lambda^4}{1+\alpha/\lambda^2}, 0)$. The right panel of Figure 5 considers the $r = k = 10$ case, and the green line is obtained by numerically averaging over a ten-dimensional Haar distribution. AMP-RBM might fail to converge for $k = 10$ due to numerical difficulties in solving (17), and we filter these failed iterations with negative objective values. The red, purple, and yellow curves are asymptotic rank-1 predictions. As noted after Corollary 1, we can assume the overlap matrix to be diagonal to find the convergent point, so the rank-1 asymptotics actually locate one of the possible saddle points, though other saddle points may also exist.

In Figure 5, AMP-RBM and GD show significant advantages over SVD especially for rank-10, because SVD cannot distinguish the signals. AMP-RBM and GD perform similarly to the rank-1 case, as predicted by Corollary 1. The right side of Figure 5 considers the $r = k = 10$ case, where the gap between SVD and Bayes optimality is larger, while the performances of AMP-RBM and GD are similar for large SNRs. For small SNRs, however, there is a larger gap between rank-$r$ AMP-RBM/GD and rank-1 AMP-RBM/GD around $\lambda = 1$ when $r$ gets larger, suggesting that the landscape becomes more complex, and GD converges to stationary points different than the one specified by Corollary 1. The gap between AMP-RBM/GD and RBM also gets larger for small noise and large rank, possibly due to the error in Monte Carlo sampling.

Finally, we note that the Bayes optimal overlap is in terms of the rank-1 case, and thus there is a larger gap between the Bayes optimal overlap and other algorithms for $r = k = 10$. In fact this gap also exists for various kinds of AMP algorithms [Lesieur et al., 2017, Montanari and Venkataramanan, 2021, Pourkamali et al., 2024]. We present the performances of different AMP algorithms in Figure 6, analogously to Figure 5. AMP denotes the algorithm in Lesieur et al. [2017] and decimation AMP denotes the algorithm in Pourkamali et al. [2024], which is conjectured to be optimal in terms of

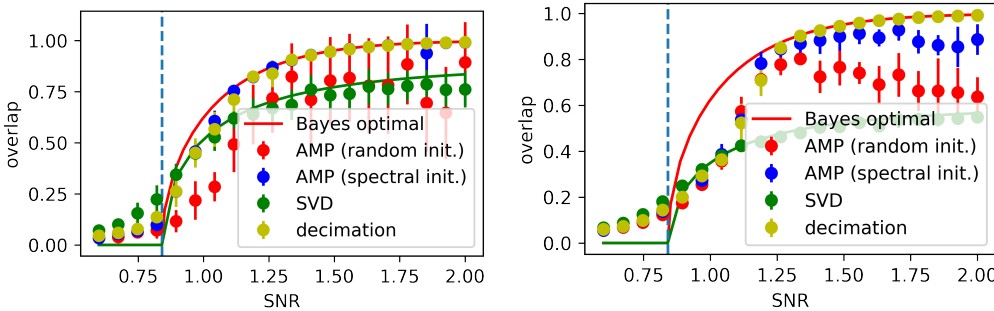

**Figure 6:** The performance of AMP and decimation AMP. **Left**: $r = k = 2$. **Right**: $r = k = 10$.

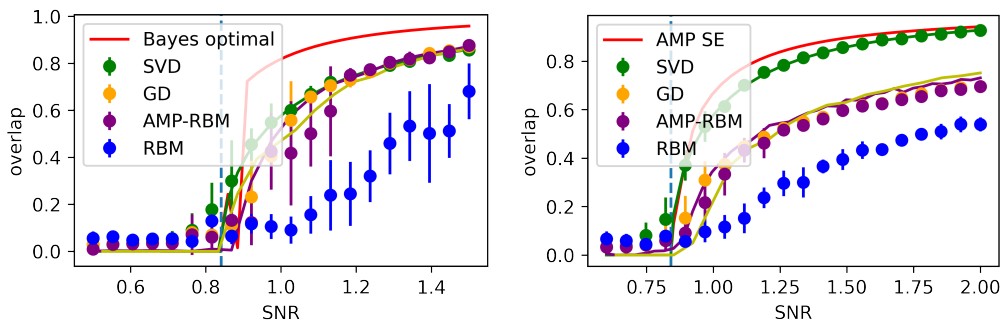

**Figure 7:** Comparision between the standard RBM (trained by CD) and other methods for $r = k = 1$. **Left**: Rademacher hidden units and Bernoulli-Gaussian weights. **Right**: Bernoulli-Rademacher hidden units and Gaussian weights. The lines represent the same thing as those in Figure 5.

estimating $\boldsymbol{U}^*\boldsymbol{\Lambda}(\boldsymbol{W}^*)^\top$ (rather than estimaing $\boldsymbol{W}^*$). Besides random initialization, we also try to initialize AMP with leading singular vectors (spectral initialization). We note that the Bayes optimal AMP algorithm in Montanari and Venkataramanan [2021] cannot be extended to the degenerate case. Figure 6 suggests that the performance of AMP (with random initialization) has a high variance, because it might fail to recover all signals together. On average, its performance is similar to SVD. Moreover, AMP with spectral initialization and the decimation AMP [Pourkamali et al., 2024] can reach the rank-1 Bayes optimal overlap for large SNRs but not for small SNRs, similar to Figure 1, which suggests that it might be fundamentally difficult to distinguish different signals for small SNR. A similar phenomenon is found in previous physics-based analysis of RBMs [Hou et al., 2019, Thériault et al., 2024], where it is called permutation symmetric breaking. This suggests that we are able to perfectly distinguish different signals only for high SNRs.

In Figure 7, we present some experiments with sparse priors. For both cases, the sparsity is $0.9$. Figure 7 suggests that GD and AMP-RBM still converge well to its SE, but RBM does not.

### F.2 Learning dynamics of RBMs

We first consider the case where different signals have different SNRs. The results of Li et al. [2023] suggest that from random initialization, AMP weakly recovers a signal of SNR $\lambda$ at time $O\left(\frac{\log d}{\lambda}\right)$ (for $\alpha$ sufficiently small). Therefore, we can expect that RBMs recover these signals sequentially, which leads to a series of emergence phenomenon, as demonstrated on the left side of Figure 8. Note that this reproduces the results in Bachtis et al. [2024], which is called a "cascade of phase transitions". If we further assume that the $r-$th signal has an SNR $\lambda_r = \lambda^{-r}$, we expect that at the iteration $t = \lambda^r \log d$, $r$ signals are recovered sufficiently and other signals are not weakly recovered. This gives a power-law scaling

$$L := \sum_{k=r+1}^{\infty} \lambda^{-k} = \frac{1}{\lambda - 1}\lambda^{-r} = \frac{\log d}{\lambda - 1}t^{-1}, \tag{143}$$

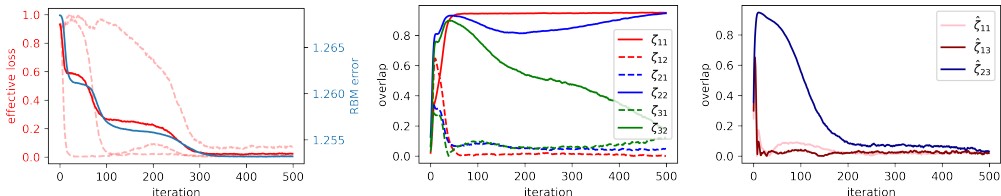

**Figure 8: Left**: RBMs (trained by CD) learn three signals sequentially, leading to a stairwise loss ($\lambda = 2, 2.5, 3$, $\alpha = 2$). The solid red line denotes the effective loss $\sum_{i=1}^{3} \lambda_i \min_j ||\boldsymbol{w}_j - \boldsymbol{w}_i^*||^2$, and dashed red lines denote the effective loss $\min_j ||\boldsymbol{w}_j - \boldsymbol{w}_i^*||^2$ for each signal, where we normalize them to $[0, 1]$ for better visualization. The blue line denotes the error of RBMs (trained with CD), which does not go to zero because we are considering a model with strong noise. **Middle and Right**: overlap between weights and signals, and overlap between weights, for an RBM with $r = 2$, $k = 3$ (overparameterization) and $\lambda = 3$. The weights of the RBM align well with the signals, and remain orthogonal to each other.

where the effective loss $L$ evaluates the total strength of the unrecovered signals.

The case where some signals share the same SNR is more challenging, as the naive SVD cannot distinguish these signals. In Figures 5 and 8, we test the performance of RBMs in this setting, where we find that the stationary points indicated by Corollary 1 are dominant for random initialization when the SNR is large enough. To see the dynamics, we plot the iteration curve of standard RBMs (trained by CD) under the random initialization in the middle and right part of Figure 8 for $r = 2, k = 3$, which shows that RBMs could learn all components of the signal effectively. Moreover, during iteration, different hidden units gradually get aligned with the signal and disaligned with each other.

To understand this effect and understand how the alignment changes during iteration, we can observe that $\boldsymbol{w}^\top \hat{\boldsymbol{Q}} \boldsymbol{w}$ in (16) works as an effective regularization term that makes the weights of different hidden units more disaligned with each other. In fact, suppose that $P_h$ is separable and symmetric, we have

$$\nabla \eta_2(\boldsymbol{Q})_{ij} = \frac{\int \mathrm{d}P_h(h_i)\mathrm{d}P_h(h_j)h_i h_j e^{Q_{ii}h_i^2/2 + Q_{ij}h_i h_j/2 + Q_{jj}h_j^2/2}}{\int \mathrm{d}P_h(h_i)\mathrm{d}P_h(h_j)e^{Q_{ii}h_i^2/2 + Q_{ij}h_i h_j/2 + Q_{jj}h_j^2/2}}, \tag{144}$$

which is positive when $Q_{ij} > 0$ and negative when $Q_{ij} < 0$. Therefore, the regularization term $\boldsymbol{w}^\top \hat{\boldsymbol{Q}} \boldsymbol{w}$ in (16) makes the overlap between $\boldsymbol{w}_i$ and $\boldsymbol{w}_j$ smaller when the overlap is positive, and makes the overlap larger when it is negative, and thus makes $\boldsymbol{w}_i$ and $\boldsymbol{w}_j$ more disaligned. On the other hand, the alignment effect comes from the $\eta_1(\boldsymbol{h})$ term, which encourages the hidden units to align with the signals.

