# OpenReview forum: "Learning with Restricted Boltzmann Machines: Asymptotics of AMP and GD in High Dimensions"
_NeurIPS.cc/2025/Conference — NeurIPS 2025 poster_

### Official Review · Reviewer_qSCD · 2025-06-30

**Clarity:** 2
**Significance:** 3
**Originality:** 2
**Rating:** 5
**Confidence:** 4

**Summary:**

This work considers Restricted Boltzmann Machines (RBMs) in the classic high-dimensional limit of growing input dimension and a growing number of data points (asymptotically proportional) and contributes to the analysis of RBMs in this setting. First, a simplified form of the log-likelihood function is derived in this asymptotic setting (Theorem 1), complemented by corresponding forms of the gradient and some classic further simplifications. This is then used together with the classic Spiked Covariance Model to analyse Approximate Message Passing (AMP) on the RBM (which in turn forms a new approach to train RBMs), in particular, the gradient of the simplified log-likelihood at the AMP fixed point (Theorem 2) and the convergence behaviour of AMP under certain initializations (Theorem 3). Furthermore, again using AMP, an asymptotic characterization of stationary points of the log-likelihood are provided (in Theorem 4) and a result on the convergence behaviour of gradient descent (Theorem 5), which is complemented by numerical experiments on synthetic data and Fashion MNIST. The main results are complemented by generalizations (e.g., a more general AMP algorithm) and a result on the global optimum and the AMP fixed point for a simplified variant of the Spiked Covariance Model under standard assumptions.

**Questions:**

**Question 1** Related to Theorem 1
The following aspects are relevant for *Clarity*, *Quality*, *Significance*, and *Rating*

Q1.1 Lack of precision
For example, Equ (4) can only hold if
$d,n\rightarrow\infty$ simultaneously with
$n/d\rightarrow \alpha$, cf. L123.
I guess the setup is the following.
Let $(n_l)_l$ and $(d_l)_l$  be two nondecreasing, unbounded sequences of integers such that
$\frac{n_l}{d_l} \rightarrow \alpha$ for $l\rightarrow\infty$. Furthermore, let $X(l)\in\mathbb{R}^{n_l \times d_l},\ldots,\theta(l)\in\mathbb{R}^{d_l},b(l)\in\mathbb{R}^k$
be such that the conditions in L143 hold. Then for $l\rightarrow \infty$, we have
$\frac{1}{d_l}|\log \mathcal{L}(W(l),\theta(l),b(l)) - \log\tilde{\mathcal{L}}(W(l),\theta(l),b(l))| \rightarrow 0$.

That this is the intended setup is indicated by LL142, where $b(n)$ and $b(d)$ appear (this is also confusing due to two different parametrizations).

A similar issue appears in (20), (23), (24), (29), (30), Lemma 1 (in (52), cf. the corresponding comment), and (61).

Q1.2 Proof strategy for Theorem 1
In L519, it is stated that "For the following we will prove a stronger result". But how does Theorem 1 follow from (32)? In particular, (32) involves the gradient of the likelihood, whereas (4) concerns the likelihood itself. The connection between the two is not clear at all.

Furthermore, the term involving $\eta_1$ in (5) seems plausible (after doing a brief, elementary calculation), but it is not clear at all how the form of $\eta_2$ appears.

Q1.3 Equ (39)
How exactly does it follow from (38)? It seems to be a rather lengthy computation.
Similarly for equ (40) and (41).

Q1.4 L523, "The first term is exactly $\eta_1$"
This does not seem to be the case. $\alpha$ is defined as the limit of $n/d$ for $n,d\rightarrow\infty$, but in (33) (using $\frac{1}{\sqrt{d}}=\sqrt{\frac{n}{d}}\frac{1}{\sqrt{n}}$) only $n/d$ appears, but not its limit. Furthermore, (33) considers the likelihood from (3), not its gradient, so how does this help to show (32)?

Q1.5 Equ (34)
* First line: What happened to $P_h(h)$ from (33)?
* Second line: Where is the second term in the sum inside the exponential inside the integral?
* Third line: How does this follow?
* Fifth line: How does this follow? Where does the exponential come from?

Q1.6 Equ (35)
Where does the $P_v$ in (35) come from? It has been integrated out in (34)

Q1.7 Equ (38)
How does the second equality follow? Where does the exponential come from?

**Question 2** Related to Theorem 2
The following aspects are relevant for *Clarity* and *Quality*

Q2.1 Equ (17)
How is it ensured that $B_t^{-1}$ is invertible?
Does the minimum in the definition $\tilde{h}^t$ exist? Is it unique? If it is not unique, which minimizer is chosen?

Q2.2 Assumptions
$\eta_1$ and $\eta_2$ are defined in Theorem 1 analytically, so these functions are either continuously differentiable or not. So are they continuously differentiable or not? And where exactly is this used in the proof of Theorem 2? It seems that only their differentiability is used (to define $\nabla \eta_1$, $\nabla \eta_2$).

Q2.3 Theorem 2, "at the fixed point"
Does a fixed point always exist, and if so, is it unique? This is not discussed, and no reference is given.

Q2.4 Equ (18)
Where does $\alpha$ come from? Using the definition of $Q(W)$ from L153, there should be $1/d$ instead.

**Question 3** Related to Theorem 3
The following aspects are relevant for *Clarity*, *Quality* and *Significance*

Q3.1 Choice of $\psi$
What would be an interesting PL(2) function $\psi$ in this context?
Is the choice in L279 in the context of Theorem 4 the only relevant one?

Q3.2 Limit notion
And what can be inferred from the Cesaro convergence (26)? In other words, which statements about the behaviour of the RBM in the asymptotic setting can be given using this limit notion?

Q3.3 Extension
Does a similar result like Theorem 3 hold also for the $u_i^\ast$?

**Question 4** Additional questions related to (significant) technical details
The following aspects are relevant for *Clarity*, *Quality* and *Rating*

Q4.1 Equ (3)
Where is the $dP_v(v)$ term? Is it just assumed that it equals 1? If not, which adjustments are needed for the remaining developments?

Q4.2 LL209
"We deal with the term $\eta_2$ by viewing it as a set of constraints and introducing Lagrange multipliers to fix them"
This is unclear and requires further elaboration. In particular, $\eta_2$ is just an additive term in the objective function of the optimization problem, so in which sense is it a constraint?
Furthermore, What is meant by "to fix them"? In general, a Lagrangian formulation is a relaxation, and one needs conditions to enforce equivalence.

Q4.3 Equ (73)
How does this follow from Theorem 3?

Q4.4 LL627
How exactly does the zero convergence stated in line L627 follow from Theorem 2? The latter does not involve the diverging factor $\sqrt{d}$, and since Theorem 2 does not provide any convergence rate, it is not clear that the convergence is fast enough to overcome the growth of $\sqrt{d}$.

Q4.5 Equ (82)
* L653, "Then it suffices to map (81) to a general AMP iteration as the following"
	Which general AMP iteration? The one in (56), (57)?
* Theorem 5 uses $t\geq 0$, but some subexpressions are in (82) are not defined for $t=0$.
* Something seems to be off with the mapping to the general AMP iteration. (25) can indeed be rewritten as (81), however, it is not clear how to get from (81) to (82). In particular, when plugging in the various quantities in (82), $\hat{W}^t$, $\overset{\circ}{W^t}$ etc get mixed up. More details need to be provided on how to perform this mapping.

**Ethical Concerns:**

["NO or VERY MINOR ethics concerns only"]

**Final Justification:**

As stated in the initial review, the present manuscript forms a solid contribution to a relevant and active field of machine learning. While the submission contains quite a few issues (primarily typos and imprecisions), the authors have been able to address all of my major concerns, cf. Questions 1-4. For these reasons, I recommend acceptance and raise my overall score to 5. However, I would like to emphasize that this evaluation assumes that the authors implement all the discussed changes.

**Limitations:**

In general yes. However, while the setting used is classic, its inherent limitations should be briefly stated.
* Apart from Theorem 1, mostly a synthetic data setup is considered, which might be unrealistic from a practical perspective.
* This work considers a particular asymptotic setting. While interesting, it is still an approximation, and since no convergence rates are given, it is unclear when this approximation can be considered sufficiently accurate.

**Quality:**

2

**Strengths And Weaknesses:**

This work contains a thorough theoretical investigation of RBMs in a classic asymptotic setting, with a particular focus on AMP and gradient descent, both under the Spiked Covariance Model. This is a relevant and active area, and the aim of the authors to provide a rigorous analysis (instead of using heuristics) makes the work particularly relevant. Furthermore, the proposed AMP algorithm in the asymptotic setting might be of independent interest as a new training approach to RBMs.

As clearly stated in the manuscript, using AMP to analyse RBMs is not new, and essentially all of the techniques used are established (though might be relatively recent, for example the developments in Section D), though making everything rigorous would still constitute a very relevant contribution.

Unfortunately, the work suffers from many missing details and imprecisions, cf. the detailed comments and suggestions below and the questions, in addition to typos and some language problems. Since the explicitly stated goal of the present work is to provide a rigorous asymptotic analysis, the imprecisions and missing details are a major problem, and indeed quite a few steps and results were difficult to check for the present reviewer. If these problems are addressed and resolved (which would require a careful revision of the manuscript), this work can be a very interesting and promising contribution to the analysis of RBMs, with potentially considerable follow up work.

**Detailed comments and suggestions**
L10
Language, "via the dynamical mean-field theory" is probably more common

L11, 13
Language, "RBMs" is probably better in terms of grammar

L14
Abbreviation BBP not introduced (in contrast to all other abbreviations in the abstract)

L20
Potentially ambigous sentence: Not state of the art for image generation anymore, but for other tasks? Made add an "for example"

L36
Grammar, "yields"

L38
Language, a connecting word is missing

L59
Unclear, does "mathematically" refer her to general analytical techniques (including formal derivations), or rigorous results?

L64
"BBP threshold" has not been introduced yet. In fact, the first reference regarding this is given only in L249.

L107
Typo, "Spiked"

Section 2
There are some imprecisions / missing details
* What are $\theta$ and $\beta$? These quantities are never introduced
* How exactly is the data $X$ used? After its introduction in L113 it is never explicitly mentioned anymore. I guess it enters (3) via $x_\mu$, the rows of $X$ according to the convention at the end of the section.

L113
Something is wrong here (superfluous "using"?)

LL115
Something is wrong here. $W$, $\theta$, $b$ appear to be trainable parameters, $v$ and $h$ are data ($v$ visible, $h$ hidden)

L118
Why is "prior" in quotes?

L118
Maybe add ", respectively." at the end for clarity

Equ (3)
* Is $Z^n(W,\theta,b)$ actually the quantity from (2) raised to the $n$-th power? If so, maybe use $Z(W,\theta,b)^n$ instead to avoid confusion.
* I strongly suggest to use $h$ instead of $h_\mu$ since this is a free variable (it is integrated over), so using an index can lead to confusion.

Equ (3)
Maybe use a different symbol than $\mu$ for the running index. Since you work a lot with probability distributions, and $\mu$ is one of the standard symbols for probability measures, this might lead to confusion.

L124-126
Maybe include examples to illustrate these conventions.
Furthermore, I strongly recommend to introduce these conventions before they are used for example in Equ (3).

Assumption 1
* Is separability an established terminology in the present context? Because technically, this just means that the random variables $v_1,\ldots,v_d$, where $v \sim P_v$, are independent (the density w.r.t. Lebesgue measure of their law factorizes)
* What is meant by "a bounded distribution"? Bounded support? The density is a bounded function? Or do you mean by "distribution" any density w.r.t. Lebesgue measure, leading to a (nonnegative) measure that is not necessarily a probability measure?

Equ (6)
Incompatible dimensions (wrong use of transpose). The first argument of $\eta_1$ in (5) needs $W^\top x_\mu$, not $x_\mu^\top W$

L147
Inconsistent use of commas, "i.e." and "i.e.,", "e.g." and "e.g.,". The remainder of the manuscript should also be checked for this issue.

L148
Grammar, missing article

Equ (8)
Where exactly is the proof of this result?

L154-157
A reference should be given for this.

Equ (9), (10)
The redefinition of the symbols on the lefthand side in these equations is potentially confusing, maybe choose different symbols.

L155, 157
While the notation $\langle\cdot\rangle$ is very common in statistical physics, it rarely appears in statistics and (theory of) machine learning, so it should be briefly introduced somewhere.

L181
Wording, something is wrong here. Maybe "slightly generalized"?

Equ (12)
Which matrix norm is used here?

L187, "then we have"
Better to use $\approx$ instead of $=$

L197
Wording. Maybe "the teacher-student setting"?

L204
Typo, duplication of "now"

L204
This can be made a bit more precise, e.g., "following the spiked covariance model (11)".

Equ (15)
Technically, $\eta_1$ and $\eta_2$ with just one argument are not defined (I guess it corresponds to $\eta_1,\eta_2$ from Theorem 1 with the remaining arguments set to zero)

Equ (15)
Minor remark: Optimization problems are usually formulated with $\max$, and $\mathrm{argmax}$ is only used when one defines a solution to it.

L205
Wording, "direct" is not appropriate here. Similarly in L237

LL206, "While it is common to use AMP for supervised generalized linear regression"
Reference?

L209
* Grammar, use singular "aspect is", or "Other new aspects"
* Typo, "penalty"

L210
Technically, the following display is not the new loss, but rather an optimization problem with the new loss.

LL212
Confusing. Is Algorithm 1 already "AMP-RBM", or is the latter a specific instance of the former?

Algorithm 1
On which AMP variant is Algorithm 1 based? It is not the one from Montanari and Venkataramanan, 2021 (e.g., there is an additional "damping term" here)

L213
Grammar, "the denoisers"

L219
Grammar, missing "that"

L220
Wording, better to use "Let ... be the"

Figure 1, caption
Typo, "initialization"

L228
The notation $U_{[1:r]}$, $W_{[1:r]}$ is not defined. Furthermore, what does $[G]_1^k$ mean?

Theorem 3 / Footnote 1, "Lipschitz continuous, uniformly at"
This terminology is not quite standard, so it might be better to defined it explicitly in Theorem 3.

LL244
A reference regarding the weak recovery threshold should be added here.

LL251, "AMP-RBM as well as RBM (because their fixed points match)"
Where exactly does this follow from?

L275
Inconsistent use of cases, "Assumptions".
Same issue in L287.

Corollary 1
What does "can be diagonal" exactly mean?

L282
Typo, "techniques"

L289
It would be nice to provide a pointer to Section C.5 with the proof of Corollary 1 for the reader's convenience.

L303
Grammar, "preferred to"

Equ (25)
How exactly is this equation related to what came before?

L317, "converges to $\hat{W}$ uniformly,"
Uniformly w.r.t. what?

L319
Wording, better to use something like "corroborated" or "illustrated" instead of "verified"

References
The reference entries should be carefully checked, upper / lower case use is inconsistent with the main text (e.g., boltzmann instead of Boltzmann in Salakhutdinov et al '07)

L502
The inverse temperature $\beta$ is not introduced yet, the paragraph needs to be adapted accordingly.

L503
Missing comma

L505
What is $\beta^\ast$? I guess it refers to L192

L512
Missing comma

L512
"maximizes... instead" Reference / derivation?

L514, "impossible for inference"
Language, something is wrong here

L514-516
Maybe refer to the location of this result in the main text

L519
Missing comma, or use "the following"

L522
Where are the arguments $\theta$, $b$?

L523, "the derivative of the second term"
Second term of what?

Equ (35)
This uses the separability from Assumption 1. This should be made explicit.

Equ (36)
$\mathcal{Z}$ not defined
How does the last line follow?

L529, "(36) also directly proves (4)."
How? (4) does not involve any derivative.

L535
Grammar and Language

L536, "The gradients can be dealt with in a similar way"
What exactly does this mean?

L547
Missing comma

L552
Typo

Assumption 3
Is this assumption new? Or has it been used in the literature before? It is not clear what the relation to the references in LL560 is.

L555
What is $\|\mathcal{F''}\|$? Supremum norm? Measurable essential supremum?

Equ (5)
Inconsistent / undefined notation. I guess $\mathcal{F}^{1}=\mathcal{F}'$.

L557
Wording. Better to write something like "i.e., the data are well-normalized"

L559
Language

L564
What exactly is meant by "normalized" here? That the matrix is normalized?

L564
What is "the error matrix"?

Equ (52)
There is an imprecision here. In general, $n/d\not=\alpha$, only $n/d\rightarrow\alpha$, cf. also the comments and questions regarding Theorem 1.

L570
How exactly does $\mathbb{E}[\tilde Z_{ij}]=1$ follow from Assumption 3 (presumably last part of (ii))?

L572
Inappropriate notation. I guess $L^2$ refers to the supremum norm (the $\ell_2$ norm for a matrix space would be the Frobenius norm).

Section C.2
* The AMP variant outlined here is not the one from Montanari and Venkataramanan, 2021, Supplement Section J, since here the full history of the iterates are considered in the denoising functions. On which AMP scheme is this section based? It seems to be (a variant of) Zhong et al '24, Section 2.2
* Inconsistent notation for Onsager coefficients (subscript with and without comma)

Equ (56) and Equ (57)
$F^t$, $E^t$ in (56) and $F$, $E$ in (57 are not defined. I guess these are generic side information, cf. Zhong et al '24, Section 2.2

Theorem 6
* The AMP variant and the setup is sufficiently different from Zhong et al '24 to require a full proof.
* The wording of this result could be improved.

L592
Grammar, use plural

L604
Typo, "Lipschitz"

Equ (64)
Some details on how to derive the second equation should be given. I guess the argument is as follows: By definition, $h_t$ is a stationary point of the objective function in the minimization in (17), so the gradient of this objective function at $h_t$ is zero, and it can be rearranged for the latter, and in can be plugged in the update rule for $U^t$ in Algorithm 1. Finally, the update rule can be rearranged for $h_t$, and this expression is plugged in as the argument for the derivative of $\eta_1$.

L614, "Moreover, at the fixed point we have..."
How? I guess it follows from the last line of Algorithm 1.

L625, "which suggests that the limit in (73) exists"
Imprecise. Does the preceding constitute a rigorous proof or not?
Similar issue in L627

Equ (74)
Missing closing bracket

Equ (76)
Superfluous opening bracket. Furthermore, the multiplicative factor needs to be $\alpha^{-1}$

L653, "as the following"
Wording, I guess what is meant is "as follows"

L641
Wording

L673-679
Putting this part into an Example environment would improve readability.

L681, "we need to give some notations"
Wording, better to use something like "we need to introduce some notation"

L696, "Now we present our main results,"
Problematic formulation since this is part of the supplementary part and only briefly mentioned in the main text. Maybe use "main results in this section" or something similar.

Equ (96)
What is $A$? I guess it is supposed to be $\mathcal{A}$, similar to $\mathcal{A}^\ast$ in (8) in Viluchio et al '25

LL712
What exactly is meant by uniformly here?

(102), second line
Something is wrong here. I guess in the second term in the sum a $w$ is missing (and indeed it appears in (103))

L726
Why can Lemma 4 be used here? It requires minimization / maximization over compact sets.

L727
What is the resulting direction of $f$, and how does (104) then follow from it? More precisely, how does the second term in the first line of (104) follow in this way?

(105)
$\phi$ is not included in the min / max anymore

L734-736
How exactly does this follow from the (strong) law of large numbers?

L740 Typo

LL753
How exactly does this follow?

LL793
Terminology, "contrastive divergence"

(137)
Problematic notation. $\zeta_{ij}$ is already defined, (137) is just an estimation based on SE. Maybe use $\hat\zeta_{ij}$ instead.

L827
Typo, "realistic"

Figure 4
What are the "error bars"? Standard deviation?

LL835, "AMP-RBM might fail to converge for k= 10,"
Why?

L851, "corresponding to Figure 5"
I guess it means something like "with the same setting" or "analogously"

L858, "decimation AMP"
What exactly is this?

---

> ### Author Rebuttal · Authors · 2025-07-27
>
> Thank you very much for your careful reading. We will carefully revise the manuscript to improve the language, notations and add references according to your suggestions. Due to the length limit of the rebuttal, we focus below on addressing the main issues. Please let us know if there are any remaining unclear points.
>
> Q1.1. **Lack of precision**: There are indeed some typos in Theorem 1. We should regard everything as a function of $d$ with $d\to\infty$, and thus we should write $X(d),W(d),\theta(d),b(d)$ etc. Moreover, although it is possible to consider $n(d)/d\to\alpha$, we decide to define $n(d)=\alpha d$ with a constant $\alpha$ to avoid unnecessary techinical difficulties.
>
> Q1.2. **Proof strategy for Theorem 1**: We cannot directly obtain eq.(4) from eq.(32). Eq.(4) is obtained by combining eqs.(33) and (36). The first term of $logL$ remains the same, and the second term is $logZ$, which is equal to $\eta_2+o(1)$ by eq.(36). We prove eq.(32) because it is useful later. See Q1.5 for the calculation of $\eta_2$.
>
> Q1.3. **eq.(39)**: In eq.(39), when we take derivative w.r.t. $\theta$, the denominator is $Z(W)$, and we can use eq.(36). The nominator is eq.(38). The right side of eq.(38) divided by the right side of eq.(36) is exactly the derivative of $\eta_2$ w.r.t. $\theta$, which gives eq.(39). eqs. (40) and (41) are similar.
>
> Q1.4. **L523**: We decide to directly define $n(d)=\alpha d$, and then it follows. Then we take the derivatives of eq.(33), which is eq.(34), and thus we can prove eq.(32) which only concerns the derivatives of eq.(33).
>
> Q1.5. **eq.(34)**: In eq.(34) we do not consider the integral over $P_h$, and we add this integral back in eq.(35) and exchange the integral and derivative. There is a typo in the second line. We should have the $v_i\theta_i$ term. The third line comes from Taylor’s expansion: $e^{(v_ih^Tw_i+v_i\theta_i)/\sqrt{d}}=1+(v_ih^Tw_i+v_i\theta_i)/\sqrt{d}$+higher order terms. For the fifth line, we also use the Taylor’s expansion $e^{(h^Tw_i+\theta_i)^2/2d}=1+(h^Tw_i+\theta_i)^2/2d$+higher order terms.
>
> Q1.6. **eq.(35)**: It is a typo. There should not be the integral over P_v.
>
> Q1.7. **eq.(38)**: The right side of eq.(38) is obtained by first integrating over $P_v$ and then using $e^{(h^Tw_i+\theta_i)^2/2d}=1+(h^Tw_i+\theta_i)^2/2d+$higher order terms.
>
> Q2.1. **eq.(17)**: If $B_t$ is not invertible, the algorithm fails. Theorem 2 is based on an implicit assumption that AMP-RBM has a fixed point, and then $B_t$ is invertible at that point. In terms of the minimizer, it might not be unique, and we can choose any one as long as f,g are Lipchitz continuous (required by Theorem 3). We make this explicit in L688-690.
>
> Q2.2. **$\eta_1$ and $\eta_2$**: $\eta_1$ and $\eta_2$ defined in Theorem 1 are continuously differentiable, but our results in Section 5 hold for more general $\eta_1$ and $\eta_2$. It is true that the differentiability is only used to define their derivatives and to obtain the property of the minimum in eq.(17).
>
> Q2.3. **fixed point in Theorem 2**: A fixed point might not exist and might not be unique. The existence of a (stable) fixed point is related to the replicon stability condition (L269-273). There might be more than one fixed point (L295-299).
>
> Q2.4. **eq.(18)**: Notice that in eq.(15) it is $n\eta_2$, so after taking derivatives it becomes $n/d=\alpha$.
>
> Q3.1. **choise of $\psi$**: Except the choice in L279 (the overlap), we can also choose $\psi$ to be the L1 or L2 distances between $W^t$ and $W^\*$.
>
> Q3.2. **Limit notion**: The convergence (24) can e.g. give the overlap, L1 or L2 distances between RBM’s saddle points $\hat{W}$ and the signal $W^\*$ by choosing different $\psi$.
>
> Q3.3. **Extension of Theorem 3**: We can write a similar arguments for $u^\*$. See eq.(61).
>
> Q4.1. **eq.(3)**: Eq.(3) is the standard definition of RBMs' likelihood. Note that the visible units are actually the data $X$, so there should be a $P_v(x)$ term. We drop this term because it does not depend on the training parameters (L121-122).
>
> Q4.2. **L209**: Our original phrasing was imprecise. To clarify: we introduce the Lagrange multiplier to decouple the joint optimization over $Q$ and $W$ and enable alternating optimization.
>
> Q4.3: **eq.(73)**: By the definition of $M_t$ and the Stein’s lemma, we have $M_t=E[Z^tW^\*]/E[(W^\*)^2]$. By eq.(61) we have $E[Z^tW^\*]=\sum_iz_i^t(w_i^\*)^T/d$. Finally we can replace $z_i^t$ with $\hat{z}_i$ by Assumption 2. By eq.(74) the limit on the right side of eq.(73) exists, and thus the limit on the left side of eq.(73) exists.
>
> Q4.4. **L627**: There is a typo in eqs.(32) and (42). It should be $\sqrt{d}$ rather than $1/\sqrt{d}$ (eqs.(37) and (39) are correct). Theorem 2 suggests that $\nabla\log\tilde{L}(\hat{W}(d))=0$ for any $d$, and thus eq.(32) gives the desired results.
>
> Q4.5. **eq.(82)**: Yes, one can find that eq.(81) is indeed a special case of the general AMP eq.(56). Moreover, one can verify that eqs.(81) and (82) are equivalent using the definition eq.(77). Notice that there is no $\mathring{W}$ and $\mathring{Z}$ in eqs.(81)-(82)! When we take eq.(77) into eq.(81), we only have $\tilde{W}$ and $\tilde{Z}$ as the arguments.
>
> The following are additional questions in the weakness part:
>
> 1. **L115**: There is a typo. $\theta, \beta$ are the learnable biases and $v,h$ are visible and hidden units.
>
> 2. **L118**: $P_v$ and $P_h$ are called the “priors”, because the real joint distribution of v,h is given by eq.(1).
>
> 3. **Assumption 1**: We will say that $P_v$ is a factorizable distribution with a bounded support.
>
> 4. **eq.(8)**: It is exactly the gradient of eq.(5) if we set the biases to 0.
>
> 5. **eq.(12)**: $||.||$ should be $||.||_F$.
>
> 6. **eq.(15)**: We will add the definition of $\eta_1,\eta_2$ with one argument (i.e. eqs. (6),(7) with other arguments set to 0), and replace argmax with max.
>
> 7. **Algorithm 1**: We refer to Algorithm 1 as "AMP-RBM". While inspired by Montanari and Venkataramanan (2021), our key modification is the introduction of the Lagrange multiplier $\hat{Q}$. The proof in Appendix C primarily builds on Fan (2022) to handle memory terms, which is necessary for analyzing GD dynamics (Theorem 5). However, Algorithm 1 itself can be analyzed within the simpler framework of Montanari and Venkataramanan (2021).
>
> 8. **L228**: We define $U_{[1:r]}$ to be the first $r$ elements of $U$. $[G_i]_{i=1}^k$ are iid Gaussian (i.e. G is a Gaussian random vector).
>
> 9. **L251**: Theorem 2 suggests that the fixed points of AMP-RBM and RBM match.
>
> 10. **Corollary 1**: We will state that when we initialize the SE (eq.(19)) with diagonal overlaps, then it remains diagonal. Thus, if it converges, the matrices in Theorem 4 (e.g. $\bar{C}_\infty$, $M\_\infty$) are diagonal.
>
> 11. **eq.(25)**: It is the GD over $log\tilde{L}$, which comes from eq.(8).
>
> 12. **L317 and L712**: “uniform convergence” means the same thing as Assumption 2.
>
> 13. **L502-505**: $\beta$ is defined in eq.(31), which we will move to the beginning. We will also mentioned that $beta^*$ is defined in eq.(14).
>
> 14. **L512**: By eq.(31), the posterior of $W$ concentrates around the maximizer of $\sum_\mu x_\mu^TWh$ when $\beta$ goes to infinity.
>
> 15. **L514**: Previous literature predicts that the system is in the spin glass phase and it is statistically intractable to infer the planted signal $W^\*$. We will refer to the weak recovery threshold in eq.(22).
>
> 16. **L522**: There is a typo. It should be $log L(W,\theta,\beta)$.
>
> 17. **L523**: it should be "the second term of eq.(34)".
>
> 18. **eq.(36)**: It should be $Z$ rather than $\mathcal{Z}$, which is defined in eq.(2). For the last line, we integrate over $v_i$, and use $1+(h^Tw_i+\theta_i)^2/2d\approx e^{(h^Tw_i+\theta_i)^2/2d}$.
>
> 19. **L536**: It means that the gradient of logL can be dealt with similarly to eqs.(34-35) when $P_v$ has non-zero mean.
>
> 20: **Assumption 3**: Assumption 3 is from Assumptions (H2),(H3) in Guionnet et al'23 and Hypothesis 2.1 in Mergny et al'24. We refer to these literatures for potential generalization.
>
> 21: **L555**: |F’’| means the supremum of F’’, and indeed $F^{(1)}=F’$.
>
> 22: **L564**: “normalized” means that each element has unit variance. We call $E$ “the error matrix” because it vanishes in high dimensions, and thus the data $X$ look like spikes plus iid matrix $\tilde{Z}$.
>
> 23. **L570**: We obtain $E[\tilde{Z}_{ij}^2]=1$ from eq.(53), using $\theta_0(F^2)-\theta_0(F)^2=1$ (Assumption 3(ii)).
>
> 24: **L572**: We refer to the $L^2$ norm of a matrix rather than the Frobenius norm.
>
> 25: **Section C.2**: The AMP here can be regarded as a variant of Zhong et al’24. Following their notations, we add a comma in the subscript if necessary, e.g., we write $C_{ti}$ and $C_{t+1,i}$. $E^t$ and $F^t$ are generic side information of finite dimensions, which almost surely converges to $E,F$. Theorem 6 is close to Zhong et al’ 24. The main difference is to include the non-linearity F, which is dealt with in Mergny et al'24. Another difference is to generalize the side information from E,F to E^t, F^t, which is quite straight forward.
>
> 26. **L727**: In eq.(103) we optimize over f with fixed $||f||=\kappa\sqrt{d}$. Thus $c^Tf$ reaches its maximum $||c||\kappa\sqrt{d}$ when $f=\kappa\sqrt{d}c/||c||$, which gives the second term of eq.(104).
>
> 27: **eq.(105)**: There should be a supreme over $\phi$.
>
> 28: **L734-736**: By definition, $\epsilon_d$ is the sum of means of independent random variables, which converges almost surely by the law of large numbers.
>
> 29. **L753**: Because a family of pseudo-Lipschitz continuous functions on a bounded domain is necessarily equicontinuous (by the Arzelà-Ascoli theorem).
>
> 30. **Figure 4**: Error bars represent the standard deviation over 10 runs.
>
> 31. **L835**: The optimization problem eq,(17) becomes difficult to solve for k=10, and might suffer from numerical instability.
>
> 32. **L858**: Decimation AMP refers to the AMP algorithm in Pourkamali et al.'24 (L852).

---

> > ### Comment · Reviewer_qSCD · 2025-08-01
> > **Answer to rebuttal**
> >
> > Thank you very much for taking my questions and comments into account. Here are some clarifying questions and comments.
> >
> > **Q1.1**
> > To clarify: In the whole manuscript, you work (or you intend to, and some changes need to be made in the revision) in the setting $d\rightarrow\infty$ and given a constant $\alpha$, you set $n=n(d)=\alpha d$? This would resolve e.g. my Q1.4
> >
> > **Q1.2**
> > To summarize: You start with (33), leading to two additive terms. Using $n(d)=\alpha d$, the first term can be identified with the $\eta_1$ term. For the second term, (36) is applied, the first term in the last equation of the latter is identified by $\eta_2$. In particular, (34) and (35) are not used for the calculation of $\eta_2$. Is this correct?
> >
> > **Q1.3**
> > What is meant by denominator and nominator? I cannot identify a fraction in (39).
> > Furthermore, the steps outlined in your answer sound indeed like a longer calculation, and I would highly recommend to explicitly state them in the proof.
> >
> > **Q1.5**
> > Some of the answer is not clear to me.
> > * Regarding the fifth line of (34), how is the Taylor expansion you stated in your answer used there? In particular, around which point is the Taylor expansion done, and how does the corresponding update of the $\mathcal{O}(\cdot)$ term follow?
> > * Why can you interchange integration and differentiation in (35)?
> > * How is the calculation you outline here used to get $\eta_2$ as you stated in your answer to Q1.2?
> >
> > **Q1.7**
> > I recommend to include this explanation in the proof in the manuscript.
> >
> > **Q2.1**
> > Thank you for the clarification. Ideally, this should be mentioned in the main text.
> >
> > **Q2.3**
> > Thank you for the explanation. Including it in the main text might be helpful for the reader.
> >
> > **Q2.4**
> > Could you point out where exactly the right form of $\nabla \eta_2$ is stated, so that the term $n/d=\alpha$ appears?
> >
> > **Q3.1**
> > Thank you for the clarification. It might be helpful for the reader to state this explicitly somewhere.
> >
> > **Q3.2**
> > I guess by L1, L2 you mean the usual $\|\cdot\|_1$, $\|\cdot\|_2$ norms on $\mathbb{R}^n$. How exactly would get the corresponding convergence (24)? Simply choosing $\psi$ as these distances does not seem to work due to the Cesaro limit notion.
> >
> > **Q3.3**
> > Thank you for pointing this out.
> >
> > **Q4.1**
> > I might have overlooked something, but if $P_v$ is not constant, then this term (evaluated over the data) acts as a weighting term in the likelihood, which in turn influences the optimization outcome (i.e., the weights). Is this correct? If so, why can this term then be dropped?
> >
> > **Q4.3**
> > This seems to be a routine, but not trivial argument. I strongly recommend to include it explicitly in the proof.
> >
> > **Additional remark**
> > If my understanding of your answer to Q1.2 and Q1.4 is correct, then the proof of Theorem 1 contains actually another result (related to the gradient of the likelihood function). I strongly recommend to state this result explicitly, and ideally prove it separately (this simplified the proof of Theorem 1 considerably).

---

> > > ### Author Response · Authors · 2025-08-01
> > > **Reply to further questions**
> > >
> > > Thank you for your further comments. We appreciate your careful reading of our manuscript and have addressed all your concerns point-by-point below. Please don't hesitate to let us know if any clarification is needed.
> > >
> > > **Q1.1** Yes, your understanding is correct. We will work in the setting $n(d)=\alpha d$ with a constant $\alpha$ and $d\to\infty$. This will simplify a lot of calculation.
> > >
> > > **Q1.2**  Yes. Your understanding is completely correct. eqs. (34) and (35) are used to prove eq. (32) rather than eq. (4). $\eta_2$ is exactly the logarithm of the right side of eq.(36).
> > >
> > > **Q1.3** We refer to the following calculation
> > >
> > > $$\frac{\partial}{\partial \theta_i}\log\int dP_h(h)dP_v(v)e^{\frac{1}{\sqrt{d}}v^\top Wh+\frac{1}{\sqrt{d}}v^\top \theta+h^\top b}=\frac{\frac{\partial}{\partial \theta_i}\int dP_h(h)dP_v(v)e^{\frac{1}{\sqrt{d}}v^\top Wh+\frac{1}{\sqrt{d}}v^\top \theta+h^\top b}}{\int dP_h(h)dP_v(v)e^{\frac{1}{\sqrt{d}}v^\top Wh+\frac{1}{\sqrt{d}}v^\top \theta+h^\top b}}$$
> > >
> > > By eq.(36), the denominator is $\int dP_h(h)e^{\frac{1}{2d}\sum_{i=1}^d(h^\top w_i+\theta_i)^2+h^\top b}+o(1)$. By eqs. (36) and (38), the numerator is $\int dP_h(h)\frac{\partial}{\partial \theta_i}e^{\frac{1}{2d}\sum_{i=1}^d(h^\top w_i+\theta_i)^2+h^\top b}+o(1/d)$. Thus we have $\frac{\partial}{\partial \theta_i}\log\int dP_h(h)dP_v(v)e^{\frac{1}{\sqrt{d}}v^\top Wh+\frac{1}{\sqrt{d}}v^\top \theta+h^\top b}=\frac{\partial}{\partial \theta_i}\eta_2+o(1/d)$, which gives eq.(39). We will surely include the above calculation in the proof.
> > >
> > >
> > > **Q1.5**
> > > We refer to the following expansion
> > > $\frac{\partial}{\partial w_{ai}}e^{\frac{1}{2d}(h^\top w_i+\theta_i)^2}=\frac{1}{d}h_a(h^\top w_i+\theta_i)e^{\frac{1}{2d}(h^\top w_i+\theta_i)^2}=\frac{1}{d}h_a(h^\top w_i+\theta_i)+O((h^\top w_i+\theta_i)^3/d^2)$.
> > >
> > > To update the $\mathcal{O}(\cdot)$ term, we use the inequality $(h^\top w_i+\theta_i)^2\leq2(|\theta_i|^2+||h^Tw_i||^2)\leq2(|\theta_i|^2+||h||^2||w_i||^2)$. Then we use $\sum_{i=1}^d\frac{(|\theta_i|^2+||h||^2||w_i||^2)}{d^{3/2}}=o(1)$ by the boundedness of $P_h$ and the boundedness assumption in Theorem 1. Similarly we have $\sum_{i=1}^d(h^\top w_i+\theta_i)^3/d^2=o(1)$. This is what we mean by eq.(35). (Thus eq.(35) is not so accurate. We should sum over i and obtain an o(1) factor on the right side.)
> > >
> > > We can interchange the integration and differentiation because the integrand is continuously differentiable (L525-526). Note that by Assumption 1 the integral is over a bounded domain.
> > >
> > > To obtain $\eta_2$, we can notice that $\eta_2$ is exactly the logarithm of the right side of eq.(36). eqs. (34) and (35) are actually used to obtain the derivatives of $\eta_2$ (See reply to Q1.2).
> > >
> > > **Q1.7, 2.1, 2.3 and 4.3**
> > > Thank you for your suggestions. We will surely include them in the revision.
> > >
> > > **Q2.4**
> > > This is simply the chain rule. $n\nabla_W\eta_2(Q(W))=\frac{n}{d}W\nabla_Q\eta_2(Q(W))$, where $Q(W)=W^TW/d$. Note that $\nabla\eta_2$  actually refers to $\nabla_Q\eta_2(Q(W))$, which is different from $\nabla_W\eta_2(Q(W))$.
> > >
> > > **Q3.1** Yes, we will comment on it after Theorem 4.
> > >
> > > **Q3.2** If we choose $\psi(x,y)=\sum_{a=1}^r|x_a-y_a|$, the left side of eq.(24) is exactly $\sum_{i,a=1}^{d,r}|W_{ia}^\*-\hat{W}\_{ia}|$, which is the L1 distance between the vectorization of $W^*$ and $\hat{W}$. Similarly, by choosing $\psi(x,y)=\sum_{a=1}^r(x_a-y_a)^2$, we obtain the L2 distance. It is not related to the Cesaro limit.
> > >
> > > **Q4.1** Note that  $P_v(x)$ is a multiplicative factor that depends only on the input data $x$, not on the model parameters $(W,\theta,b)$. Since we are optimizing over $(W,\theta,b)$, maximizing $\Pi_\mu P_v(x_\mu)L(W,\theta,b)$ is the same as maximizing $L(W,\theta,b)$.
> > >
> > > **Additional remark** Your understanding is completely correct. We believe that you are referring to eq.(32), which is a result related to the gradient of the likelihood and independent of Theorem 1. We will emphasize it more.

---

> ### Comment · Reviewer_qSCD · 2025-08-04
>
> Thank you very much for your additional answers
>
> **Q1.3** Perfect, thank you. I agree that this calculation should be explicitly included, it definitely helps the reader.
>
> **Q1.5** Thank you, I agree
>
> **Q2.4** OK, thank you. However, in light of L216, this is potentially confusing. Maybe add a brief comment after the computation explaining this.
>
> **Q3.2** But in (24) there is a factor 1/d? So with the first choice of $\psi$ you gave in the preceding answer we get
>  $\frac{1}{d} \sum_{i,a=1}^{d,r}|W_{ia}^*-\hat{W}_{ia}|$
>
> and not
>
> $\sum_{i,a=1}^{d,r}|W_{ia}^*-\hat{W}_{ia}|$.
>
> My question was more targeted towards the use or interpretation of Theorem 4 and (24).
> Suppose $d$ were constant and the elements of the sequence in (24) were index by some other variable, say, $k$. With the choice of $\psi$ you gave in your previous answer, we would indeed get convergence in $\|\cdot\|_1$ or $\|\cdot\|_2$. However, in (24) we have $d\rightarrow\infty$, and the series in (24) is normalized by $d$ through the factor $1/d$.
> So in which sense do we have convergence in (24)? Or in a less formal way, how can we interpret the convergence stated in (24)?
>
> I agree that technically (24) is not a Cesaro limit, and I apologize for the confusion this caused. I was referring to the superficial similarity between (24) (the limit of a sequence of the form $1/k \sum_{i=1}^k a_{ki}$ with $k\rightarrow\infty$) and Cesaro limits (the limit of a sequence of the form $1/k \sum_{i=1}^k b_i$).
>
> **Q4.1** Just to confirm: This means that $\mathcal{L}$ in (3) is not the likelihood arising from (1) when setting $v$ to the data points in $X$ and assuming independence, but rather the likelihood after this multiplicative constant is removed?
>
> Regarding the maximization: I see the argument now and agree, thank you very much.
>
> **Additional remark** Yes, I was referring to (32), thank you for confirming this.
>
> (This comment has been edited to avoid an Openreview formatting problem)

---

> > ### Author Response · Authors · 2025-08-04
> >
> > Thank you very much for your reply. We are glad that our previous answers have addressed most of your questions, and we will incorporate the corresponding clarifications into the revised version as you suggested. Below, we address the remaining open points:
> >
> > **Q3.2** You are correct in noting that the L1 and L2 distances do not vanish. The use of (24) is to measure the overlap, L1 and L2 distances between $W^\*$ and  $\hat{W}$, but it does **not** imply convergence in L1 or L2 distance.
> >
> > Rather, the interpretation of (24) is that the **empirical joint distribution** of $(W^*,\hat{W})$ **converges in Wasserstein-2 distance** to the law of random variables $(W,f(M\_\infty W+\Sigma\_\infty^{1/2}G,\bar{\hat{Q}}\_\infty,\bar{C}\_\infty))$. This type of convergence is standard in high-dimensional statistics. See, e.g., Section 6 of Montanari et al. (2021) and references therein for further discussion.
> >
> > **Q4.1** Yes, your understanding is correct.

---

> > > ### Author Response · Authors · 2025-08-07
> > >
> > > Dear reviewer,
> > >
> > > Thank you for your constructive feedback on our paper. Since it is approaching the end of the rebuttal period and we notice that you have not updated your score, we would like to kindly check if our responses resolved the issues you raised. If you have any additional question, we are happy to clarify them and revise the manuscript accordingly.
> > >
> > > Thank you again for your valuable input!
> > >
> > > Best regards,
> > > Authors

---

> > > > ### Comment · Reviewer_qSCD · 2025-08-07
> > > > **Rebuttal discussion and updates**
> > > >
> > > > Thank you very much, you have indeed addressed all my questions so far. I just need a bit more time to reflect on it and re-evaluate.

---

> > > > > ### Comment · Reviewer_qSCD · 2025-08-09
> > > > >
> > > > > Dear authors,
> > > > >
> > > > > I have updated my evaluation. Thank you very much for the discussion and your efforts in addressing my questions and concerns.

---

### Official Review · Reviewer_KcNC · 2025-06-30

**Clarity:** 3
**Significance:** 2
**Originality:** 3
**Rating:** 4
**Confidence:** 4

**Summary:**

The paper considers RBM in the asymptotic limit where the input dimension goes to infinity but the number of hidden units remain finite, and proves that the RBM training objective is equivalent to that of an unsupervised multi-index model with a non-separable penalty in this regime. The paper then proceeds to apply and extend established mathematical techniques for the multi-index model to analyze the asymptotic behavior of RBM trained on the spiked covariance data, and derived closed-form equations describing the gradient-descent dynamics of RBM.

**Questions:**

1. The paper suggests around line 80-84 that statistical physics approaches is closer to Bayesian sampling and considers the full posterior. However, isn't their approaches equivalent to maximum likelihood when taking the temperature to 0 limit?
2. In section 4, when comparing the spiked covariance model to an RBM teacher, you need to assume U is binary and W is approximately orthogonal, these are additional assumptions on the spiked covariance model that you don’t need for the following theoretical results, is that correct? If my understanding is correct, maybe you should be more explicit about it in this section.
3. Some numerical results are present in the Appendix but not explicitly referred to in the main text. For example, when discussing the convergence of different AMP algorithms in line 295-299, it is maybe useful to point to Appendix F.1 where you showed numerical results comparing the performance of AMP with random initializations or more informed initialization schemes.
4. What are the actual distributions P_u and P_w used in your numerics? Do their forms affect the convergence of AMP-RBM?
5. I am a bit confused about the results showing overlap vs. SNR (or lambda) in the main text vs. the appendix. Just to confirm my understanding, in the main text you showed results with r=k=2, and the lines are SE corresponding to Eq. 19, which should hold irrespective of Theorem 4 and the k=r=1 assumption. In the appendix you stated that the lines are for rank-1, which I suppose refer to k=r=1, where the line now corresponds to Theorem 4, which assumes the convergence of AMP-RBM, is this correct? What would SE look like for k=r=10?
6. This may be a broad and less relevant question. This paper proposes potential simplification of the gradient of the log-likelihood in the asymptotic limit, could you perhaps discuss whether the results may provide any insights to score-based diffusion models, a very different type of generative model that is more commonly used in recent applications, and also considers approximating the gradient of the log-likelihood of data?
And some typos...
7. Line 115, the biases should be theta and b?
8. Line 117, isn’t the energy term order 1 after adding the 1/sqrt(d) normalization, so should be ‘intensive’ instead of ‘extensive’?
9. In the appendix, there should no longer be integrating over v in the second line of Eq 35.

**Ethical Concerns:**

["NO or VERY MINOR ethics concerns only"]

**Final Justification:**

The rebuttal solved my questions and I believe the paper should be accepted with detailed revision, incorporating the points raised by both me and other reviewers.

**Limitations:**

Yes.

**Paper Formatting Concerns:**

No major formatting issues.

**Quality:**

3

**Strengths And Weaknesses:**

Strengths:
The paper presents theoretical results on the sharp asymptotics of RBM. The connection established between the RBM and the unsupervised multi-index model is general and does not rely on specific data assumptions, this connection enables simplified approximations of the log-likelihood and its gradient, and could potentially be used to accelerate training, as the authors pointed out. For more precise theoretical calculations, the authors then model the inputs as a spiked covariance model, and derived the AMP iterations and its corresponding deterministic SE, and derived the asymptotics of GD dynamics. Combined with thorough numerical validation and empirical results, this paper provides rigorous theoretical understanding of the learning dynamics of RBM. The paper is also clear and well-written.
Overall, while the paper is still limited by its data assumptions (see weaknesses), it presents elegant and accurate theoretical results for the RBM and verified them numerically, extending mathematical results in supervised learning to a type of unsupervised generative model.

Weaknesses:
As mentioned earlier, the main theoretical result of this paper relies on the data assumption, i.e., the spiked covariance data model, which is a relatively simple model for unsupervised learning. The authors did not discuss how their results may provide insights for data beyond this restrictive assumption. I believe the paper would largely benefit from some numerical experiments showing that behavior on some benchmark datasets for unsupervised learning can be also (perhaps qualitatively) captured by the theory. For example, the authors already showed results verifying theorem 1 in the appendix using Fashion-MNIST, is it possible to approximate Fashion-MNIST using a spiked-covariance model and show comparison between AMP-RBM and GD on this data?
Also, the main technical contribution of this paper is 1) proposing the analogy between RBM and the multi-index model in the asymptotic limit 2) deriving the AMP equations and analyzing GD dynamics of RBM in the asymptotic limit and under the spiked-covariance data assumption. While the first point is novel and insightful, the second point largely build on existing work on multi-index models, and the major theoretical advancement is the extension to coupled regularization. The extension itself is relatively straightforward technically as the number of hidden units is finite. While this is not necessarily a weakness, I do think that because the technical advancement is relatively straightforward, there should be more emphasis on how the new theoretical result on RBM can provide new insights. For example, the paper would benefit from a discussion highlighting properties predicted by the RBM theory that are not in the multi-index model.
The paper could also benefit from a conclusion/summary section to discuss the potential impact and possible extensions of the work.

---

> ### Author Rebuttal · Authors · 2025-07-27
>
> Thank you for the detailed feedback. We will address both the weaknesses and the questions below.
>
> * *As mentioned earlier, the main theoretical result of this paper relies on the data assumption, i.e., the spiked covariance data model, which is a relatively simple model for unsupervised learning. The authors did not discuss how their results may provide insights for data beyond this restrictive assumption. I believe the paper would largely benefit from some numerical experiments showing that behavior on some benchmark datasets for unsupervised learning can be also (perhaps qualitatively) captured by the theory. For example, the authors already showed results verifying theorem 1 in the appendix using Fashion-MNIST, is it possible to approximate Fashion-MNIST using a spiked-covariance model and show comparison between AMP-RBM and GD on this data?*
>
> We appreciate the suggestion and agree that extending beyond the spiked covariance model is important. Most of our theoretical results (Theorems 2-5) do rely on this assumption, but we emphasize that the connection between RBMs and multi-index models holds for more general cases (Theorem 1), which paves the way for future generalizations.
>
> In Appendix F.2, we provide a preliminary analysis of GD dynamics under the spiked covariance model, including the cascade of emergence phenomena and the repulsion effect between hidden units, which have also been observed in Bachtis et al. and Harsch et al. for real data. These results support the relevance of our theory.
>
> However, approximating Fashion-MNIST with a spiked covariance model is challenging: AMP-RBM does not converge reliably, and DMFT becomes invalid. We will clarify these limitations and highlight extending the theory to more realistic data as a promising direction for future work.
>
> * *Also, the main technical contribution of this paper is 1) proposing the analogy between RBM and the multi-index model in the asymptotic limit 2) deriving the AMP equations and analyzing GD dynamics of RBM in the asymptotic limit and under the spiked-covariance data assumption. While the first point is novel and insightful, the second point largely build on existing work on multi-index models, and the major theoretical advancement is the extension to coupled regularization. The extension itself is relatively straightforward technically as the number of hidden units is finite. While this is not necessarily a weakness, I do think that because the technical advancement is relatively straightforward, there should be more emphasis on how the new theoretical result on RBM can provide new insights. For example, the paper would benefit from a discussion highlighting properties predicted by the RBM theory that are not in the multi-index model. The paper could also benefit from a conclusion/summary section to discuss the potential impact and possible extensions of the work.*
>
> While our analysis builds on prior work on multi-index models, we emphasize that the unsupervised version considered here is itself novel. Previous analyses largely focused on supervised settings, whereas our work extends the framework to the unsupervised case—leading to significantly different phenomena, such as distinct weak recovery thresholds.
>
> Moreover, in Appendix F.2, we show that the RBM model captures dynamics like the cascade of emergence and hidden unit repulsion, similar to findings in Bachtis et al. and Harsch et al., but absent in standard multi-index models. We agree that identifying more such insights is a promising direction, and we will highlight this part as well as the broader implications and potential extensions.
>
> * *The paper suggests around line 80-84 that statistical physics approaches is closer to Bayesian sampling and considers the full posterior. However, isn't their approaches equivalent to maximum likelihood when taking the temperature to 0 limit?*
>
> Thank you for pointing this out. As clarified in Appendix A (L511–516), the zero-temperature limit of statistical physics approaches does not recover maximum likelihood, but instead maximizes a different objective. We will clarify this more explicitly in the main text.
>
> * *In section 4, when comparing the spiked covariance model to an RBM teacher, you need to assume U is binary and W is approximately orthogonal, these are additional assumptions on the spiked covariance model that you don’t need for the following theoretical results, is that correct? If my understanding is correct, maybe you should be more explicit about it in this section.*
>
> Your understanding is correct. The reduction of the spiked covariance model to a teacher RBM requires assuming binary U and approximately orthogonal W, but these assumptions are only used in this illustrative comparison—not in our theoretical results. We will make this distinction clearer in Section 4.
>
> * *Some numerical results are present in the Appendix but not explicitly referred to in the main text. For example, when discussing the convergence of different AMP algorithms in line 295-299, it is maybe useful to point to Appendix F.1 where you showed numerical results comparing the performance of AMP with random initializations or more informed initialization schemes.*
>
> Thank you for the helpful suggestion. We will revise the main text to explicitly refer to Appendix F.1, where we provide numerical results showing how the convergence behavior of AMP depends on initialization, in line with the discussion around Corollary 1.
>
> * *What are the actual distributions P_u and P_w used in your numerics? Do their forms affect the convergence of AMP-RBM?*
>
> As stated at the beginning of Appendix F.1, we use the Rademacher prior in our main experiments. We have also tested other distributions, including Gaussian–Bernoulli and sparse Rademacher priors. While the convergence speed of AMP-RBM varies across priors, we observe that it consistently converges to the correct fixed point at large SNR, similar to Figure 5. We will add these additional experiments and clarify this point in the Appendix.
>
> * *I am a bit confused about the results showing overlap vs. SNR (or lambda) in the main text vs. the appendix. Just to confirm my understanding, in the main text you showed results with r=k=2, and the lines are SE corresponding to Eq. 19, which should hold irrespective of Theorem 4 and the k=r=1 assumption. In the appendix you stated that the lines are for rank-1, which I suppose refer to k=r=1, where the line now corresponds to Theorem 4, which assumes the convergence of AMP-RBM, is this correct? What would SE look like for k=r=10?*
>
> Your understanding is correct. In Figures 1, 2, and 4, we present results for k = r = 2, where the lines show the SE predictions from Eq. (19), which holds independently of Theorem 4.
>
> For k = r = 10, the SE would involve a 10*10 overlap matrix, which is numerically difficult to compute. However, as noted after Corollary 1, we can assume the overlap matrix to be diagonal to find the convergent point, allowing us to reduce the SE to a scalar fixed-point equation similar to the rank-1 case. This simplification enables us to locate one of the possible saddle points of AMP-RBM, though other saddle points may also exist.
>
> * *This may be a broad and less relevant question. This paper proposes potential simplification of the gradient of the log-likelihood in the asymptotic limit, could you perhaps discuss whether the results may provide any insights to score-based diffusion models, a very different type of generative model that is more commonly used in recent applications, and also considers approximating the gradient of the log-likelihood of data?*
>
> This is an insightful comment. While the objective of score-based diffusion models differs from that of RBMs, and our approximation of the log-likelihood gradient does not directly apply, we believe similar simplifications could emerge in high-dimensional settings. In fact, vanilla diffusion models have been analyzed for high dimensional problems and linked to AMP methods (see e.g. Posterior Sampling in High Dimension via Diffusion Processes by Montanari and Wu). Extending such analyses to score-based diffusion models is a promising research direction, and we will briefly mention this connection in the revised manuscript.
>
> * *Line 115, the biases should be theta and b?*
>
> Yes. We will correct this typo.
>
> * *Line 117, isn’t the energy term order 1 after adding the 1/sqrt(d) normalization, so should be ‘intensive’ instead of ‘extensive’?*
>
> Thank you for the observation. You're correct that the exponent in Eq. (2) is of order 1—it represents the energy per data point. In this context, we use “intensive” to mean that the energy scales linearly with the number of data points n, i.e., the total energy is of order n. We will clarify this wording in the revised manuscript to avoid confusion.
>
> * *In the appendix, there should no longer be integrating over v in the second line of Eq 35.*
>
> Yes. We will correct this typo.

---

> > ### Comment · Reviewer_KcNC · 2025-08-06
> >
> > Thank you for the detailed rebuttal. It further clarifies my understanding of the contributions of this paper. I will keep my score unchanged but raise my confidence score. I have also read the other reviews' comments and agree that there are many points in the paper that could benefit from careful revision, incorporating the point raised in both of our reviews.
> >
> > Furthermore, a technical point, to push this results and increase its applicability, it would be nice to push the numerics to converge for larger k and r and check how it would deviate from the rank-1 approximation, i'm not sure what's the root of the technical difficulty to converge, so I'm not sure if it's doable within the rebuttal timeframe.

---

> > > ### Author Response · Authors · 2025-08-06
> > >
> > > We sincerely appreciate your constructive feedback and are grateful for your acknowledgment of our work. We will carefully incorporate all the valuable suggestions from the reviews into our revision.
> > >
> > > Regarding the numerical experiments with large k and r, we would like to clarify that both AMP-RBM and gradient descent (GD) perform well, as evidenced in Figure 5. The convergence challenge is specific to the state evolution (SE). Our experiments reveal that SE converges to the rank-1 solution when initialized with near-diagonal overlap matrices. However, for initializations with large non-diagonal elements, SE exhibits numerical instability. We hypothesize that this instability may stem from the presence of additional stationary points beyond those described in Corollary 1. Resolving this instability—either analytically or computationally—and characterizing these potential stationary points will be a key focus of our future research.

---

### Official Review · Reviewer_MNio · 2025-07-01

**Clarity:** 3
**Significance:** 3
**Originality:** 3
**Rating:** 5
**Confidence:** 5

**Summary:**

This work studies RBM in the limit where the input space has large dimensionality while the number of hidden units remains fixed. The authors connect RBM training objective to a form equivalent to a multi-index model with non-separable regularization. This reduction enables them to use established analytical tools from the study of multi-index models—such as Approximate Message Passing (AMP) and its state evolution, as well as the analysis of Gradient Descent (GD) via dynamical mean-field theory—to investigate RBM training dynamics. They derive rigorous asymptotic results for RBM training on data generated by the spiked covariance model, which serves as a prototypical example of structure amenable to unsupervised learning. In particular, they show that the RBM achieves the optimal computational weak recovery threshold, corresponding to the BBP transition in the spiked model.

**Questions:**

If possible, it would be interesting to provide more results on the GD part: (i) can the authors provide at least heuristic arguments for the the weak recovery threshold of GD backing up the empirical results of Fig (1) ? (ii) can the authors provide an analysis of the GD dynamics  ?

In "related works" the discussion of Bachtis et al, and Harsch et al should be improved. These works provided a heuristic analysis of the GD dynamics as a function of training time - the sentence "Their analysis is non-rigorous, whereas our results provide exact asymptotics, including the characterization of saddle points" seems to imply that this work contains a more complete analysis of GD dynamics, which is not the case.

**Ethical Concerns:**

["NO or VERY MINOR ethics concerns only"]

**Final Justification:**

I stand by my initial review + the authors replied satisfactorily. I give a 5 and not a 6 because although interesting, it is not at a ground-breaking level.

**Limitations:**

yes

**Paper Formatting Concerns:**

no issues

**Quality:**

3

**Strengths And Weaknesses:**

Strengths: Establishing the connection with multi-index model, and applying AMP to study RBM in the asymptotic limit of large dimensionality, obtaining asymptotic results for data generate by spiked covariance model.

Weaknesses: the part of the analysis of GD is weak. The asymptotic DMFT eqs are derived and proven but no analysis, even a heuristic one, is provided. The work of Bachtis et al, and Harsch et al provide such analysis in a non-rigorous setting.

---

> ### Author Rebuttal · Authors · 2025-07-27
>
> Thank you for the detailed and constructive feedback. We address the raised concerns as follows:
>
> * *Can the authors provide at least heuristic arguments for the the weak recovery threshold of GD backing up the empirical results of Fig (1)*
>
> Yes, we note in L320 that a linearization of GD dynamics can be carried out analogously to the linearization of AMP-RBM. This yields an effective update rule equivalent to the power method, suggesting that the weak recovery threshold remains unchanged. We will clarify this point in the main text by including this heuristic argument.
>
> * *Can the authors provide an analysis of the GD dynamics ?*
>
> Regarding the GD dynamics, except the weak recovery threshold, we also provide a preliminary analysis in Appendix F.2, where we recover the sequence of emergence phenomena observed in Bachtis et al. Additionally, we offer an intuitive explanation in the same appendix for why the learned weights become disaligned during training. This aligns qualitatively with the repulsion effect between place/receptive fields observed in Harsch et al. A more detailed and quantitative analysis of the GD dynamics remains an important direction for future work, and we will explicitly mention this in the revised manuscript.
>
> * *In "related works" the discussion of Bachtis et al, and Harsch et al should be improved. These works provided a heuristic analysis of the GD dynamics as a function of training time - the sentence "Their analysis is non-rigorous, whereas our results provide exact asymptotics, including the characterization of saddle points" seems to imply that this work contains a more complete analysis of GD dynamics, which is not the case.*
>
> Thank you for your suggestion. We will revise the discussion of related works to better acknowledge the contributions of Bachtis et al. and Harsch et al. In particular, we will clarify that while our work provides exact asymptotics and characterizations of saddle points, their analyses offer valuable heuristic insights into GD dynamics over time, which complement our results rather than compete with them.

---

> > ### Comment · Reviewer_MNio · 2025-08-01
> >
> > Thanks for the reply, and the proposed edits that I find fully satisfactory.

---

### Note · Authors · 2025-08-14

Dear Area Chair and Reviewers,

Thank you for the constructive discussion, which has greatly improved our work. We are encouraged by the reviewers’ recognition of the novelty and technical validity of our contributions—for example, **Reviewer KcNC** noted that our work “presents elegant and accurate theoretical results for the RBM and verifies them numerically, extending mathematical results in supervised learning to a type of unsupervised generative model,” and **Reviewer qSCD** described it as “a very interesting and promising contribution to the analysis of RBMs, with potentially considerable follow-up work.”

To the best of our knowledge, this is the first work mapping the likelihood landscape of RBMs to an effective multi-index model, yielding sharp predictions for RBM optimization and dynamics, and paving the way for extending these methods to more complex generative architectures.

We have carefully addressed the reviewers’ suggestions. In particular:

1. Further analysis of RBMs and GD (**Reviewers MNio and KcNC**): We clarified that, beyond the weak recovery threshold, we provide a preliminary analysis of RBM dynamics in Appendix F.2, recovering the sequence of emergence phenomena reported in Bachtis et al., and offering an intuitive explanation for weight misalignment during training.

2. Technical details (**Reviewers KcNC and qSCD**): We added detailed derivations, clarified assumptions, and corrected typos.

All reviewers indicated that their concerns were addressed. We will revise our manuscripts based on their advice, and we believe that these revisions will significantly strengthen our work.

We thank you again for your valuable feedback and consideration.

Best regards,

The Authors

---

### Decision · Program_Chairs · 2025-09-17

**Decision:**

Accept (poster)

**Comment:**

The paper considers RBM in the asymptotic limit where the input dimension goes to infinity but the number of hidden units remain finite. The authors then prove that the RBM training objective is equivalent to that of an unsupervised multi-index model with a non-separable penalty in this regime and  proceed by applying techniques for the multi-index model to analyze the asymptotic behavior of RBM trained on the spiked covariance data, and deriving closed-form equations describing the gradient-descent dynamics of RBMs

During the rebuttal the authors answered  satisfactorily to all concerns the reviewers had and the reviewers agreed that the paper should be acceptad with detailed revision, incorporating all points raised by the reviewers.